# Radar-based assessment of hail frequency in Europe

Elody Fluck[1,*], Michael Kunz[1,2], Peter Geissbuehler[3], and Stefan P. Ritz[3]

[1]Institute of Meteorology and Climate Research (IMK), Karlsruhe Institute of Technology (KIT), Karlsruhe, Germany
[2]Center for Disaster Management and Risk Reduction Technology (CEDIM), Karlsruhe, Germany
[3]RenaissanceRe Europe AG, Zurich, Switzerland
[*]now at: Department of Earth and Planetary Sciences, Weizmann Institute of Science, Rehovot, Israel

**Correspondence:** Elody Fluck (elody.fluck@weizmann.ac.il)

**Abstract.**

In this study we present a unique 10-year climatology of severe convective storm tracks for a larger European area covering Germany, France, Belgium, and Luxembourg. For the period 2005-2014, a high-resolution hail potential composite of $1 \times 1$ km$^2$ is produced from two-dimensional reflectivity radar data and lightning data. Individual hailstorm tracks as well as their physical properties, such as radar reflectivity along the tracks, were reconstructed for the entire time period using the Convective Cell Tracking Algorithm (CCTA2D).

A sea-to-continent gradient in the number of hail days is found to be present over the whole domain. In addition, the highest number of severe storms is found on the leeward side of low mountain ranges such as the Massif Central in France or the Swabian Jura in Southwest Germany. A latitude shift in the hail peak month is observed between the northern part of Germany where hail occurs most frequently in August, and southern France where the maximum of hail is two months earlier. The longest footprints with high reflectivity values occurred on 9 June 2014 and on 28 July 2013 with lengths reaching up to 500 kilometers. Both events were associated with hailstones measuring up to 10 cm which caused damage in excess of €2 billion.

## 1 Introduction

Severe convective storms (SCS) and related hail constitute a major atmospheric hazard. These events have the potential to cause substantial damage to hail-susceptible objects such as buildings, crops or automobiles, in various parts of Europe, including France, Germany, Austria, and Switzerland (e.g., Dessens, 1986, Puskeiler et al., 2016, Nisi et al., 2016). Prominent examples are the two hailstorms related to the low-pressure system Andreas that occurred on 27/28 July 2013 over central and southern Germany with total economic losses estimated to approximately €3.6 billion (SwissRe, 2014, Kunz et al., 2018). Hail occurs in organized convective storms (Auer, 1972) that is, multicells, supercells or Mesoscale Convective Storms (Markowski and Richardson, 2010), and results from the interaction between diverse processes and mechanisms on different spatio-temporal scales. Several authors have studied environmental conditions favoring hail production (Dessens, 1986, Houze J, 2014, Kunz et al., 2020 among others). In general, a subtle interplay between three main ingredients supports the formation of deep moist convection (Kunz, 2007): 1) Thermal instability represented, for example, by a decrease of equivalent potential temperature with height permits an air parcel to rise vertically to a considerable height (Holton, 2004) due to positive buoyancy; 2) A high

moisture content in lower atmospheric levels lowers the level of free convection (LFC) in a cloud and increases convective available potential energy (CAPE); and 3) a lifting mechanism to trigger convection such as orographic lifting (Kirshbaum et al., 2018, Barthlott et al., 2016) or lifting associate with synoptic cold-fronts (Kunz et al., 2020). Vertical wind shear is another parameter mainly relevant for the organization form of the storm and, thus, also for its lifetime, and severity. Several authors found that large hail preferably occurs in strongly sheared environments, supporting the formation of supercells (Kunz et al., 2020, Pilorz and Łupikasza, 2020). Aside of the parameters mentioned above, some authors found that an increased frequency of SCS in Europe can be associated with specific large-scale flows or teleconnection patterns (Piper et al., 2019, Mohr and Martius, 2019, Kunz et al., 2020). Several authors found for example a configuration where the East of the Atlantic Basin is dominated by a low pressure area and where France lies under a ridge (Piper et al., 2019, Fluck, 2018). This latter weather pattern favors the advection of moist and warm air on the lower atmospheric levels coming from Iberia and moving towards West Europe. Such a weather setup is termed Spanish Plume by Morris (1986). This feature is related to a trough centered over Western Europe or the Eastern Atlantic and the subsequent southwesterly winds crossing the Mediterranean Sea permits the impinging of moist and warm air masses into Central Europe. Such a configuration may increase the hail potential over western Europe by the creation of low thermal stability, a capping inversion and sometimes an elevated mixed layer (EML).

A major obstacle when investigating hail events and their climatology is the lack of accurate and comprehensive observations. This observation deficit is because of the local-scale nature of SCS and the even smaller hailstreaks with a small spatial extent (Changnon, 1977). There are only some high-density, regional-scale ground detection networks using hailpads for recording hail fall in operation, such as in southwestern and central France (Dessens, 1986, Vinet, 2001), parts of Spain (Fraile et al., 1992) or norther Italy (Eccel et al., 2012). The majority of Europe, however, remains uncovered by a hail network leading to a gap in direct hail observations. Therefore, little is known about the local-scale hail probability and related hail risk across Europe.

Numerous authors haved used hail signals derived from conventional weather radars for the identification and analysis of hail because of their high temporal and spatial resolutions. For example, Nisi et al. (2016) and Nisi et al. (2018) established a hail climatology for Switzerland from 2002 to 2014 based on both Probability of Hail (POH) and Maximum Expected Severe Hail Size (MESHS) estimated from volumetric (3D) radar data. Puskeiler et al. (2016) used 3D radar reflectivity together with modelled melting layer, lightning data and the cell-tracking algorithm TRACE3D (Handwerker, 2002) to reconstruct hailstreaks and, from that, to estimate the hail frequency across Germany between 2005 and 2011. Combining 3D radar reflectivity and insurance loss data for buildings, Kunz and Puskeiler (2010) found the highest hail frequency in Southwest Germany to be located downstream of the Black Forest mountains. This hot spot was also confirmed by Kunz and Kugel (2015) using five different hail criteria based on 2D and 3D radar reflectivities and different heights (melting layer, echo top). Lukach et al. (2017) computed a hail frequency map for Belgium from 2003 to 2012 using 3D radar data. Outside of Europe, Cintineo et al. (2012) produced a high-resolution hail frequency map for the USA from 2007 to 2010 using MESH (Maximum Expected Size of Hail) product. The authors found a high hail frequency during March to September (with June as a maximum) in the Great Plains. More precisely, the highest hail frequency is mainly centered over the southern part of the Great Plains from March until May,

while from July to September, hail is more frequent in the central and northern plains. The MESH product was also used in the studies conducted by Warren et al. (2020) in Australia where the authors used daily grids of merged radar data including MESH at a 1 km resolution from 2009 until 2017. A pronounced peak of hail appeared during (souther-hemisphere) summertime, in December on the coastal slopes of the Great Dividing Range. In Czechia, Skripniková and Řezáčová (2014) used the Waldvogel criterion on single-polarisation radar data to retrieve hail signals for the period 2002 until 2011. The authors found that hail occurred mostly during May, June and July during the afternoon throughout the country. Despite the use of improved radar-based techniques, most of the studies cited above were restricted to smaller regions or a single country. While some authors have estimated hail frequency from other sources such as Overshooting Tops ,"a domelike protusion above the cumulonimubs anvil, representing the intrusion of an updraft through its equilibirum level" according to the American Meteorological Society (Glickman and Walter, 2000), in satellite imagery (Punge et al., 2014), model data (Mohr et al., 2015b, a; Rädler et al., 2018) or a combination thereof (Punge et al., 2017), the link between the observed quantities and hail occurrence at the surface is less reliable than using radar measurements. Numerical models, such as weather forecast or regional climate models (RCM), on the other hand, are not able to reliably reproduce hail due to a high degree of uncertainty in the initial conditions, a lack of knowledge in cloud microphysics, and the high computer costs when running a two- or three-moments microphysics scheme.

The objective of our study is to analyze the spatiotemporal variability of hail signals over a 10-year period (2005 to 2014) covering the four European countries of France, Germany, Belgium and Luxembourg. Hail signals were estimated from 2D radar reflectivity available for each country, which permits a homogeneous hail analysis. The results help to identify regions frequently affected by hail and allow us to relate hail frequency to topographic features such as terrain height or the proximity to the sea. A thorough study of hail events gives further insights into the relation between orography and deep moist convection. Improved understanding of these mechanisms and processes is crucial to improve the nowcasting and forecasting skill of hail storms. Finally, as hail constitutes a considerable risk for the insurance industry, improved knowledge about hail frequency and hailstorm characteristics will help to better understand the related risks.

The paper is structured as follows: Section 2 gives an overview of the remote-sensing and reanalysis datasets used for this study. Section 3, describes the combination of radar data with lightning data and the application of the tracking algorithm. The remote-sensing output is then used to generate European composites at 5-minute time steps. Section 4 assesses the hail variability between 2005 to 2014 in relation to the distance to the sea and the presence of orography near hailstorms. This section also presents results on seasonal and diurnal variations in hail frequency and provides some characteristics of the hail cells. Concluding remarks follow in Section 5.

## 2    Datasets

### 2.1    Remote-sensing data

In this paper, we present a hail climatology retrieved from radar reflectivity datasets available from the first phase of the project HAMLET (Hail Model for Europe by Tokio Millennium) that lasted from 2013 until mid-2017. 2D radar reflectivity for the summer half-years (April to September) from 2005 to 2014 for Germany, France, Belgium, and Luxembourg are considered

(Figure 1). The French national radar composites were available until 2014 only, due to the installation of five new X-band radars in the Alpine region in 2014 (See Section 2.1.1) that requested some computation adjustment into the national radar composite. The French national radar composites from 2014 up to nowadays including the X-band radars in the Alpine region installed in 2014 were available only later. The radar products used in this study are composites of the Maximum Constant Altitude Plan Position Indicator (MaxCAPPI), where the composite is a merger of the data from all local radar stations in a single image at time steps of five minutes. 2D radar data are used here because of the large domain and their long-term availability.

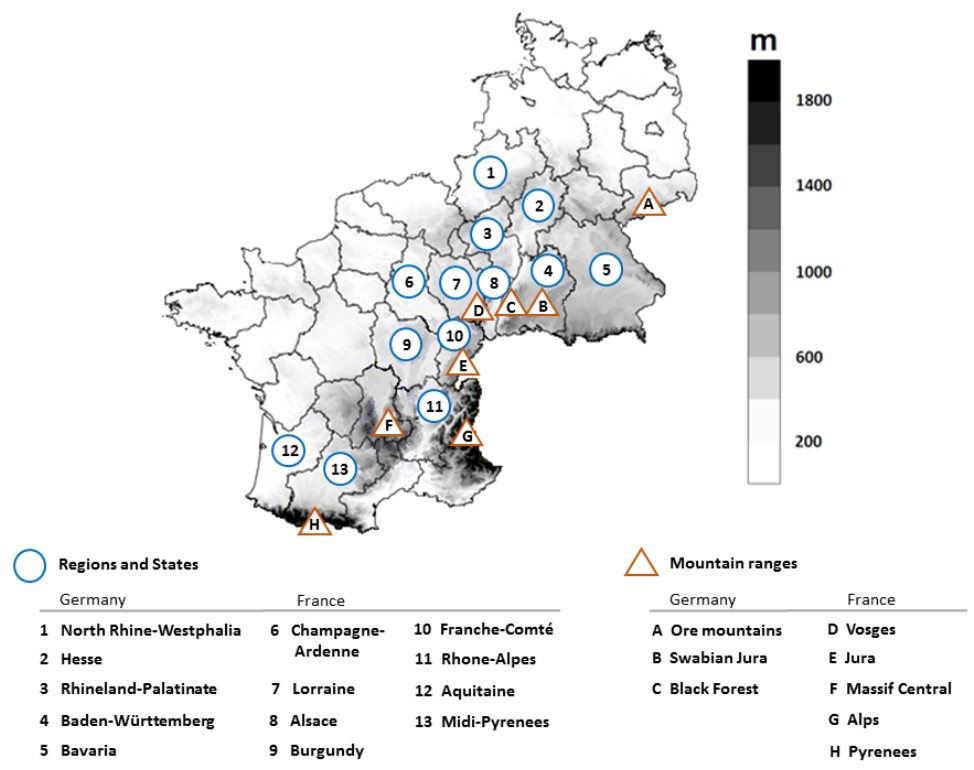

**Figure 1.** European regions and mountain ranges mentioned in this study.

### 2.1.1 French radar data

The French radar network operated by Météo-France and the derived radar products evolved constantly through times, mainly via national projects. A brief overview of the French radar network is given hereafter. The project Panthère launched in 2002 (Parent du Châtelet et al., 2005) permitted to add 6 new radars to the previous 19 radars constituting the French radar network in 2001. During this operation, some of the 19 radars were replaced by dual polarization radars (Tabary et al., 2006, Bousquet et al., 2008). In 2005, 24 radars were in operation including 19 C- band radars and five S- band radars (both with a radius of up to 120 km). Two years later, in 2007, the radar stations of Toulouse in southwestern France and Trappes (near Paris)

were renewed in the course of the same project Panthère (Tabary, 2007) but this replacement did not affect the radar national composite. During the period from 2007 to 2011, the radars of Plabennec located in northwestern France, Abbeville in northern France, Nîmes in southern France and Grèzes in the southern part of central France were replaced as well with dual polarization

radars. In 2014, five X-band radars with an average coverage radius of 50 km were added to the French national radar composite (Figure 2) via a project named RHyTMME (Risques Hydrométéorologiques en Territoires de Montagnes et Mediterranéens) described in the study of Beck and Bousquet (2013) with the objective to improve the hydrological risks management in the southern Alps region (Champeaux et al., 2011). As the data from the X-band radars were only recently implemented into the French national composite (Yu et al., 2018), only S- and C-band radars were considered in this study.

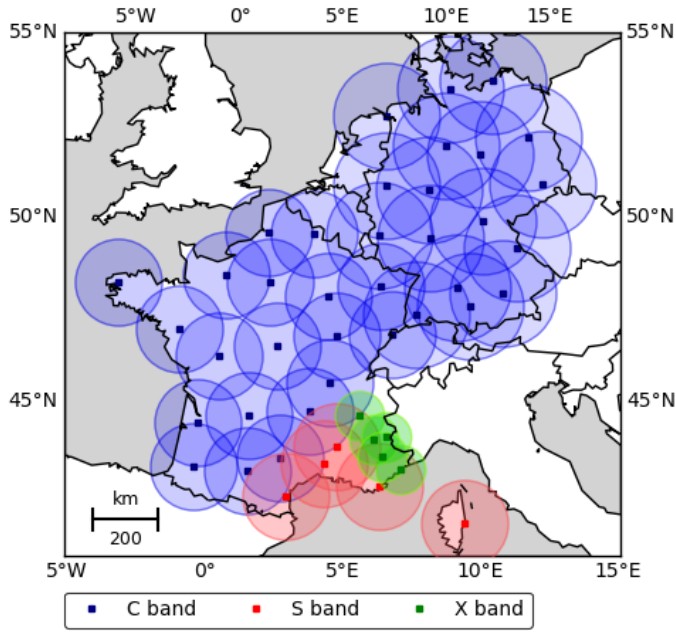

**Figure 2.** Locations (squares) and coverage of the radar stations (circles) in 2014 used in this study. See text for further explanations.

Concerning the scanning strategy, four to six scans are performed every 15 minutes at elevation angles ranging from 0.4° up to 15° (Figueras i Ventura and Tabary, 2013). Only the lower elevation angles below 2.7° are scanned every 5 minutes. The radar stations of Avesnois (located in northern France) and Réhicourt-la-petite in Lorraine (mentioned as region number 7 in Figure 1) cover a large part of eastern France, and permit to integrate Luxembourg completely, as well as a significant part of Belgium, into the French national composite. The spatial resolution of the composite is $1 \times 1$ km$^2$ with a size of 1536

$\times$ 1536 grid points for each image referred to a plane Cartesian coordinate system (Tabary et al., 2006). Radar data from all stations are pre-processed via an algorithm (Tabary, 2007) named Castor2 (Figueras i Ventura et al., 2012), which corrects for several errors, such as antenna positioning errors, and quantifies horizontal reflectivity $Z_h$ (and differential reflectivity $Z_{dr}$ or correlation coefficient $\rho_{hv}$ for dual-polarized radars) in polar coordinates (Tabary, 2007). During the pre-processing

stage, each radar pixel receives a weighted quality index (QI) ranging from 0 to 1 (Tabary, 2007), updated throughout the whole pre-processing chain. The first pre-processing step is to eliminate ground clutter, i.e., fixed echoes at the surface, using Doppler velocity (Tabary et al., 2013). Then an orographic mask is applied at each elevation angle in order to assess the beam occultation rate. After that, an "anthropogenic" mask, including buildings, trees, or other fixed objects in the vicinity of the radar is computed with the help of long-term accumulated radar products. These masks allow to remove radar pixels with artificially high reflectivity at each elevation angle. Beam widening, i.e., the increase of the radar volume with distance to the radar, is taken into account (Tabary et al., 2013) using vertical reflectivity profiles. Attenuation by oxygen is corrected depending on the wavelength, the elevation, and the distance to the radar (Doviak and Zrnić, 2006). For example, a correction of 1.79 dBZ at 100 km away from the radar site is applied to C- band radars for an elevation angle of 0.4° (Tabary et al., 2013). One of the last pre-processing step is the correction of the bright band with the help of vertical reflectivity profiles. After performing all the steps described above, individual plan position indicators (PPIs) are combined to 2D composites produced every 5 minutes available for each radar station to estimate the intensity of rainfall by converting the reflectivity data into rain rate using the $Z - R$ relation according to a Marshall-Palmer distribution (Tabary et al., 2013). During this step each pixel is assigned a QI ranging from $100\%$ (excellent) to $0\%$ (poor) resulting from the previous weighted QIs (Champeaux et al., 2011). A pixel with a QI of less than $80\%$ is automatically removed from the rainfall product. The final step of the pre-processing is the Quantitative Precipitation Estimation (QPE) calibration with rain gauge data by using the large rainfall network operated by Météo-France including approximately 900 automatic stations (Champeaux et al., 2011). A weighted calibration of the QPE is performed with real-time, hourly rain gauge data. After performing the pre-processing chain, all individual radar products are combined into a national mosaic (Augros et al., 2013). For areas with overlapping radar coverage, weighted reflectivity data are computed depending on the distance to the nearest radar (Tabary et al., 2013). At the boarders of France, radar data from other national weather services are integrated into the French national mosaic.

Reflectivity values from the French radar mosaic used in this study were coded in a table and stored in GeoTIFF format, i.e., georeferenced TIFF images. The resolution is $2 \times 2$ km$^2$ from 2004 until mid-June 2009, and a finer resolution of $1 \times 1$ km$^2$ is available from mid-June 2009 to 2014. For data homogenization, each of the $2 \times 2$ km$^2$ composite was interpolated linearly from 2004 to June 2009 to the finer grid of $1 \times 1$ km$^2$.

### 2.1.2 German radar data

The German radar composites for the period from 2005 to 2014 are provided by the German Weather Service (DWD), which operated a network of 17 C- band radar systems in 2014. During the investigated period, a new radar in Memmingen, southern Germany, was added to the network in 2012 (Puskeiler, 2013). As the horizontal range detection for each radar is 180 km and a maximum distance of 200 km separates the radar stations, an extensive overlap of the detection areas permits almost a complete coverage of the German territory. Only some peripheral regions remain as gaps in the composite, for example, in the far North near the Danish border or in southeastern Bavaria (Figure 2). In the complex terrain of southern Germany, weather radars are preferably located on hills and mountains to minimize beam shielding by orography. Concerning the scanning strategy, the lowest elevation angles between 0.5° and 1.8° (Bartels et al., 2004) at that time were scanned every 5 minutes, whereas a

complete volume scan took 15 minutes. The maximum reflectivity values of the lowest elevations are used for the national 2D reflectivity composite. The steps of the pre-processing are similar to those of Météo-France, and include among others: An

160 elimination of clutter pixels using a clutter filter, orographic shading correction using an elevation model, transformation of reflectivity Z to rain rate R. After the pre-processing, the local radar data are merged into the German national composite. For areas with overlapping radar coverage the maximum reflectivity value from all radar scans is used in the composites, while for the neighboring regions of foreign countries, a weighted adjustment is performed between radar products from other national weather services and the German rain-gauge dataset (Kreklow et al., 2020). The quality of German radar data has improved

over the last decade with continuous algorithm corrections and adjustments (Kreklow et al., 2020) used for RADOLAN (Radar-Online-Aneichung, which means Radar-Online Adjustment) and can be assessed by a quality flag provided for each pixel on the reflectivity product. The spatio-temporal resolution as well as the time period available for the German radar data is the same as the French, namely $1 \times 1$ km$^2$ with a 5-minute time step for the composite and available from 2005 until 2014, so that both data sets can be merged. The encryption of each scan of the DWD network entails raw terrain-following near ground

reflectivity (CAPPI) values (named RX product) in so-called RVP6 units. The advantages of this data are the high temporal and spatial resolution which enables us to properly identify footprints of SCS. RX data are projected on a Cartesian grid so that each grid box is equidistant at 1.0 km. In the end, the German radar composite has a size of $900 \times 900$ km$^2$ covering the whole of Germany.

### 2.1.3 Uniform Pan-European Grid

It is important to note some limitations in both the German and French national composites. Long-term QPE maps for the French national composite reveal some regions with low data accuracy. This is mainly the case for the central and eastern part of the Pyrenees mountains and the entire Alpine region (Tabary et al., 2013). In the other parts of France, the QI is mostly higher than 90% with especially high QI close to the radar site (Tabary et al., 2013). Radar data failure, for example during radar calibration, or radar replacement, were estimated by Puskeiler et al. (2016) to be approximately $4.5 \pm 3.9\%$

on average (mean $\pm$ standard deviation for the German national composite). Furthermore, the combination of the German and French national composites, each calibrated and pre-processed in different ways, may lead to inhomogeinities in relative hail frequency in some regions. Based on manual investigation of several cases with severe hailstorms in the border region between Germany and France, it was found that the signal of the French mosaic is between 0.5 and 1 dBZ lower compared to that obtained from the DWD composite (Schmidberger 2020, personal communication). This uncertainty is acceptable when

projecting the two national composites onto a uniform Pan-European Grid. Radar reflectivity data and thus radar-derived hail signals were projected on the same uniform European grid (not shown) with a resolution identical to that of the national radar network ($1 \times 1$ km$^2$). We used the geographic coordinate system WGS84 for the Pan-European Grid and a Lambert Conformal Conic Projection, as recommended by Gregg and Tannehill (1937) and Varga (1990). In the center at about 47°N and 6°E, the meridional grid spacing is equal to the zonal direction to minimize the grid distortion.

### 2.1.4 Lightning data

To remove artificial clutter still present in the data, we additionally implemented a filter based on lightning data, which was already used by Puskeiler et al. (2016). Here we used only cloud-to-ground (CG) lightning (strokes) from the low-frequency lightning detection system BLIDS (BLitz InformationsDienst Siemens), which is part of the EUCLID (EUropean Cooperation for LIghtning Detection) network. The detection efficiency of the system is 96% for strokes with a peak current of at least 2 kA (Schulz et al., 2016). Because the sensors and the algorithm implemented until 2015 had a significantly lower detection efficiency of intra-cloud and cloud-to-cloud lightning according to Pohjola and Mäkelä (2013), these types of lightning were not considered.

### 2.1.5 ERA5 reanalysis

To assess the mean wind flow during hail days, we used the ERA5 global reanalysis (Hersbach et al., 2020). ERA5 is a new global atmospheric reanalysis recently released by the ECMWF and aims to replace ERA-Interim reanalysis (Dee et al., 2011) whose data extend from 1979 to 2019. For the moment, ERA5 is available from 1979 onwards and will be soon extended to 1950. The ERA5 4D-Var analysis dataset is assimilated by the Integrated Forecasting System (IFS) and is available on a horizontal resolution of 31km on 137 vertical levels every hour.

## 3 Methods

### 3.1 Correction of erroneous signals

Concerning the homogenization of the French and German national composites, several corrections had already been performed by both national meteorological services. Radar reflectivity data still contains noise and systematic errors that have to be eliminated using various approaches. Errors mostly concern individual radar pixels with significantly higher reflectivity values (e.g., more than 70 dBZ) compared to the surroundings. To avoid this problem, reflectivities below 35 dBZ or above 70 dBZ were set as missing values. Following Puskeiler et al. (2016), an additional verification and correction filter was applied for reflectivity values of $Z > 45$ dBZ with a difference of $\Delta Z > 5$ dBZ to the adjacent pixels. The affected pixel is set to the mean value of its 8 surrounding pixels and this filter was applied to all consecutive radar scans:

$$Z(x,y) = \frac{1}{8}\left( \sum_{i=-1}^{1} \sum_{j=-1}^{1} Z(x+i, y+j) - Z(x,y) \right) \tag{1}$$

In addition, a high reflectivity value at least twice as higher compared to the 8 neighboring values that cannot be observed in the scan before or afterward is considered an artifact and removed.

## 3.2 Lightning filter

Despite the radar tracking routine (see next paragraph) has included a clutter filter, several erroneous signals are still present in
the radar data. For example, isolated non-meteorological targets such as electronic signals or reflectivities from wind turbines
can emerge in radar scans (Steiner and Smith, 2002).

Since hail occurs only in association with thunderstorms (Baughman and Fuquay, 1970; Changnon, 1999; Wapler, 2017),
lightning is expected near high reflectivity cores. In addition to the gradient filter described above, we used lightning detections
to further remove artificial clutter. If high reflectivity values occur during a 24-hour period without lightning, the values at the
affected grid points are set to zero. A maximum distance of 10 km was chosen between a lightning discharge location and the
pixels with high reflectivity. Distances of 5, 15, and 20 km were also tested; a distance of 5 km led to the disruption of several
hail tracks due to gaps in reflectivity values; the other two thresholds affected the results only marginally. An example of the
lightning filter application during a hailstorm can be found in Fluck (2018) for the 27 July 2013 at 15:30 UTC.

## 3.3 The convective cell tracking algorithm CCTA2D

The object-based Convective Cell Tracking Algorithm (CCTA2D) permits the reconstruction of tracks of individual convective
cells using 2D radar data. The algorithm is based on the tracking algorithm TRACE3D (Handwerker, 2002), originally devel-
oped and optimized for 3D radar reflectivity from a single radar in spherical coordinates. TRACE3D was further extended to
radar reflectivity data in Cartesian coordinates such as those provided by the DWD radar network (Puskeiler et al., 2016). A
second version was adapted to 2D terrain-following near ground reflectivity (CAPPI) using both the RX product from DWD
and the French mosaic including France, Belgium, and Luxembourg (Fluck, 2018).

The first step of CCTA2D is to identify, regions of intense precipitation (ROIP) delimited by $Z \geq 35$ dBZ and to determine
the corresponding maximum reflectivity values ($\text{Max}_{ROIP}$). In order to distinguish individual reflectivity cores (RCs) within
each ROIP, a value of $\Delta Z = 10$ dBZ is subtracted from $\text{Max}_{ROIP}$ to set the minimum threshold necessary to delimit a single
RC ($\text{Min}_{thresRC}$). Thus, the value of $\text{Min}_{thresRC}$ remains the same for all identified RCs inside a ROIP. If $\text{Min}_{thresRC}$ is less
than 55 dBZ, the RC is rejected and not tracked by CCTA2D. Two additional conditions are required for a RC to be classified as
a potential convective cell and to be tracked by CCTA2D: A minimal surface area of 5 km$^2$ is needed to define a RC with at least
3 radar bins of $Z \geq 55$ dBZ. The thresholds detailed above to identify potential convective cells in CCTA2D are summarized
in Table 1. The 55 dBZ threshold is referred to as the hail criterion according to Mason (1971), and was successfully used
in several studies (e.g., Hohl, 2001,Hohl et al., 2002,Kunz and Kugel, 2015). Schuster et al. (2005), for example, found the
55 BZ to be a good indicator for damaging hail on the ground in Eastern Australia. Puskeiler et al. (2016) estimated a slightly
higher threshold of 56 dBZ best separating between days with and without insured damage to buildings, but confirmed the
55 dBZ to estimate at best insured damage to crops. Categorical verification using insurance loss data over a 7-year period
in southwest Germany for this threshold yields a Heidke Skill Score HSS of 0.6, a quite high value confirming the detection
skill (it should be noted that this value increases to HSS = 0.71 when using an adjusted version of the Waldvogel et al. (1979)
criterion requiring 3D radar data). In the same study, Puskeiler et al. (2016) found that the Probability of Detection (POD) has

**Table 1.** Thresholds required in radar composites to identify potential convective cells with the CCTA2D algorithm.

| Description | Value | Units |
|---|---|---|
| Minimum reflectivity of a ROIP | 35 | dBZ |
| Minimum reflectivity of a RC | 55 | dBZ |
| Reflectivity to subtract from ROIP Maximum | 10 | dB |
| Minimum RC area | 5 | km$^2$ |
| Minimum number of elements inside a RC | 3 | - |

reached 0.65 and the False Alarm Rate (FAR) has attained 0.4, indicating that 35% of the observed hail events are missing while 40% of those predicted events are false alarms.

The second step of CCTA2D is the temporal and spatial tracking of all detected convective cells. The algorithm assigns the RC of the previous radar composite to the actual composite according to the estimated propagation velocity and the position of the RC. Prerequisite of the tracking is that similarities between different RCs, for example, in intensity and size, from one time step to the next must exist within a certain search radius for accurate RC assignment and tracking. The search radius is given by the estimated distance of an initial RC displaced during a time step of 5 minutes multiplied by a velocity factor of 0.6.

Special attention is given to cell splitting and merging. Cell splitting is a prominent feature of supercells associated with vertical pressure disturbances. In most cases, the left-moving cell weakens very quickly, whereas the right-moving supercells further exist. In order to track both cells after they have been splitted, a splitting (merging) option in the tracking algorithm is necessary. Furthermore, without splitting or merging options, the physical characteristics of SCS tracks such as their length or their angle of orientation could be incorrectly computed by CCTA2D. To detect cell splitting, the initial RC is first spatially displaced to the position of the following RC (e.g., the successor), and their respective areas are compared (Handwerker, 2002). If the area of the successor differs significantly from the initial RC, it is assigned a potential splitting. In the case of merging, the opposite calculation from cell splitting is performed. The maximum distance between two RC centers that could merge is set to 10 km. Initial and successor areas are then compared, and the successor is placed at the weighted center of all initially detected cores. Merging occurs when the successor area is larger than the initial RC. To avoid reflectivity core crossings or overlapping, each RC is enumerated and recorded separately.

After the construction of entire cell tracks, the composite of maximum reflectivity on a given day does not provide a smooth result, but a rather scattered product. This effect is most pronounced when the cells propagate further than their horizontal extent during a measuring interval. The faster the storms move, the more scattered is the maximum reflectivity projected on a 2D plane. This can substantially reduce reflectivity values between two scans, even though a high-intensity storm crossed the area. A gap of reflectivity values can also appear on radar scans in regions with overlapping radar data, especially on neighboring countries such as in the Rhine Valley. To consider this effect, an advection correction was performed following Puskeiler et al. (2016). A translation of the reflectivity cores is computed from one time step to the next considering the horizontal wind field estimated by CCTA2D along a track. The field of motion vectors as well as the track direction of the

convective cells are computed and projected on the German and French grid. Each point along a track includes a velocity shift-vector in north-to-south **dv** and west-to-east **du** directions. The so-called shift-vector **U** is denoted as:

$$\overrightarrow{\boldsymbol{U}}(x,y) = \begin{pmatrix} du(x,y) \\ dv(x,y) \end{pmatrix} \tag{2}$$


As the CCTA2D algorithm only determines the center of a track, an n-time parallel duplication of the track is required with a vector field from all locations of the thunderstorms. The parallel shift at the position $(b,c)$ is done with normalized vectors **t1** and **t2** of the cell motion direction with a spacing on each side of the track depending on the size of the RC and a maximum spacing of 20 km. The position $(b,c)$ obtained from the shift vector **U** is calculated as follows:

$$(b,c) = \left( x + |\frac{\delta \mathbf{t}_{1,2}}{\delta x}| \cdot n, y + |\frac{\delta \mathbf{t}_{1,2}}{\delta y}| \right) \tag{3}$$

Equation 3 thus provides a displacement field of the entire cell complex.

In our analysis, long-living SCS tracks were compared with hail reports archived by the European Severe Weather Database (ESWD) operated by the European Severe Storms Laboratory (Dotzek et al., 2009) along the reconstructed storm tracks to
assess the reliability of CCTA2D. In fact, in the recent paper of (Kunz et al., 2020), the authors separate all SCS events used in this study from the hailstorm events by assessing the presence of hail using ESWD reports in the vicinity of SCS tracks. Out of 26 012 SCS events in total, only 985 events could be confirmed by hail reports. The main reason of this significant reduction of confirmed hail events is that ESWD reports are by far not complete. Whereas most of the reports prevail for Germany, only a low number is available for France, Belgium, and Luxembourg.

## 4 Results

### 4.1 Spatial distribution of hail

Figure 3 presents the hail probability map for the radar domain (cf. Figure 2) in terms of annual average hail days during the period from 2005 to 2014 with a resolution of $1 \times 1$ km$^2$ based on 2D radar reflectivity. A day is considered as hail day when the threshold of 55 dBZ is exceeded in the daily maximum reflectivity composite after (i) data correction, (ii) filtering with
lightning data, and (iii) tracking with the object oriented algorithm CCTA2D as described in the previous section. If the hail criterion of $Z \geq 55$ dBZ is fulfilled on a specific day at a single grid point, this grid point is set to 1, otherwise it is counted as zero. The total of all days with hail over the entire 10-year period divided by the number of years yields the radar-based "hail climatology". In accordance with other hail frequency analyses (e.g., Puskeiler, 2013, Nisi et al., 2016, Junghänel et al., 2016, Nisi et al., 2018), the term climatology is used here, even though our investigation refers to a period far below a climatological
time scale of $\geq$30 years. Note that this climatology represents the spatial distribution of convective cells with high reflectivity, but not directly of hail as the 55 dBZ threshold does not guarantee hail on the ground. Similarly, the absence of high reflectivity

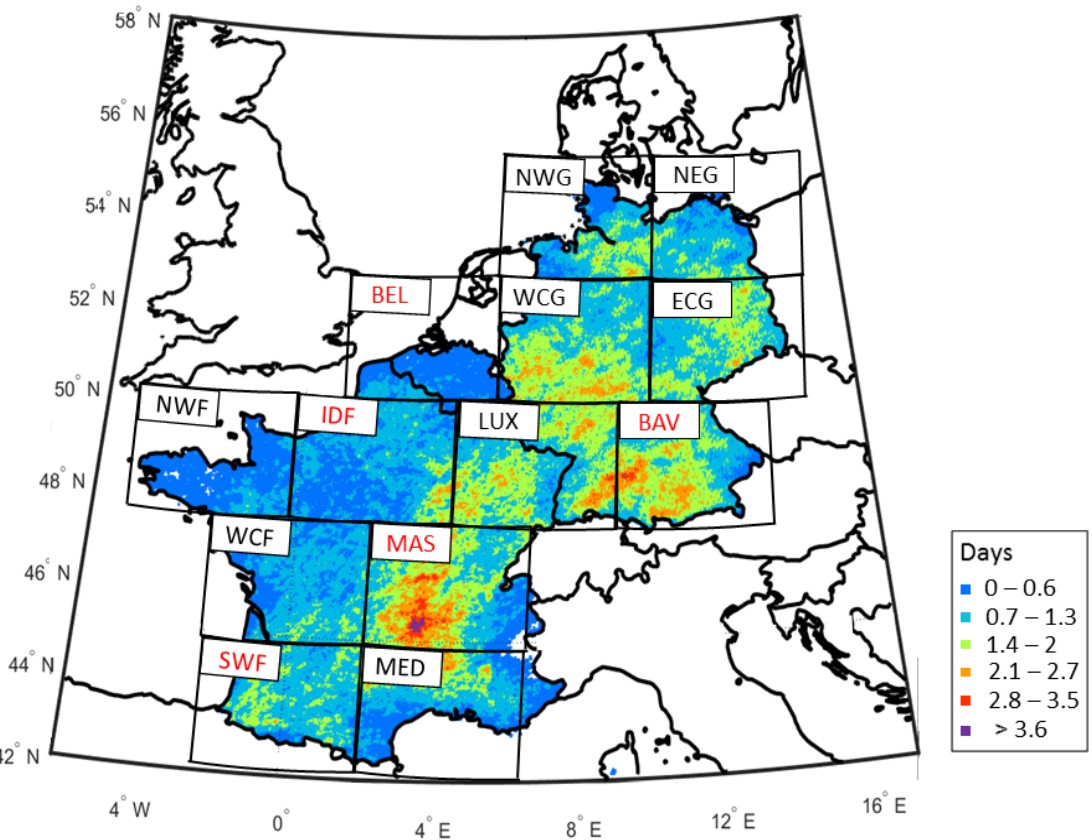

**Figure 3.** Annual radar-derived hail frequencies for $1 \times 1$ km$^2$ grid points in France, Germany, Belgium, and Luxembourg between 2005 and 2014. Squares represent boundaries of subdomains further investigated in this study (See subsection 4.3 for further details). The subdomains were named as follow: NWG (Northwest Germany), NEG (Northeast Germany), BEL (Belgium), WCG (West-Central Germany), ECG (East-Central Germany), NWF (Northwest France), IDF (Île-de-France), LUX (Luxembourg), BAV (Bavaria), WCF (West-Central France), MAS (Massif Central), SWF (Southwest France), MED (Mediterranean). Special emphasis is given for subdomains written in red.

does not ensure that hail did not occur (See Section 3.3). The term hail days used in the following parts of this study refers to the exceedance of reflectivity, but not to confirmed hail observations.

As can be seen in Figure 3, the spatial variability of hail days is very large, but some patterns with distinct minima or maxima can be identified. The lowest number of hail days is around the coasts, both the Atlantic and the Mediterranean with low frequency over northwestern France, Belgium and North Germany. Conversely, the highest number of hail days is located towards the East of France, with maxima present in contiguous area such as in central France (area MAS) or in southwestern Germany. Besides the recognizable structures of maxima and minima, some very patchy patterns appear for example in area ECG or LUX. As a result, an increasing gradient in the number of hail days can be recognized from northwestern France


towards central France; and a predominant gradient pointing from North towards South Germany can also be mentioned.

A close-up investigation of the hail hot spots is detailed hereafter. Figure 4 represents the location of the mean hail days overlaid with the 10 m mean wind during hail days (in terms of speed and direction) from 2005 to 2014 on the high-resolution global relief ETOPO1 having a 1 arc-minute resolution (Amante and Eakins, 2009). The 10 m mean wind was computed using

the hourly and 31km horizontal resolution ERA5 global analysis (Hersbach et al., 2020).

The area with the highest average number of hail days during the 10-year investigation period is situated on the leeward side of the highest mountains of the Massif Central averaging up to 4.6 hail days. This maximum extends over a few kilometers over the central part of the Massif Central (named Livradois region), composed by a plain and middle-range mountain measuring up to 1200 m high (named Livradois mountains). During days with hail, a strong flow is coming from the Mediterranean Sea

with a northern direction, thus impinging the southern and southeastern mountains of the Massif Central at a sharp angle. Another general westerly flow reaches the western part of the Massif Central. Interestingly, it seems that not only the location of the Massif Central is responsible for the increased number of hail days downstream, but also the flow convergence where the westerly flow meets with the flow coming from the Mediterranean. One may speculate that even without the Massif Central hail days might be increased in that area of low-level flow convergence. The large valleys on the western side of the Massif

Central, oriented from southwest to northeast, facilitate the passage of the flow coming from the southwest into the Livradois region. This region with an average number of 3.2 hail days per year is located in an area where the wind vectors converge both in the direction and velocity. In order to better understand the flow characteristics over the Massif Central shown in Figure 4, we calculated the Froude number on radar-derived hail days from ERA5 (Queney, 1948; Smith, 1979) for a region covering the Massif Central entirely and ranging from 44.0° to 46.5°N and from 2.0 to 4.7°E. The Froude number is calculated as follows:

$$Fr = \frac{U}{NH} \tag{4}$$

where $U$ represents the wind speed perpendicular to the mountain and was computed by applying a density weighted integration over the lowest 2000 m. $H$ is a characteristic mountain height set to 1300 m for the Massif Central region and $N$ is the Brunt–Väisälä frequency. According to Huschke (1959), the squared Brunt–Väisälä frequency $N^2$ is defined as :

$$N^2 = \frac{g}{\theta_{va}} \frac{\partial \theta_{va}}{\partial z} \tag{5}$$

Where g is the gravitational acceleration equal to 9.8 m.s$^{-1}$, $\theta_{va}$ is the ambient virtual potential temperature, and $\frac{\partial \theta_{va}}{\partial z}$ represents the vertical gradient of the virtual potential temperature. In our analysis, we considered the root-mean-square of the Brunt–Väisälä frequency $N$ in order to exclude imaginary values.

The mean Froude number on hail days over the Massif Central from 2005 to 2014 is $Fr = 0.39 \pm 0.3$. According to Smith (1979) and Smolarkiewicz and Rotunno (1989), a Froude number below 1 suggest a flow that goes around the mountain rather

than directly over it. Thus, it can be assumed that the flow around the Massif Central is deviated by the mountains peaks leading to convergence downstream at low levels on the leeward side of the Massif Central, where the hail hot spot is located.

Several authors have found an increased hail frequency rather downstream than upstream or directly above the mountains. This is for example the case in the Pyrenean region (Vinet, 2001, Berthet et al., 2011, Hermida et al., 2013, Merino et al., 2013),

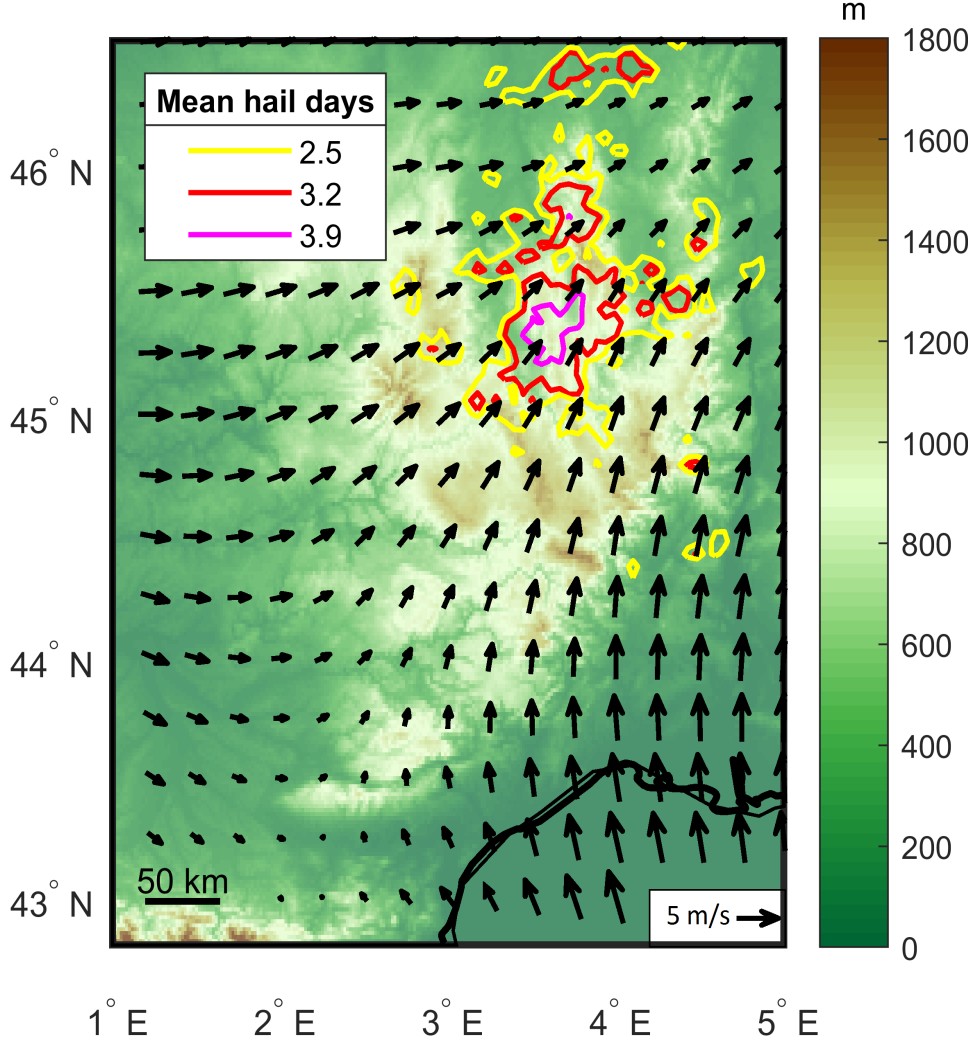

**Figure 4.** Contours of the average number of hail days from 2005 to 2014 overlaid with the orography and the 10m mean wind flow during hail days.

near the Black Forest in Germany (Kunz and Puskeiler, 2010, Puskeiler et al., 2016), or in the vicinity of the Alps (Eccel et al.,
2012, Nisi et al., 2018). In a review about orographically induced convection, for example, Kirshbaum et al. (2018) found that leeside convergence produces ascent required for convective initiation on the leeward side of mountains by referring to the studies of Mass (1981) about lee-side convergence near the Olympic mountains in the Washington State and of Barthlott et al. (2016) on convection initiation near Corsica mountains during HyMeX (Hydrological cycle in the Mediterranean eXperiment). Low-level flow convergence could explained the high frequency of hail on the leeward side of the Massif Central; however, this

is still a hypothesis that require additional observations and numerical simulations in this region to assess convection initiation.

The northeastern part of France, including the regions of Burgundy (referred to as region 9 in Figure 1), Champagne-Ardenne (region 6), Alsace (region 8), Lorraine (region 7) and Franche-Comté (region 10), are affected with a maximum of 3.1 hail days in the central part of Burgundy, and more precisely on the eastern side of mountains ranging from approximately 300

to 900 m.. In Champagne-Ardenne, the number of hail days reaches up to 2.9 days over the mainly rolling terrain. In Lorraine, where the terrain is almost flat and the climate is more continental, on average 2.7 hail days were counted in its central part. A local and lower maximum of 2.4 hail days can be recognized in South Alsace, representing an area with complex terrain with mountains up to 1.424 m agl. Another hail hotspot in the northeastern part of France is found along the northern ridge of the Jura Mountains (mentioned as E in Figure 1) in Franche-Comté with 2.5 hail days. Note that the Jura mountains represent a

natural obstacle frequently triggering thunderstorms (Piper and Kunz, 2017) and hailstorms by orographical lifting (Langhans et al., 2013, Schemm et al., 2016, Nisi et al., 2018 among others).

The Rhone-Alpes (referred to as region 11 in Figure 1) is a region likewise frequently affected by hail. This region contains the large Rhone valley and is bordered by the Massif Central in the west and by the Alps to the east. The southwestern part as well as the southeastern edge of the region show a local hail maximum with up to 3.1 hail days. The existence of these two

hot spots may be explained by their proximity to the Mediterranean as during southerly flows, warm and moist air is advected preferably through the Rhone valley. The warm and moist air can then be lifted, for example, near a front system crossing the country from northwest to southeast, leading to forced convection. This effect was confirmed by Schemm et al. (2016), who analyzed the relation between radar-based hail streaks over Switzerland and adjacent regions and cold fronts identified in high-resolution model data (COSMO-2; Steppeler et al., 2003, Jenkner et al., 2010) during a 12-year period (2002 to 2013).

The authors found that around 45 % of the detected hail cell initiations located on the windward side of the pre-Alps (in the Rhone valley) are associated with cold-fronts coming from the West during the summer months (May to September).

Southwestern France, including both the Aquitaine and Midi-Pyrenees regions (referred to as regions 12 and 13 in Figure 1), is also frequently affected by hail with up to 2.6 hail days in the southwest range of the Massif Central. Aquitaine and Midi-Pyrenees regions are the two regions well known in the literature for their high hail probability (Vinet, 2001, Punge

et al., 2014). Hermida et al. (2015) used data from the ANELFA (Association Nationale d'Etude et de Lutte contre les Fléaux Atmosphériques) hailpad network and found that the Gers Department, located on the west side of the Midi-Pyrenees region, is the area the most affected by hail in southwestern France. The western and northern sides of the Pyrenees are also frequently affected by hail with up to 2.5 hail days.

According to Berthet et al. (2011), hail in that region frequently occurs when a low-pressure system is located over the

western part of Spain leading to southwesterly flow over France associated with the advection of warm and moist air over the Pyrenean mountain range.

In Germany, the main hail hotspot is located in the Southwest in the federal State of Baden-Württemberg (referred to as region 4 in Figure 1), specifically over the Swabian Jura (mentioned as B on Figure 1), south of the city of Stuttgart, with a maximum of 3.1 hail days. This hotspot has already been identified in previous studies of Puskeiler (2013) and Junghänel et al.

(2016). Using equation 4, we found a Froude number of $Fr = 0.51 \pm 0.6$ for 207 hail days during the period 2005 to 2014 for a region covering 48° to 49.2° N and 7.8° to 10.5° E, including the Swabian Jura as well as the Black Forest and considering a maximum elevation of 1400 m for the entire area. The Froude number found in our study in the southwestern part of Germany matches the results of Kunz and Puskeiler (2010) who estimated a Froude number for a region covering the Vosges mountains, the Rhine valley, the Black Forest and the Swabian Jura of $Fr = 0.32 \pm 0.15$ for 65 haildays (1997—2007) using radiosondes at 12 UTC. This low Froude number suggests a flow-around regime of the southern and northern mountains of Black Forest causing a zone of horizontal flow convergence downstream. This convergence zone coincides with the area of the highest number of hail days (Kunz and Puskeiler, 2010; Koebele, 2014). Moreover, Kunz and Puskeiler (2010) hypothesized that the southwesterly flow meets the Swabian Jura at a very sharp angle, which reduces the Froude number considerably and align the wind parallel to the mountain chain. This flow modification is assumed to be responsible for the flow convergence at low levels as was also found in model simulations using COSMO-DE by Koebele (2014).

Another local maximum of up to 2.6 hail days is found North of the Alps, on the western part of the State of Bavaria (referred to as region 5 in Figure 1). This result is in good agreement with the conclusion of Nisi et al. (2018) who found that this region can be affected by around 3 hail days (2002—2014).

In the northeast of Germany, a local maximum of up to 3.2 hail days is positioned over the Saxon Ore Mountains (referred to as A in Figure 1) South of the city of Dresden. Note, however, that this maximum is mainly caused by a high number of SCS in the year of 2007 (Piper, 2017), which was characterized by frequent upper air troughs over Western Europe and ridges over Central Europe (Wernli et al., 2010), leading to high-pressure gradients on the eastern part of Germany in combination a southeast-to-northeast flow regime from the Czech Republic (note that the almost same situation occurred in 2019).

The northwestern part of Germany, including the States of Hesse (region 2 in Figure 1) and Rhineland-Palatinate (region 3 in Figure 1), and the southern part of North Rhine-Westphalia (region 1 in Figure 1) are regions affected by approximately 1.4 hail days on average. The location of the hail patterns is partly caused by the local orography with a pronounced maximum in North Hesse that lies directly on the leeward side of the Westerwald low mountain range, which is characterized by rolling terrain.

## 4.2 Annual variability

The frequency of SCS shows a very large annual and multi-annual variability (e.g., Puskeiler et al., 2016; Nisi et al., 2018). This variability is partly related to large-scale flow mechanisms such as the presence of specific northern Hemisphere Teleconnection patterns representing the low-frequency mode of the climate system (e.g., North Atlantic Oscillation, NAO, or East Atlantic pattern, EA) or by variations in the sea surface temperature (Piper et al., 2019). Having reconstructed a very large event set of SCS/hailstorms as presented in the previous section, we are also interested how the frequency of these events vary across the whole domain and regionally.

Averaged over the entire investigation area, the annual number of hail days is between 72 (2010) and 103 (2006) with a mean of 86 (Figure 5). In 2006, large parts of Europe, including Germany, Belgium, Luxembourg, and northwest France, experienced higher temperatures than on average, especially during the end of June and July (NOAA, 2007), where two (moderate) heat

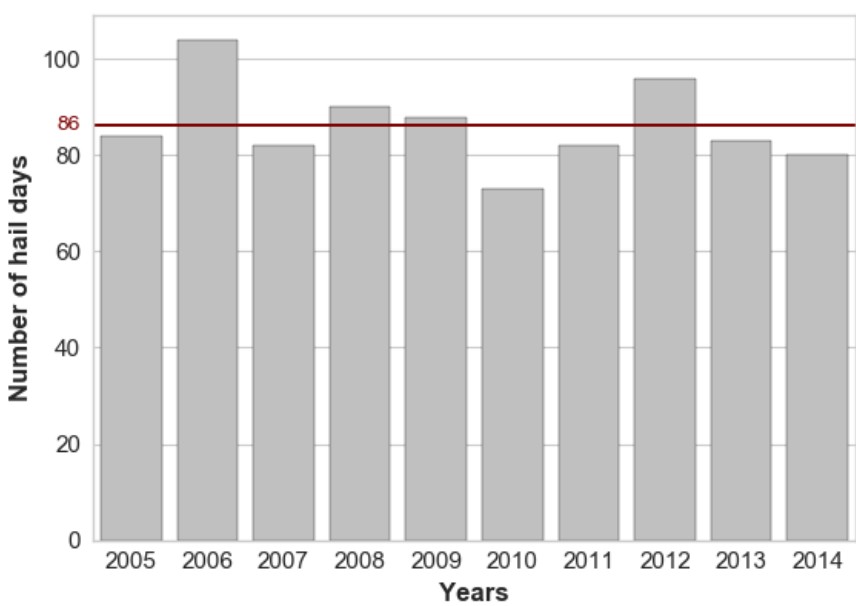

**Figure 5.** Yearly number of hail days from 2005 to 2014. The red line indicates the overall mean of hail days from 2005 to 2014.

waves occurred (Fouillet et al., 2008). As a result, the sea surface temperature over the Mediterranean showed a positive

anomaly (NOAA, 2007, Lenderink et al., 2009), leading to intense evaporation rates and, consequently, to an increase in the amount of water vapor in the atmosphere (Chaboureau et al., 1998).

The spatial distribution of hail days in 2006 (Figure 6) strongly resembles the climatology, with several maxima near hilly terrains and minima near the coastlines. Some hot spots can also be detected over the northwest part of France and in south-western Germany.

The year with the second highest number of hail days, 2012, was dominated by an episode with intense thunderstorm activity over southwestern Germany and France during the end of June and in July (DeutscheRück, 2013).

Even though the year of 2010 showing the lowest number of hail days was very warm on the global scale (NOAA, 2011), summer temperatures over large parts of Europe including Germany were below average. Furthermore, several persistent large-scale ridges occurred during the summer, which may have suppressed the formation of SCS (DeutscheRück, 2013). No clear

spatial pattern can be found in this year with only a few hailstorms in Central France. Almost no hailstorm could be detected in an arc spanning from northwestern France to northern Germany. There are further regions where hail was less present compared to the mean of 2010, including entire Belgium, Luxembourg, and the northwest of France, especially Normandy, Brittany and the coastlines. Interestingly, the year of 2013 likewise shows a negative anomaly in the number of hail days even if at the end of July two supercells associated with the low pressure Andreas caused an unprecedented economic loss of EUR 3.6 billion in

several densely populated areas of Germany (Kunz et al., 2018).

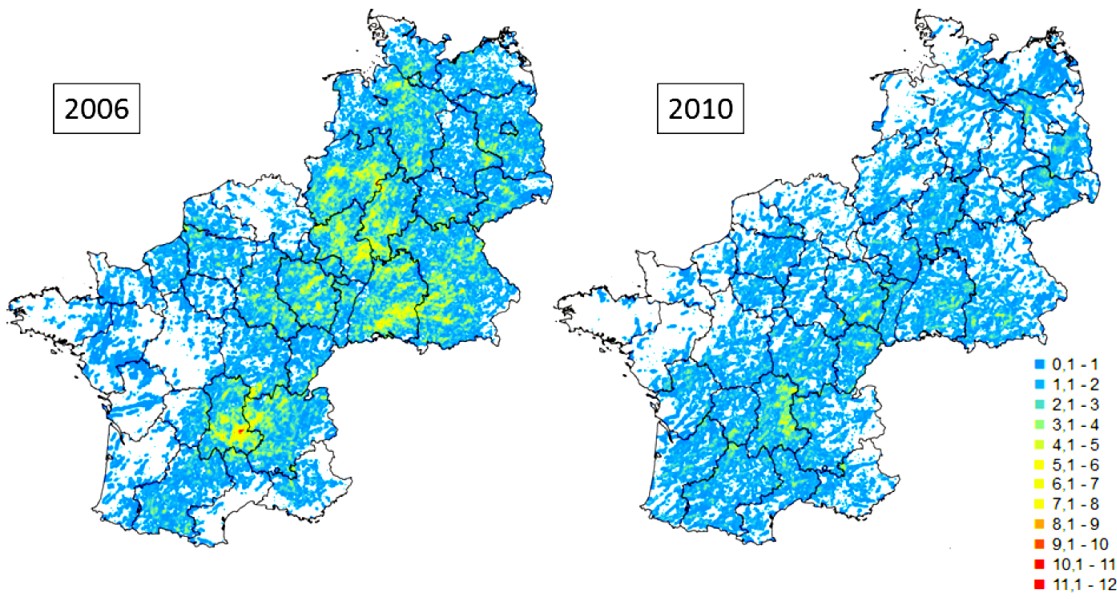

**Figure 6.** Number of radar-derived hail days exemplary shown for the years with the highest (2006, left) and lowest (2010, right) hail day frequency.

### 4.3 Seasonal and diurnal cycles of SCS

The large spatiotemporal variability of hail discussed in the previous sections leads us to the question of the seasonal and diurnal cycle of SCS at the regional level. For this purpose, the entire study area is divided into 13 subdomains of similar size (around 75,000 $\text{km}^2$) framed in Figure 3. We selected five subdomains with different terrain and climatological characteristics for further discussion: Belgium (BEL), Ile-de-France (northern France; IDF), Bavaria (southeastern Germany, BAV), the Massif Central (central France; MAS), and Southwest France (SWF). Subdomains BEL, SWF and IDF, have a climate strongly influenced by maritime air masses. Among them, subdomains BEL, and IDF represent flatlands, while subdomain SWF contain the high mountains of the Pyrenean. Subdomains BAV and MAS both have a rather continental climate, but have a different orography: While mainly hilly terrain characterizes subdomain BAV, subdomain MAS comprises the higher mountains of the Massif Central.

To quantify the number of hail days in each subdomain, a 10-day moving average of the number of hail days was calculated for the period 2005 to 2014 (Figure 7). Despite the large variability seen in the seasonal cycles of the subdomains considered, some similarities can be recognized. All time series of the different subdomains feature a clear annual cycle with a minimum of hail days in spring and autumn and a maximum during the summer. This characteristic cycle with a strong increase in the hail day frequency during April/May, a significant decrease around September, and with a maximum during the summer months was found by several other authors such as Dessens (1986) and Vinet (2002) for France, Belgium and Luxembourg, Gudd

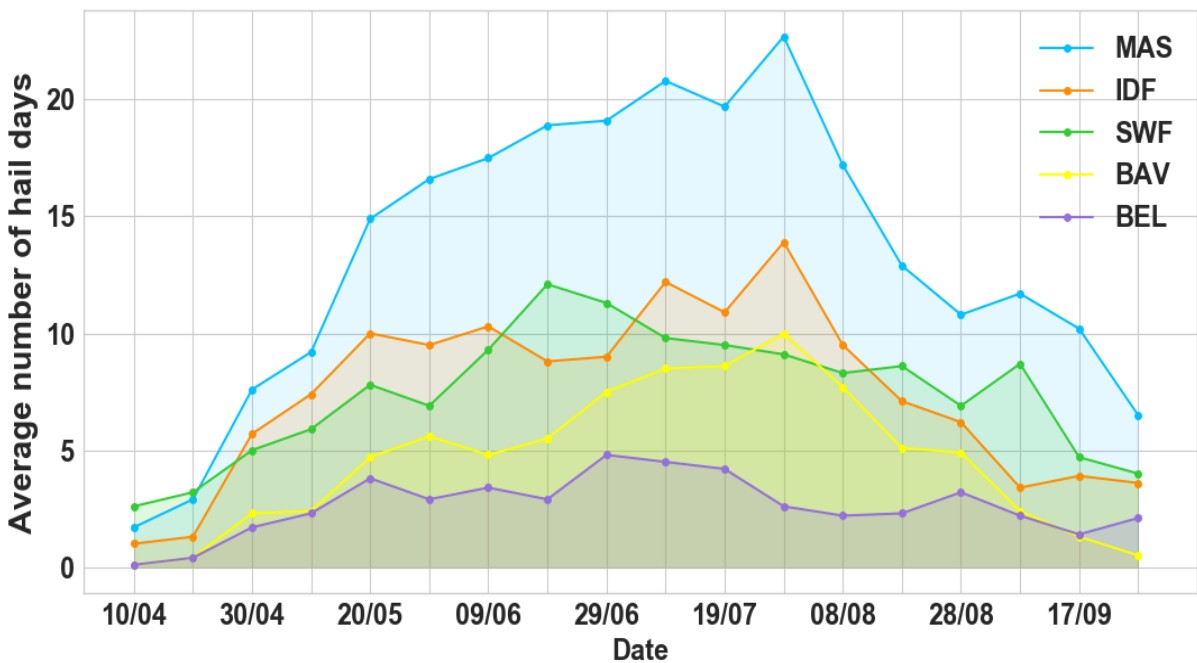

**Figure 7.** Time-series of the mean number of radar-derived hail days (10-days moving average) for the subdomains BEL, IDF, BAV, MAS and SWF shown in Figure 3.

(2003), Deepen (2006), Mohr and Kunz (2013) and Puskeiler et al. (2016) for Germany, and Nisi et al. (2014) and Nisi et al. (2018) for Switzerland and northern Italy.

The mountainous subdomain MAS shows the largest average number of hail days and has the most pronounced annual
cycle. Until the end of April, the average number of hail days for the 10-day running mean is below 10. During May and beginning of June, the number increases substantially from 9 around May 10 up to 17 days on June 9. The more pronounced diurnal temperature cycle for continental regions, associated with a higher lapse rate in combination with orographic lifting, may explain this increase (Berthet et al., 2011). After June 9, the average number of hail days increase steadily until to reach the overall maximum for the MAS region at the end of July with 23 days.
Subdomain IDF likewise show a high hail frequency during the summer with up to 14 days, mainly at the end of July. This subdomain is under the influence of the Atlantic Ocean (Cantat, 2004), leading to an increased frequency of troughs (Vinet, 2001, Berthet et al., 2013).

Within this subdomain, the number of hail days increases slightly until the peak with a first local maximum in the middle of May (around 10 hail days) and a second local maximum at the beginning of July with around 12 hail days. Spring hailstorms

may be associated with subtropical air masses coming from Spain, while summer storms preferably form ahead of cold fronts (Berthet et al., 2011). The number of hail days decreases sharply from the hail-peak season toward the end of September.

Subdomain SWF, located in the very southwest of France, has a very broad hail peak in the middle of June with 12 hail days during the 10-day moving average centered around June 19. Afterwards, the number slightly decreases, thus showing a right-skewed distribution. This maximum found in June differs from the analysis of Dessens et al. (2015), who found that May

is the most active month followed by July over the southwestern part of France and the Mediterranean area (situated along the Rhone valley). Also Fraile et al. (2003) and Hermida et al. (2013) found that May is the month with the highest hail kinetic energy in southwestern France. Reasons for this discrepancy can be due to a longer period analyzed by Dessens et al. (2015), while Hermida et al. (2013) and Fraile et al. (2003) focused on a time range starting from the 90s. Furthermore, the scattered network of hail pads is denser near the subdomains influenced by maritime air mass.

Subdomain BAV, located in Southeast Germany, has the maximum of hail days at the end of July, later in the year than the other subdomains. Kunz and Puskeiler (2010) and Puskeiler (2013) also found that July is the month with the highest number of hail days in central and southern Germany.

Subdomain BEL, covering the North of France as well as the upper western part of Belgium, peaks at the end of June. A 10-year radar-based climatology conducted for Belgium by Lukach and Delobbe (2013) confirmed May and June to be the

most favorable months for hail.

The diurnal cycle of hailstorms shown in Figure 8 represents the times where the CCTA2D detects the first radar reflectivity of 55 dBZ or more. Since the local time (LT) varies through Europe with approximately one hour from Brittany in France to Saxony in Germany, all times originally given in UTC are converted to LT, representing four minutes per degree of longitude. In all subdomains, hail occurs most frequently in the afternoon between 13 and 18 LT, while between midnight and 10 LT the

fewest events are detected (Figure 8).

Some discrepancies appear in the daily cycle, mainly depending on the location and characteristics of the respective subdomain. For example, the frequency of hailstorms in BEL situated along the North Atlantic (but also in other subdomains located roughly North of latitude 48 °N and Mediterranean coastlines; not shown) reveals a large increase during the afternoon (14-15 LT) and a slow, but gradual decrease toward the morning.

In contrast to the subdomains located in the northern part of Europe, domains MAS over the Massif Central and SWF in South France peak one hour later at around 16 LT. The peak during the late afternoon for more continental regions is presumably due to local orographic effects, such as slope or valley winds (Nesbitt and Zipser, 2003).

For subdomain SWF, located in southern France between the Atlantic Ocean and the Pyrenees Mountains, the average number of hail days reminds high in the late evening (20 to 22 LT).

A plausible effect is that severe storms may develop from pre-existing scattered thunderstorms that form during the afternoon as was found by Nisi et al. (2016) and Nisi et al. (2018). This feature might be decisive for the hailstorm maximum in the evening in the canto of Ticino in southern Switzerland.

Some literature exists regarding the diurnal cycle of hail in Europe (Punge and Kunz, 2016). Bedka (2011), for example, recognized a diurnal cycle of overshooting tops that is related to the presence of orography and/or to the distance to the sea.

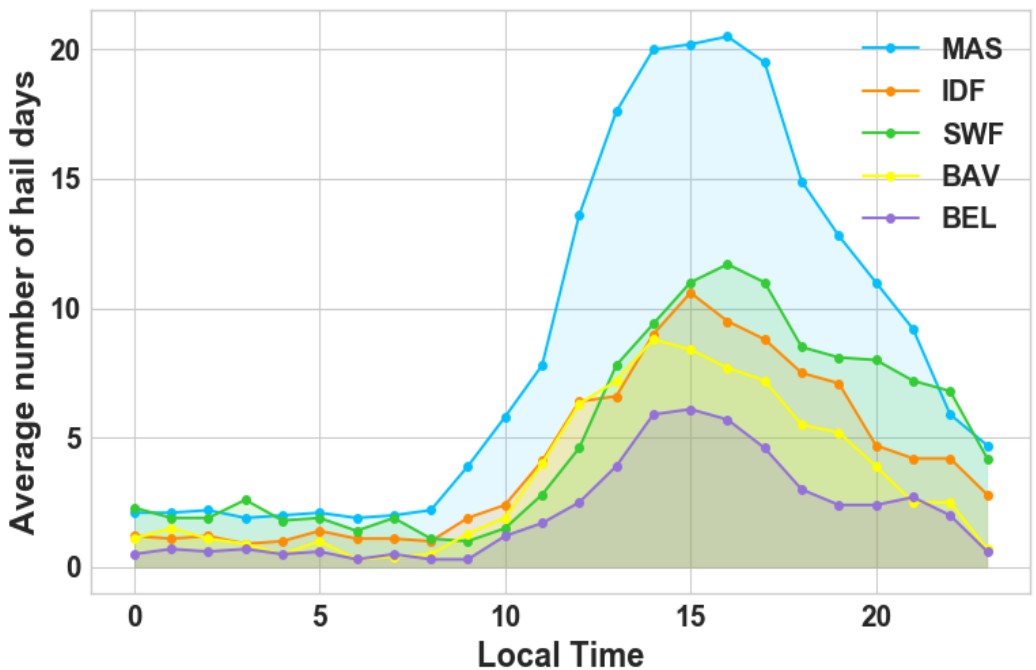

**Figure 8.** Hourly distribution of the mean number of radar-derived hail days for each subdomain.

Kaltenböck et al. (2009) found a peak in hail occurrence in the middle of the afternoon through Europe. Kunz and Puskeiler (2010) identified for southwestern Germany a maximum in the number of damaging hail events during 13 and 18 LT. The same peak is found in Alpine regions such as Italy (Morgan, 1973) or Switzerland (Nisi et al., 2016), where the maximum of hail occurs in the late afternoon and the minimum in the morning according to radar data analysis. Lukach et al. (2017) demonstrated for southeast Belgium that hail falls mostly during 15-16 UTC, which is in accordance to the daily cycle in subdomain BEL that includes Belgium. As for subdomain SWF in this study, Mallafré et al. (2009) found a peak later in the afternoon, around 18 LT.

## 4.4 First radar detection of SCS

To compare the spatial distribution of hailstorms during the night and day and to distinguish between mechanisms triggering nighttime events and convection being triggered within the boundary layer occurring preferably in the afternoon and early evening, we spatially analyzed the first detected signal of the radar-derived hailstorm tracks (hereinafter referred to as track onset). Figure 9 shows exemplary the temporal evolution of the spatial distribution of track onsets grouped into 3 hour intervals. As expected already from the daily cycles presented in Figure 8, the occurrence probability during the night is much lower than during the day (for example 573 onsets were detected from 2005 to 2014 during 0 LT and 3 LT compared to more than

2670 during 15 LT and 18 LT ). Furthermore, during nighttime (0-3 LT and 3-6 LT), the location of the onsets spreads more or less randomly along coastlines and the continent without any recognizable structure. Local-scale flow effects such as sea breezes may affect the triggering of convective cells near the coastlines, which may be reinforced over the Mediterranean as well as the Atlantic coastlines (Simpson, 1994). The morning at 6-9 LT and 9-12 LT experiences more onsets compared to the nighttime with some event clusters over the northeastern part of France or near the Massif-Central. During the day and the afternoon (12-15 LT and 15-18 LT), the track onsets form several patterns particularly near mountains, such as the Massif Central, the pre-Alpine domain in southern Germany, or near the French Pyrenees. Local effects, such as low-level flow convergence, and orographic effects, combined with large-scale features (fronts, large-scale lifting), may contribute to the reinforcement and development of convective cells near the mountain ranges (Kunz and Puskeiler, 2010, Berthet et al., 2011, Koebele, 2014, Kirshbaum et al., 2018). During the evening (18-21 LT), the track activity is still high, especially near mountain ranges. In contrast, inland regions and coastlines are less affected by hail events compared to the day. A substantial drop in the number of track onsets occurs during the night (21-0 LT), with only a few scattered onsets in southwestern France and near the Massif-Central.

## 4.5 Main characteristics of hail tracks

In the following section, we explore the main characteristics of our sample of radar-detected hail tracks. The length is defined as the distance in kilometers between the start and the end of a track determined by CCTA2D, i.e., the period where a threshold of 55 dBZ is reached or exceeded. The distribution of the lengths shown in the histogram in Figure 10 approximately follows an exponential function with a maximum for the first class.

In general, the mean length (with standard deviation) is $41.5 \pm 36.4$ km with a median of 29.5 km for the entire investigated area. The tracks reconstructed for Germany have a mean length of $39.1 \pm 33$ km and a median of 27 km whereas in France, the mean length is slightly larger with $43.9 \pm 39.8$ km and 32 km for the median. In total, 43% of all recorded storms over Western Europe have a length between 1 and 10 km. The number of tracks having a length L between 10 and 20 km decreases to 19% of the overall sample. Approximately 30% of all tracks have a length between 20 and 150 km, and less than 8% are greater than 150 km (not shown). Longer tracks can be expected for highly-organized convective systems, such as MCSs, or supercells.

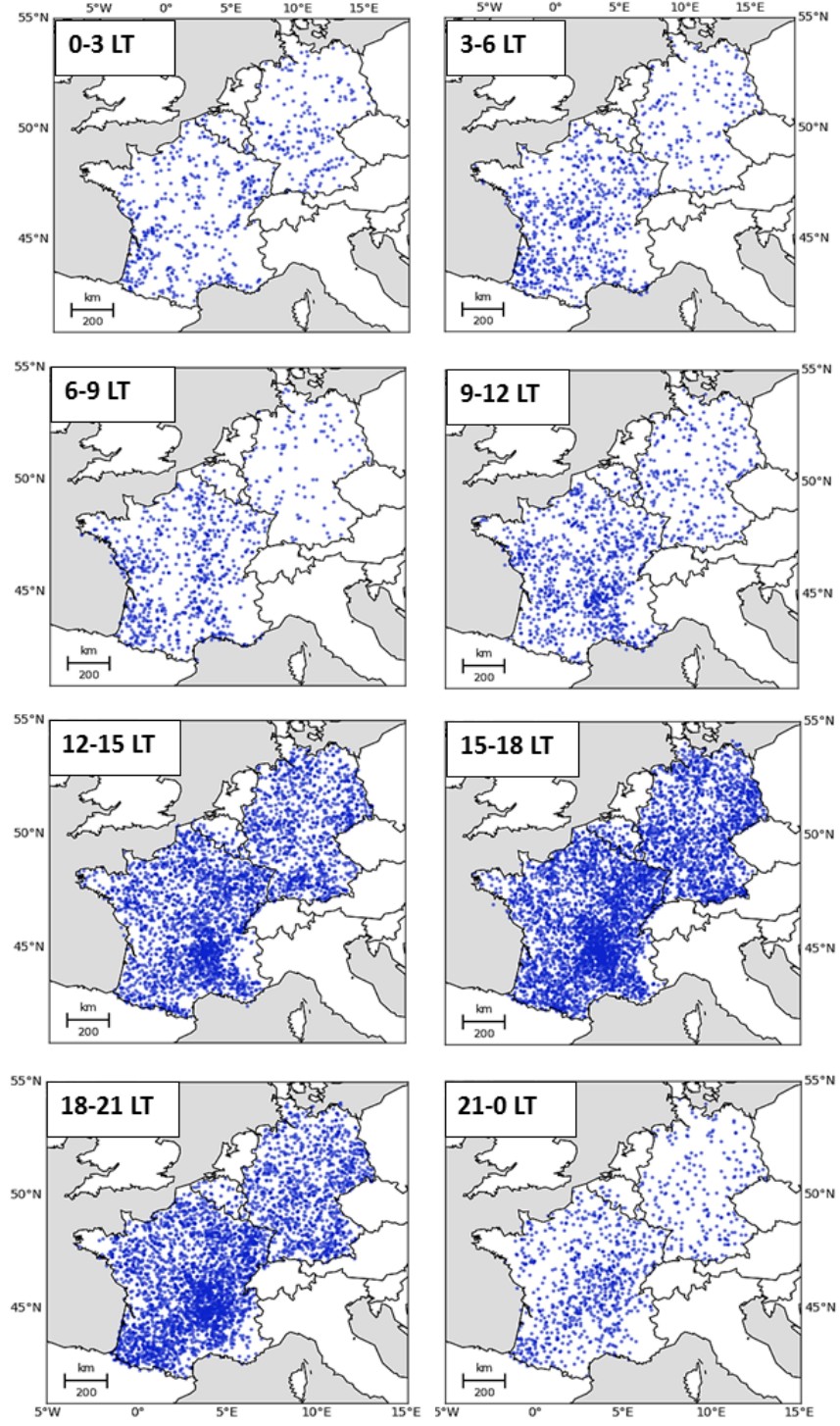

**Figure 9.** Locations of the first convective signatures detected by CCTA2D at the indicated Local Time (LT).

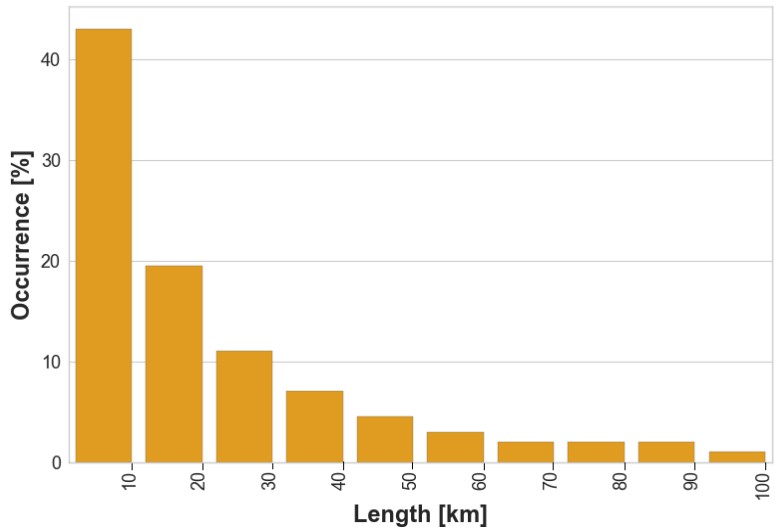

**Figure 10.** Histogram of all SCS mean length.

Only a few authors have analyzed hail tracks characteristics in West Europe, and only very few studies based their investi-
gations over a sufficiently long period. Puskeiler (2013), for example, investigated hail tracks lengths using 3D radar data in
Germany during 2005 and 2011 and found a mean length of 48 km with a strong decrease for longer streaks, inducing a high
standard deviation of 46.7 km and a median of 40 km.

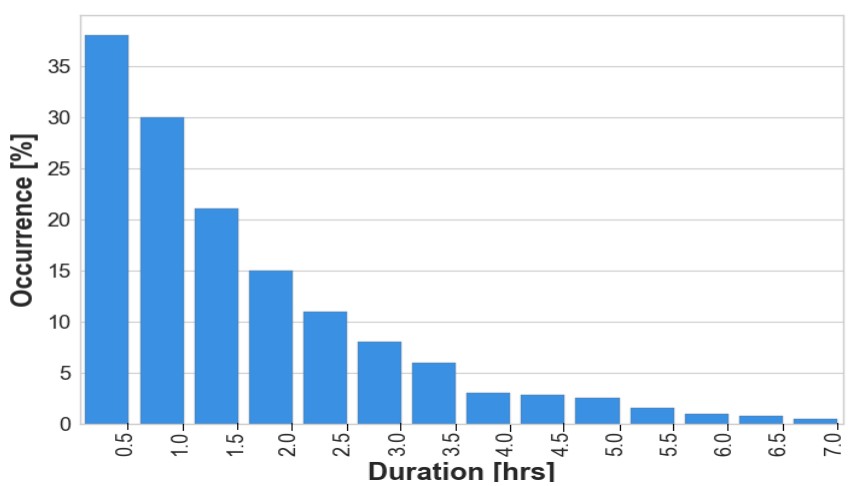

**Figure 11.** Same as Figure 10 but for the mean duration.

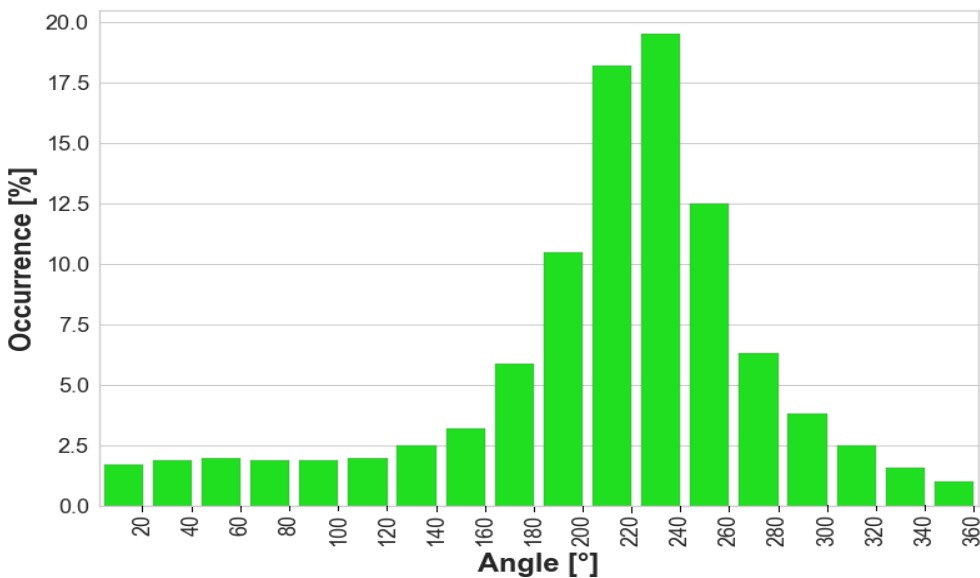

**Figure 12.** Same as Figure 11 but for the mean orientation.

Dessens (1986) found a mean length of 80 km for a small sample of 30 hailstorms in southwestern France. Note that Dessens (1986) used hail observations related to crop damage from the ANELFA network. For Spain, Mallafré et al. (2009) determined a mean hailstreak length of around $50 \pm 20$ km and used the Storm Cell Identification and Tracking algorithm (SCIT) elaborated by Johnson et al. (1998) on 3D radar data over northern Spain during 2004 and 2005 in order to identify hail cells.

The distribution of the hail track duration (Figure 11) is in accordance with the length, and also decreases almost exponentially with a peak at 30 minutes. As for the other physical characteristics, long-lived swaths are rare: only 2.4% of all cells persist over 5 hours (Fluck, 2018). The width, expressed as the maximum diameter of the largest reflectivity core ($Z \geq 55$ dBZ) during a hailstorm, or the longest distance between two cores evolving laterally in cases of cell-merging or splitting, show approximately a Gaussian distribution. The peak is between 9 and 10 km (41% of all events; not shown). However, as only the largest width of each swath is recorded, the results may be overestimated.

The track angle shown in Figure 12 represents the mean orientation of a storm track, and is the angle recorded by CCTA2D at the center of a swath, as most of the storm tracks are approximately linear. The orientation is defined as the angle between the line intersecting the reflectivity core centers, before and after the central point of a swath, and the parallel of latitude intersecting the start point. With this method, three time steps (i.e., 10 minutes) must exist for a track to be recorded. The maximum occurrence (19.3%) is found in the propagation direction between 220 and 240°, i.e., with a southwest direction. Around half (51%) of the hailstorms come from a west-southwest direction between 200 and 260°. Only 2.8% of the swaths have a northwesterly direction. The principal southwesterly swath orientation found in the statistics confirmed previous results,

such as that in the study of Puskeiler (2013), who found the highest number of hail days in southwest Germany with swaths oriented in southwesterly direction. In the Aquitaine region in France, Berthet et al. (2013) found that between 1952 and 1980 severe hailfalls came from a southwesterly direction with a mean angle of 241°.

## 5 Conclusions

This paper presents the first high resolution, radar-based hail statistics for a large central European region covering the countries
of France, Germany, Belgium, and Luxembourg over a 10-year period. A European radar composite has been created from French and German national radar composites with a five minute time step and corrected with lightning data. Tracks of SCS have been reconstructed using the tracking algorithm CCTA2D. Only grid points exceeding the Mason (1971) criterion for hail ($Z \geq 55$ dBZ) have been used for the hail assessment. From the spatial analyses of the hail signals, the following main results are obtained:

– The frequency of hailstorms shows a very large spatial variability across the investigation area. In general, there is a coast-to-continental increase in the number of hail days. While none or only several radar-derived hail days occurred in the northern parts of Germany, Brittany or along the European coastlines (0-2 hail days), the number of hail days far off the coasts is much higher.

       – Most of the hail hot spots are found on the leeward side of low-mountain ranges such as the Massif Central in France
or the Black Forest in Germany. The high spatial variability in the number of radar-derived hail days and the increasing number around orographic structures suggest a strong relationship between hailstorm occurrence and flow conditions induced or invigorated by orography such as a flow-around regime with subsequence flow convergence on the lee side.

       – On the regional-scale, significant differences in the seasonal and diurnal cycles of hail occurrences are found across Europe: In Southwest France, for instance, the hail maximum is in mid-June, but occurs two months later in August in
eastern Germany.

Our radar-derived hail frequency estimations and maps have, of course, several limitations and uncertainties. First, due to the local-scale nature of hailstorms and the lack of accurate observations, the reconstructed streaks and their statistics are difficult to validate. No homogeneous monitoring system for hail exists over the entire investigation region, but only some local networks, for example, the hailpad network over the Southwest, Central, and Southern France operated by ANELFA are
available. However, hailpad networks do not exist in Belgium, Luxembourg, and Germany. Thus, the use of radar and lightning data only provides proxys for hail occurrence.

Because only 2D radar data were available for this study, more sophisticated hail detection algorithm, such as those based on echo-top height (e.g. POH) or vertical integrals of reflectivity (e.g. MESH), which generally show higher skill in hail prediction (e.g.Skripniková and Řezáčová, 2014, Kunz and Kugel, 2015, Puskeiler et al., 2016), could not have been applied. The radar
coverage over Western Europe is reliable, but several regions are still not or not sufficiently covered by several radars, such as

the Alpine chain or some areas in the southeastern part of Germany and near Lake Constance. This leads to some data gaps in the final composite.

Despite the above mentioned limitations in our methods, the final results are in good accordance to other studies such as those for Germany based on 3D radar data (Puskeiler, 2013, Puskeiler et al., 2016, Schmidberger, 2018). The spatial distribution of hail signals in our study area is also similar to satellite-estimated hail frequency based on overshooting cloud tops as described by Punge et al. (2014) and Punge et al. (2017).

The investigations can be improved by extending the observation period until today. This is important especially in the subdomains highly exposed to hail. More accurate detections of hail can be achieved via the recently installed X-band radars in the French Alps. Furthermore, detailed investigations of the flow characteristics depending on atmospheric conditions, for example by using high-resolution numerical weather prediction models, can help to find robust evidence of the flow-around regime that may be decisive for the increased hail frequency downstream of several low mountain ranges, and can also contribute to a better understanding of the influence of orography on the triggering of convection.

*Data availability.* 2D radar data for Germany can be accessed via the German Weather Service (DWD) ftp server, while French national radar composites are available upon request to the French Meteorological Service (Météo-France). SCS/hail tracks were computed on radar data and are available upon request to M. Kunz. ERA5 data can be downloaded from the ECMWF server.

*Author contributions.* EF edited most parts of the paper, performed the statistical analyses and computed the SCS/hail tracks. MK verified in details the analytical methods and results, added crucial suggestions to the paper and contributed to the editing/revision of the manuscript. PG and SR added constructive suggestions to the paper. MK supervised the project in collaboration with PG and SR.

*Competing interests.* The authors declare that they have no conflict of interest.

*Acknowledgements.* The authors thank the French Meteorological Service (Météo-France) and the German Weather Service (DWD) for providing long-term radar data, and Siemens AG (S. Thern) for the supply of lightning data. The authors also acknowledge Tokio Millennium Re Ltd for funding the project. We acknowledge the constructive and very helpful comments from two anonymous reviewers that helped to improve the quality of this manuscript.

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
