# Peer review of "Radar-based assessment of hail frequency in Europe"

_Natural Hazards and Earth System Sciences, 2020_

## Referee Comment (RC1) · Anonymous Referee #1 · 26 Jun 2020

**Review for NHESS-2020-138**

**General Comments**

This paper presents a 10-year radar-based climatology of hail frequency in a portion of western Europe encompassing France, Germany, Belgium and Luxembourg. The authors combine 2D reflectivity composites from the French and German weather services into a single mosaic with a resolution of 1 km$^2$ and then applying storm cell tracking with a 55 dBZ reflectivity threshold to identify likely hail events. The spatial distribution of hail is analysed, with particular focus on the relation to surface topography (coastlines and mountain ranges), along with diurnal and seasonal variations in different parts of the study domain. The authors also examine the characteristics of the identified cell tracks, including their length, width, duration, and orientation.

The length and spatial extent of this analysis alone make it a novel contribution to the hail climatology literature, which often focuses on smaller regions. The paper is largely well written and the figures are mostly of a high quality. However, I see a number of issues that need to be addressed before this work can be accepted for publication. Chief among these is the unacceptable amount of speculation in the results, particularly when it comes to discussion around the role of surface topography in hailstorm formation. I would also like to see more details regarding the construction of the national radar composites and discussion on the importance of radar calibration errors. Detailed comments are provided below.

**Specific Comments**

*Major Comment 1:*
Currently, your results section contains too much speculation, particularly when it comes to the role of surface topography in the formation of hail storms. Examples include L197-207, L223-226, and L249-253. It is fine to note the clear correspondence between high hail frequencies and orographic features, but not to speculate at length on the underlying mechanisms in the absence of detailed observations or numerical simulations (either presented here or in other published studies). A bit of speculation is OK, but this should probably be reserved for the conclusions/discussion, where it can be used to motivate future investigation into physical mechanisms. Alternatively, if you do want to at least start this investigation here, you could use sounding or reanalysis data to examine the flow characteristics (wind speed and direction, Froude number, etc) on hail days in your various subdomains (c.f. section 6 of Kunz and Puskeiler 2010). This would obviously involve a bit of extra work (and additional data), but would make this study more than just "another hail climatology."

*Major Comment 2:*
I'd like to see a bit more detail regarding how the French and German radar composites are produced. For example, do they use radar from the lower (or lowest unblocked) tilt only or do they compute a column-maximum reflectivity across all tilts? How are reflectivities combined

in regions of overlapping coverage? Is the nearest radar used, the one with the lowest unblocked beam, or is a more complex quality index applied? Is any account taken for variations in beam diameter with range or differences in beam width between different radars? Such information is important for the reader to understand limitations in the composite product.

*Major Comment 3:*
Radar miscalibration is a major issue in many operational networks and can lead to significant inhomogeneities across large study domains. It is also a very tricky problem to overcome, although methods do exist (see, for example, Louf et al. 2019). However, for the purpose of this study I think you just need to mention it as a potential source of error in your results. Specifically, differences in radar calibration across the study domain may lead to an overestimation of the relative hail frequency in some regions (where radars are calibrated too high) and underestimation of the relative hail frequency in others (where radars are calibrated too low).

L3: My first question on seeing the study period is why does it end six years ago in 2014? Was one (or both) of the national composites not available for later dates? This information should be provided in section 2.

L26-36: You should also mention radar-based hail climatologies from other parts of the world, such as the USA (Cinineo et al. 2012) and Australia (Soderholm et al. 2017, Warren et al. 2020).

L39: I would argue that the issue isn't that these satellite- and model-based methods are "not as straightforward as those based on radar reflectivity". Rather it is that the link between the observed quantities and hail occurrence at the surface is less direct than it is with radar-based measurements.

L50: Your study provides information about the frequency of hail but not its intensity; as such this statement should be modified.

L53-54: Section 4 also presents results on seasonal and diurnal variations in hail frequency and the characteristics of the hail cells. Maybe mention this here.

L64: Why 2015 when your study covers 2005-2014? If the number or type of radars changed during your study period this should be mentioned.

L69-70: What map projection (coordinate system) does the French mosaic use?

L72-73: This sentence needs rephrasing. What sort of quality checks are performed?

L76: Does this mean that there is a gap in the data from mid-June to late-July 2009? Are there any other gaps during the study period? These will need to be accounted for if you estimate annual hail frequencies, as recommended below.

L77: Why did you bother processing the coarse resolution data from 1999 to 2004 when your study period only starts in 2005?

L80: Again, why discuss the state of the network in a year that falls outside your study period? How many radars were operational during 2005 to 2014 and did this number change?

L82-83: Figure 2 suggests that all of Germany is covered by the radar composite, with no gaps. The locations that you mention (the far north near the Danish border and southeastern Bavaria) are only covered by a single radar, but they will still surely feature in the composite.

L88-91: This type of data compression is quite common. Since the resolution of the data is quite high (0.5 dB) I don't think this needs to be discussed. It would only be worth mentioning if there were only a few reflectivity levels (as in Puskeiler et al. 2016).

L98: I'm not sure what you mean by the "standard coordinate system". What map projection is used? I'm guessing it differs from the ones used in the French and German composites. Was any account taken of this difference? Given that each domain covers around 1000 km, there could be some distortions introduced in this procedure.

L113-114: You say that "only reflectivity in the range of 35 to 70 dBZ was considered in the analyses", but all of your analysis considers a single threshold of 55 dBZ, so does this filtering really matter? Or are you saying that reflectivities below 35 dBZ or above 70 dBZ were set as missing values?

L114-116: This explanation is a little confusing. Looking back at Puskeiler's paper I see that reflectivities have to be >45 dBZ and 5 dBZ or more above the values at the neighbouring grid points to be filtered using Eq. 2. Please rephrase to make this clearer. Also the method doesn't really use a range of 2 km; rather it considers the 8 neighbouring grid points.

L118: What do you define as "a high reflectivity value"?

L119-120: I don't understand what you mean when you say "Reflectivity values near neighboring countries were evaluated and calibrate [sic] with radar stations close to the border." Please elaborate.

L123-125: I've personally never heard of lightning causing spurious radar signatures. If this is a real thing, surely it would represent an argument *against* using lightning data to filter out such signatures?

L125-128: While hailstorms typically do produce lightning, I am not aware of any work that shows that this lightning is always cloud-to-ground, which is the only type that you consider in your analysis. As such it is possible that you may have inadvertently filtered out hailstorms that produced only intracloud lightning. This should be noted as a caveat of the method described here.

L140-141: Is tracking only applied to reflectivity areas of ≥55 dBZ? This is a very high threshold for defining convective cells and is likely to lead to much shorter tracks than one would achieve using a more typical threshold such as 35 dBZ. It will also lead to an unrepresentative estimate of the location of convective initiation (Fig. 9), since developing cells may travel some distance before they achieve reflectivities as high as 55 dBZ. In my view, a better approach would be to identify and track cells using a lower threshold, but then only retain those that reach a reflectivity of at least 55 dBZ. This would also allow you to perform a comparative analysis of hailstorms and non-hailstorms. Perhaps this is outside of the scope of the present study, but it would certainly be a nice avenue for future work. At the very least you should note the caveats of using such a high reflectivity threshold for cell identification and tracking.

L145-148: Looking at Fig. 3a from Puskeiler et al. (2016), the difference in HSS between reflectivity thresholds of 55 and 56 dBZ is very small (both are around 0.6). You should note the corresponding values of POD and FAR: 0.7 and 0.4, respectively. While these values demonstrate reasonable skill, they also indicate that 30 % of observed hail events are missed while 40 % of those predicted are false alarms. This provides some idea of the uncertainty in your climatology.

L149-159: More details are needed concerning the tracking methodology. For example, what are the intensity and size criteria that are used to match cells between scans? How is a significantly different cell area defined for the purpose of identifying merges and splits? It might help to add a figure illustrating the process schematically.

L165: How is the horizontal wind field estimated? Also, I would describe it as a field of motion vectors, since storms do not move with the wind at any particular level.

L171: You say that ESWD reports were located close to the centre of the storm tracks "in most cases". What percentage of reports were not covered by the tracks? Can you comment on possible reasons (e.g. reflectivity <55 dBZ, erroneous report location/day)?

L180-182: It is good to reiterate that the 55 dBZ reflectivity threshold doesn't guarantee that hail occurred (and similarly, the absence of such high reflectivities doesn't guarantee that hail didn't occur). At this point you could refer back to section 3.3 where the results of Puskeiler et al. (2016) were discussed.

L191-194: If the mistral is cold and dry, is it really relevant to hailstorm formation?

L263: Hail frequencies are also lower over the high terrain of the Alps and Pyrenees, which is consistent with the results of Nisi et al. (2016, 2018).

L275: Does the average include only pixels within the area covered by radar? If not, this is how it should be done, otherwise you will artificially lower the average. You also shouldn't include points over the ocean, since these have been masked out in the map plots.

L299-300: I wouldn't say this result is particularly surprising. Large hail damage simply requires a few storms passing over densely populated areas, whereas Fig. 5 is considering the average number of hailstorms over a very large area.

L309: For simplicity, I would make all of the subdomains exactly the same shape and size. It looks like you would only need to modify boxes 11 and 13 for this to be the case. I would also suggest using a consistent 3-letter identifier for all regions, rather than numbers. These could be listed in a key/legend in Fig. 3 or in a separate table. The following would be my suggestions for the identifiers: 1 = NWG (North West Germany), 2 = NEG (North-East Germany), 3 = BEL (Belgium), 4 = WCG (West-Central Germany), 5 = ECG (East-Central Germany), 6 = NWF (North-West France), 7 = IDF (Île-de-France), 8 = LUX (Luxembourg), 9 = BAV (Bavaria), 10 = WCF (West-Central France), 11 = MAS (Massive Central), 12 = SWF (South-West France), 13 = MED (Mediterannean).

L316: It doesn't make sense to simply accumulate the number of hail days over all grid points in each subdomain. For one thing, some of the subdomains contain large areas that are over the sea and/or outside of radar coverage, which will give them a lower number than subdomains that are entirely over land and within radar coverage. It is also very difficult to interpret what these numbers mean. Instead, you should calculate the average number of hail days over all points with data (i.e. excluding those over the ocean and outside of radar coverage). This approach will give a much fairer comparison between the different regions. It's a good idea to use a moving average; however, from Fig. 3 it appears that you consider the preceding 10 days for this average. Instead, I would recommend using a 15-day moving average centred on the day in question (i.e. ±7 days).

L363: Again, you should consider the average number of hail days for each hour, not the total number of days over the domain. Also, why only consider the first time that a reflectivity of 55 dBZ is detected? Surely you should consider all times with 55 dBZ or higher in order to properly capture the diurnal cycle of hail (not just its initiation)?

L397-409: The problem with this analysis is that it assumes that the first detection of reflectivity ≥55 dBZ corresponds to the initiation of convection. In fact, developing convection may travel some distance before it reaches such an intensity, particularly in the presence of strong background flow (which would be expected in high-shear environments that favour severe storms). This is one reason why it would be advantageous to use a lower reflectivity threshold for identifying and tracking convective cells. At the very least, this caveat needs to be mentioned.

L418: You can probably just say lengths "between 10 and 20 km". Using 10.1 km as the lower bound implicitly excludes tracks with a length >10 km but <10.1 km. Alternatively, if you want to be more precise, you could define a variable $L$ to represent track length and then use "$10 < L \leq 20$ km" to represent this particular bin. Either way, the same change should be applied on L420.

L423-426: How did these studies define hail cells? If they used a lower reflectivity threshold, they are likely to get longer hail tracks because they will be including storms at earlier and

later stages of their life cycles and are also less likely to break up tracks where the reflectivity temporarily drops below 55 dBZ.

L427-428: While the results for hail track duration may be similar to those for track length, it would still be nice to include the results in Fig. 10. You could also (or alternatively) combine length and duration to compute storm motion estimates for each cell and examine the distribution of this.

L429-432: It would make more sense to compute the track width as the average diameter or the cell (computed over its lifetime) in the direction perpendicular to its movement, since this will actually correspond to the width of the underlying hail swath (under the assumption that hail falls where reflectivity ≥55 dBZ). If you make this change, I would strongly encourage you to include the results in Fig. 10.

L433-436: Again, rather than considering cells at a particular time, why not use the whole swath? For orientation, you could compute it simply as the angle between the first and last points in the trajectory (similar to how you define track length). Alternatively, you could apply a line of best fit to the set of points defining the trajectory. I would recommend a Theil-Sen fit for this purpose, as it is less sensitive to outliers for small sample sizes, compared to a linear least squares fit. Also, you should note that the angle is defined as the direction from which the storm is coming and is measured clockwise from north.

L467-468: The key issue with the lack of 3D radar data is the inability to use more sophisticated proxies such as those based on echo-top height (e.g. POH) or vertical integrals of reflectivity (e.g. MESH), which generally show higher skill in hail prediction (e.g. Skripniková and Řezáčová 2015; Kunz and Kugel 2015; Puskeiler et al. 2016).

Figure 3: I would recommend presenting these results as annual hail frequencies rather than counts of the total number of hail days. This will make it easier to compare your results with climatologies for different periods (including the maps for 2006 and 2010 in Fig. 6) and in other parts of the world (e.g. Cintineo et al. 2012; Warren et al. 2020). Also, you should mask out those grid points that fall outside of radar coverage, as shown in Fig. 2.

Figure 4: For the zoomed in view of the Massif Central region, it looks like you've just cut out and blown up a section of the fairly coarse-resolution image on the right. As such it's very difficult to make out the details in both the orography and the hail frequency contours. I don't think you need the map of the full study domain as this is already shown in Fig. 1. Instead, I would make this a multi-panel figure, showing zoomed-in views (at an appropriately high resolution) for several of the hail hotspots visible in Fig. 3 and discussed in section 4.1.

Figure 5: Please present these data as the actual number of hail days for each year rather than the difference from the mean (which can easily be inferred).

Figure 7: As noted above, rather than the total number of hail days in each subdomain, you should plot the mean number of hail days (excluding points that are over the sea or outside radar coverage). The same change should be applied to Fig. 8.

Figure 9: It would be nice to show these plots for other hours, rather than just 02 and 18 LT. For example, you could group the data into 3h blocks (00-03, 03-06, 06-09, 09-12, 12-15, 15-18, 18-21, and 21-00 LT) and present the results as an 8 panel figure.

**Technical Corrections**

I am not sure what the standard is for this journal, but in English (both American and British), a period is used as the decimal separator and a comma (or sometimes a thin space) is used to break up numbers of ten thousand or higher. For example, twelve-thousand three-hundred and forty-five point six would be written as 12,345.6.

L10-11: Change "spatially most extended" to simply "longest".

L12: Change "implied" to "produced" or "were associated with".

L12-13: Change "2 Billions Euros" to "€2 billion".

L20: This is not the correct use of "respectively" - it should only be used when describing two or more items that refer back to a previous statement. For example, "in northern Germany and southern France, hail occurs most frequently in August and June, respectively". The same comment applies to L96 and L269.

L23: Change "major part" to "majority".

L31: Puskeiler et al. (2016) consider the years 2005-2011, not 2004-2014.

L34: Change "criterions" to "criteria". Also "echo top" should be two words.

L46: Change "allows" to "allow".

L47: "sea" shouldn't be capitalized.

L213: Change "weaker" to "lower".

L229: Get rid of "recently".

L317: I think you meant to put "(Figure 7)" at the start of this line.

L386: The definition of overshooting tops should be given when they are introduced on L37.

L394: This doesn't need to be a new paragraph.

L421: Get rid of "including squall lines". Also, it should be "MCSs".

L446: "...from French and German national radar composites..."

L448: Duplication of "Mason".

L557: Get rid of "to" before "orography".

L466-467: Change "allows obtaining hail proxies" to "provides a proxy for hail occurrence".

Figure 1: Please use different line thicknesses, styles or colors to distinguish between country and state/distinct borders. The same applies to Fig. 3, 4, and 6.

Figure 6: The colour bar is incorrectly labelled. The number of hail days in a single year will always be an integer, so you don't need the range or the decimal place (i.e. the labels should just be 1, 2, 3, …, 12).

Figure 10: The x axes of these plots are incorrectly labelled. Each bin corresponds to a range of values, so the tick labels should be located under the ticks to illustrate this. So, for example, for panel (a) the ticks should be labelled 0, 10, 20, …, 310.

**References**

Cintineo, J.L., Smith, T.M., Lakshmanan, V., Brooks, H.E. and Ortega, K.L. (2012) An objective high-resolution hail climatology of the contiguous United States. *Weather and Forecasting*, **27**, 1235–1248.

Louf, V., Protat, A., Warren, R.A., Collis, S.M., Wolff, D.B, Raunyiar, S., Jakob, C. and Petersen, W.A. (2019) An Integrated Approach to Weather Radar Calibration and Monitoring Using Ground Clutter and Satellite Comparisons. *Journal of Atmospheric and Oceanic Technology*, **36**, 17–39.

Skripniková, K. and Řezáčová, D. (2014) Radar-based hail detection. *Atmospheric Research*, **144**, 175–185.

Soderholm, J., McGowan, H., Richter, H., Walsh, K., Weckwerth, T. and Coleman, M. (2017) An 18-year climatology of hailstorm trends and related drivers across southeast Queensland, Australia. *Quarterly Journal of the Royal Meteorological Society*, **143**, 1123–1135.

Warren, R.A., Ramsay, H.A., Siems, S.T., Peter, J.R., Protat, A. and Pillalamarri, A. (2020) Radar-based climatology of damaging hailstorms in Brisbane and Sydney, Australia. *Quarterly Journal of the Royal Meteorological Society*, **146**, 505–530.

---

## Referee Comment (RC2) · Anonymous Referee #2 · 28 Aug 2020

**Review**

The paper "Radar-based assessment of hail frequency in Europe" is targeted on climatology of severe convection storms (SCS) based on 10 years of radar data covering a large European region. The topic is very important and suitable for NHESS. The analysis covers a relatively large number of years (considering typical radar records) and geographic domain.

My main concern is if we are really sure these are hail cases. The paper, is it is now, is not clear about this. The text some times refer the data set as SCS and sometimes as hail cases.

On one hand, it seems some validation has been done: "Tests with long-living SCS tracks were compared with hail reports archived by the European Severe Weather Database (ESWD) operated by the European Severe Storms Laboratory (Dotzek et al., 2009) along the reconstructed storm trajectories to assess the reliability of CCTA2D (not shown). In most cases, the ESWD reports were located close to the center of SCS tracks." (Lines 169-172). But, just next to it, at the beginning of the results section, there is a "disclaimer": "Note that this climatology represents the spatial distribution of convective cells with high reflectivity, but not directly of hail. The term hail days used in the following refers to the exceedance of reflectivity, but not to confirmed hail observations" (Lines 180-182). I understand the first case refers to tracks while the second to the spatial distribution, but I think it would be better to clarify in a targeted section whether or not there is any validation that the analyzed storms are indeed hail events. Also, through out the text try to be more consistent in the use of SCS vs. hail events according to the level of assurance of the nature of these events.

Furthermore, validation seems quite crucial here, as without it, the data set may not represent hail events. So I encourage the authors to present the validation done against the hail reports from ESWD and if possible to extend it.

**Other comments:**

It would be good to provide a short background on hail formation, under what meteorological conditions we should expect hail events. This would help in understanding the interpretation of the presented results.

The data set includes radar data from two countries which goes some different processing procedures. To be sure this does not add any bias in results – is there any overlap region where analysis from both data sources can be compared?

Lightening filter: "If high reflectivity during a day occurs without lightning, the values at the affected grid points are set to zero." (Line 127-128): why only during day? Are you sure this filter is not too aggressive? Can you provide any information on percent of hail storms that are not associate with lightening? if I understand correctly these storms will be filtered out from the analysis and it is important to verify their fraction is not substantial.

Split and merge: the authors write that "Special attention is given to cell splitting and merging" (Line 154). Why is that? I did not find in the results any consideration of the splits and merges that were detected.

Line 277: "As shown in Figure 5 the annual variability is very high and without any trend" – for 10 years of data I would not consider a trend for 10 years of data.

Line 283-285: "large-scale lifting (e.g., related to differential vorticity advection) could have led to an increase in convective available potential energy (CAPE) and a low convective inhibition (CIN). The combination of high moisture in the boundary layer, low CIN, high CAPE and lifting mechanisms may give rise to a substantial increase in SCS." It is not clear if this is an assumption or analysis. Why not to check reanalysis data for CAPE, CIN, air moisture anomalies? without it, I think this sentence is too speculative.

General comment: the authors provide a very detailed description of the pattern shown in the figures. In my opinion this is too lengthy and could be shorten. I leave this however for the author decision.

---

## Author Comment (AC1) · 24 Sep 2020

**The paper "Radar-based assessment of hail frequency in Europe" is targeted on climatology of severe convection storms (SCS) based on 10 years of radar data covering a large European region. The topic is very important and suitable for NHESS. The analysis covers a relatively large number of years (considering typical radar records) and geographic domain**.
**The authors are very grateful to the reviewer who found the topic of the article important and suitable for NHESS. We are thankful for all the questions, comments and suggestions, which will all be considered in the revised version of the paper.**

**My main concern is if we are really sure these are hail cases. The paper, is it is now, is not clear about this. The text some times refer the data set as SCS and**

[Figure]

sometimes as hail cases. On one hand, it seems some validation has been done: "Tests with long-living SCS tracks were compared with hail reports archived by the European Severe Weather Database (ESWD) operated by the European Severe Storms Laboratory (Dotzek et al., 2009) along the reconstructed storm trajectories to assess the reliability of CCTA2D (not shown). In most cases, the ESWD reports were located close to the center of SCS tracks." (Lines 169-172). But, just next to it, at the beginning of the results section, there is a "disclaimer": "Note that this climatology represents the spatial distribution of convective cells with high reflectivity, but not directly of hail. The term hail days used in the following refers to the exceedance of reflectivity, but not to confirmed hail observations" (Lines 180-182). I understand the first case refers to tracks while the second to the spatial distribution, but I think it would be better to clarify in a targeted section whether or not there is any validation that the analyzed storms are indeed hail events. Also, through out the text try to be more consistent in the use of SCS vs. hail events according to the level of assurance of the nature of these events. Furthermore, validation seems quite crucial here, as without it, the data set may not represent hail events. So I encourage the authors to present the validation done against the hail reports from ESWD and if possible to extend it.

**The authors understand the concern of the reviewer. A clear identification of hail within the SCS tracks and especially a separation between hail and heavy rainfalls is not possible giving the lack of comprehensive hail observations and the use of a proxy. However, in the Puskeiler et al. (2016) paper, which is based on similar methods (in the 2 D version), we have computed different skill scores using different insurance loss data. Furthermore, in the recent paper of Kunz et al., (2020), the authors separate the SCS events (same SCS data as in this study) from hailstorm events (HS) by assessing the presence of hail using ESWD reports in the vicinity of SCS tracks. Out of 26 012 SCS events in total from 2005 to 2014, only 985 events could be considered as hail events. However, several hail events that are not observed and captured by ESWD**

reports lead to the large reduction of cases. Indeed, out of the 4 577 ESWD reports available from 2005 to 2014, Germany counts most of the reports with 76.5% followed by France with 21.1% then Belgium with 7.1% and finally Luxembourg with 0.7%. We will add a statement on the reliability and skill referring to the two papers.

**Other comments: It would be good to provide a short background on hail forma-tion, under what meteorological conditions we should expect hail events. This would help in understanding the interpretation of the presented results.**
**We will add a short background paragraph about hail formation and environmental conditions in the revised version.**

**The data set includes radar data from two countries which goes some different processing procedures. To be sure this does not add any bias in results – is there any overlap region where analysis from both data sources can be compared?**
**A huge effort is done by both French and German weather services to homogenize their radar products, in order to provide among others, high-resolution European radar composites within the program OPERA. Thus, a further quality check step compared to inland radars is applied on neighboring countries. An overlap region between the two countries is for example present over the Rhine Valley including Northeast France and the Southwest part of Germany. The authors gave a special attention to this area at the beginning of their research to "twist" the tracking algorithm and compared reflectivity values from both countries during few hail events (For example on 28th July 2013 during the severe thunderstorms and hailstorms that affected Southwest Germany (Kunz et al., (2020)). Even if the reflectivity values differ lightly from one country to another, the tracking algorithm includes corrections (which will be detailed in the revised article) to get the best tracking location. Furthermore, the advection correction used after the tracking permits a smoothing of the reflectivity values to get uniform values with no apparent differences over overlapped regions. We will comment on those points in the revised version.**

**Lightening filter: "If high reflectivity during a day occurs without lightning, the values at the affected grid points are set to zero." (Line 127-128): why only during day? Are you sure this filter is not too aggressive? Can you provide any information on percent of hail storms that are not associate with lightening? if I understand correctly these storms will be filtered out from the analysis and it is important to verify their fraction is not substantial.**

**More details about the lightning filter methods will be added in the revised manuscript. The filter runs for day and nighttime (day in the above sentence is meant as 24-hour period); we will clarify this. Note that the lightning filter only marginally affect the results as shown in Puskeiler (2013; PhD thesis, but only in Germany).**

**Split and merge: the authors write that "Special attention is given to cell splitting and merging" (Line 154). Why is that? I did not find in the results any consideration of the splits and merges that were detected.**

**Sorry, this is formulated misleadingly and will be corrected. This attention is considered during the tracking calculation only. It can happen that a part of a storm splits from the main path and develops into left (or right)-moving supercells leading to hail on the ground. This happened for example on 28th July 2013. In order to avoid any hail underestimation (overestimation), a splitting (merging) option in the tracking algorithm is necessary. Furthermore, the physical characteristics of hail swaths could be compromised/wrong without splitting or merging options. The results in this study present a global hail-statistic including both merging and splittings.**

**Line 277: "As shown in Figure 5 the annual variability is very high and without any trend" – for 10 years of data I would not consider a trend for 10 years of data.**
**We will reedit this sentence.**

**Line 283-285: "large-scale lifting (e.g., related to differential vorticity advection) could have led to an increase in convective available potential energy (CAPE)**

**and a low convective inhibition (CIN). The combination of high moisture in the boundary layer, low CIN, high CAPE and lifting mechanisms may give rise to a substantial increase in SCS." It is not clear if this is an assumption or analysis. Why not to check reanalysis data for CAPE, CIN, air moisture anomalies? without it, I think this sentence is too speculative.**

**We agree on your assessment and will delete this speculative statement. A recent article of Kunz et al., (2020) assess the ambient conditions during hail events for the period 2005 to 2014 using the same tracks and 2D radar data as this study. We will add a comment and present the main results in the revised paper version.**

**General comment: the authors provide a very detailed description of the pattern shown in the figures. In my opinion this is too lengthy and could be shorten. I leave this however for the author decision.**

**We will reread thoroughly this section and will delete what is not necessary. In particular we will delete the speculations given here.**

---

## Author Comment (AC2) · 24 Sep 2020

**General Comments**

This paper presents a 10-year radar-based climatology of hail frequency in a portion of western Europe encompassing France, Germany, Belgium and Luxembourg. The authors combine 2D reflectivity composites from the French and German weather services into a single mosaic with a resolution of 1 km 2 and then applying storm cell tracking with a 55 dBZ reflectivity threshold to identify likely hail events. The spatial distribution of hail is analysed, with particular focus on the relation to surface topography (coastlines and mountain ranges), along with diurnal and seasonal variations in different parts of the study domain. The authors also examine the characteristics of the identified cell tracks, including their length, width, duration, and orientation.

[Figure]

**The length and spatial extent of this analysis alone make it a novel contribution to the hail climatology literature, which often focuses on smaller regions. The paper is largely well written and the figures are mostly of a high quality. However, I see a number of issues that need to be addressed before this work can be accepted for publication. Chief among these is the unacceptable amount of speculation in the results, particularly when it comes to discussion around the role of surface topography in hailstorm formation. I would also like to see more details regarding the construction of the national radar composites and discussion on the importance of radar calibration errors. Detailed comments are provided below.**

**We would like to thank the reviewer for the very constructive comments and the time spent on the review of this article. We are also cheerful that the reviewer appreciated the article both for its written quality and for the analyses carried out. The authors are very grateful that the reviewer attributes this paper to a novel contribution to the hail climatology. We will delete unjustified speculations, but also add additional investigations about the role of topography in hailstorm formation. In preparing a revised manuscript, we will address all major and minor suggestions.**

**Specific Comments**

**Major Comment 1:**

**Currently, your results section contains too much speculation, particularly when it comes to the role of surface topography in the formation of hail storms. Examples include L197-207, L223-226, and L249-253. It is fine to note the clear correspondence between high hail frequencies and orographic features, but not to speculate at length on the underlying mechanisms in the absence of detailed observations or numerical simulations (either presented here or in other published studies). A bit of speculation is OK, but this should probably**

be reserved for the conclusions/discussion, where it can be used to motivate
future investigation into physical mechanisms. Alternatively, if you do want to at
least start this investigation here, you could use sounding or reanalysis data to
examine the flow characteristics (wind speed and direction, Froude number, etc)
on hail days in your various subdomains (c.f. section 6 of Kunz and Puskeiler
2010). This would obviously involve a bit of extra work (and additional data), but
would make this study more than just "another hail climatology."

**We thank the reviewer for this constructive comment. We will add some additional**
investigations to justify our speculations. We computed the flow (direction and speed at
10 m) in mountainous area (near the Massif-Central, Pyrenees and Vosges in France;
and nearby the Black-Forest and Ore Mountains in Germany) as well as the Froude
number using ERA5 reanalysis that we overlapped on ETOPO1, a high-resolution
global relief (1 arc-minute resolution). The low Froude numbers support the hypothesis
of a predominant flow-around wind regime during hail days. These assumptions will
be detailed in the reviewed paper version.

**Major Comment 2:**

I'd like to see a bit more detail regarding how the French and German radar
composites are produced. For example, do they use radar from the lower (or
lowest unblocked) tilt only or do they compute a column-maximum reflectivity
across all tilts? How are reflectivities combined in regions of overlapping
coverage? Is the nearest radar used, the one with the lowest unblocked beam,
or is a more complex quality index applied? Is any account taken for variations
in beam diameter with range or differences in beam width between different
radars? Such information is important for the reader to understand limitations
in the composite product.

**We will provide more technical details about the computation of the French and the**
German national radar composites from the single-radars in a new paragraph.

**Major Comment 3:**

**Radar miscalibration is a major issue in many operational networks and can lead to significant inhomogeneities across large study domains. It is also a very tricky problem to overcome, although methods do exist (see, for example, Louf et al. 2019). However, for the purpose of this study I think you just need to mention it as a potential source of error in your results. Specifically, differences in radar calibration across the study domain may lead to an overestimation of the relative hail frequency in some regions (where radars are calibrated too high) and underestimation of the relative hail frequency in others (where radars are calibrated too low).**

**We are aware of this problem, but cannot fix it because radar reflectivity for France is available only as a composite and not for individual radars. As suggested, we will add more comments about radar miscalibrations leading to data inhomogeinities.**

**Minor comments**

**L3: My first question on seeing the study period is why does it end six years ago in 2014? Was one (or both) of the national composites not available for later dates? This information should be provided in section 2.**

**This is a good point. The hail climatology presented in this paper was investigated within the first phase of the project HAMLET (Hail Model for Europe by Tokio Millennium) that lasted from 2013 until mid-2017 and focused mainly on the climatological aspect of hail and historical events in Europe. During the first phase of HAMLET, the French national radar composites were available until 2014 only, due to the installation of five new X-band radars in the Alpine region that requested an addition into the French national radar composite with the resulting adjustments. The calibrated French national radar composites from 2014 up to nowadays were only available on later**

times. We will add a comment on this.

**L26-36: You should also mention radar-based hail climatologies from other parts of the world, such as the USA (Cinineo et al. 2012) and Australia (Soderholm et al. 2017, Warren et al. 2020).**
**We will add further radar-based hail climatology references to the paper.**

**L39: I would argue that the issue isn't that these satellite- and model-based methods are "not as straightforward as those based on radar reflectivity. Rather it is that the link between the observed quantities and hail occurrence at the surface is less direct than it is with radar-based measurements.**
**We agree with that statement and will change the sentence accordingly. Note that within our group we have derived several satellite-based hail climatologies (e.g., Punge et al., 2017; Bedka et al., 2018), which show several similarities, but also some discrepancies mainly because of the use of overshooting tops that are weaker proxys for hail compared to radar reflectivity and because of the low resolution of the satellite data.**

**L50: Your study provides information about the frequency of hail but not its intensity; as such this statement should be modified.**
**We will remove "intensity" from the sentence.**

**L53-54: Section 4 also presents results on seasonal and diurnal variations in hail frequency and the characteristics of the hail cells. Maybe mention this here.**
**We will add these insights to the introduction.**

**L64: Why 2015 when your study covers 2005-2014? If the number or type of radars changed during your study period this should be mentioned.**

We will rephrase this sentence and add a short historical evolution of the French radar network. We mentioned here about the number and location of French radar stations in 2015 because the French Weather Service shared publically the radar stations location map in 2015 only.

**L69-70: What map projection (coordinate system) does the French mosaic use?**
**The single radars use a polar stereographic projection and the mosaic is referred to a plane Cartesian coordinate system. We will add this information in the new version of the manuscript.**

**L72-73: This sentence needs rephrasing. What sort of quality checks are performed?**
**We will add more details about how the quality checks are computed for the French mosaic.**

**L76: Does this mean that there is a gap in the data from mid-June to late-July 2009? Are there any other gaps during the study period? These will need to be accounted for if you estimate annual hail frequencies, as recommended below.**
There were no missing days without a radar national composite during the whole period from 2005 to 2014. Few local single-radar data might miss on very specific times but M é t é o-France did not report any major radar failures that could affect our climatology. The coarser data resolution was in fact available until the end of June 2009. We will modify this sentence.

**L77: Why did you bother processing the coarse resolution data from 1999 to 2004 when your study period only starts in 2005?**
**The coarser resolution was available from 1999 until the end of June 2009. Therefore we had to interpolate the data with the coarser grid from 2004 until June 2009 to the**

finer grid resolution available after June 2009. The year 1999 was just given as a general information. We will remove "1999" and change it for "2004".

**L80: Again, why discuss the state of the network in a year that falls outside your study period? How many radars were operational during 2005 to 2014 and did this number change?**
In order to avoid misunderstandings in the revised manuscript, we will only mention the year 2014. During 2005 to 2014, one radar station was added in 2012 in Memmingen (South Germany). We will mention this new radar implementation and its data implication in the German mosaic in the new paper version.

**L82-83: Figure 2 suggests that all of Germany is covered by the radar composite, with no gaps. The locations that you mention (the far north near the Danish border and southeastern Bavaria) are only covered by a single radar, but they will still surely feature in the composite.**
**Yes, but looking at long-term radar composites, the far north place near the Danish border and the southeastern Bavaria have no reflectivity values; we rather see some values near the location of the radar stations.**

**L88-91: This type of data compression is quite common. Since the resolution of the data is quite high (0.5 dB) I don't think this needs to be discussed. It would only be worth mentioning if there were only a few reflectivity levels (as in Puskeiler et al. 2016).**
We agree and will delete this statement.

**L98: I'm not sure what you mean by the "standard coordinate system". What map projection is used? I'm guessing it differs from the ones used in the French and German composites. Was any account taken of this difference? Given that each domain covers around 1000 km, there could be some distortions**

**introduced in this procedure.**
**We used WGS84 as geographic coordinate system (EPSG code: 4326) with the following three properties: 1. Datum is set to WGS84 with a 6378137 m equatorial radius for the oblate ellipsoid at the equator and a flattening of 1/298.257223563. 2. The prime meridian is Greenwich. 3. Units are in degrees. We used the software ArcGIS that by default plot a map with a Pseudo Plate Carree projection. In the revised version, we will set all maps to a Lambert Conformal Conic projection that best suit for mid-latitudes.**

**L113-114: You say that "only reflectivity in the range of 35 to 70 dBZ was considered in the Analyses", but all of your analysis considers a single threshold of 55 dBZ, so does this filtering really matter? Or are you saying that reflectivities below 35 dBZ or above 70 dBZ were set as missing values?**
We will rephrase these sentences. Reflectivity values below 35dBZ and above 70 dBZ were filter out of the dataset. The 35 dBZ threshold is used to define and to detect intense precipitation areas in the tracking algorithm. Our analyses, however, only estimate footprints for a reflectivity of 55 dBZ.

**L114-116: This explanation is a little confusing. Looking back at Puskeiler's paper I see that reflectivities have to be >45 dBZ and 5 dBZ or more above the values at the neighbouring grid points to be filtered using Eq. 2. Please rephrase to make this clearer. Also the method doesn't really use a range of 2 km; rather it considers the 8 neighbouring grid points.**
We will rephrase this sentence and switch the 2 km range with the 8 neighbouring grid points. Note that for the Puskeiler et al. (2016) study, the authors have applied slightly different methods since the focus there was on 3D reflectivity.

**L118: What do you define as "a high reflectivity value"?**
**In this context, when a grid point get a reflectivity value that is at least twice higher**

compared to the 8 neighboring grid points, then this grid point is considered to have a high reflectivity value. We will add an explanation.

**L119-120: I don't understand what you mean when you say "Reflectivity values near neighboring countries were evaluated and calibrate [sic] with radar stations close to the border." Please elaborate.**
**We will rephrase this sentence. There is a common effort between the German and the French Weather Service to homogenize the French and the German radar mosaic together. Thus the data from radars covering both Germany and France experience a further quality check step compared to inland radars.**

**L123-125: I've personally never heard of lightning causing spurious radar signatures. If this is a real thing, surely it would represent an argument against using lightning data to filter out such signatures?**
**We will provide a more appropriate echoes example in the revised paper version.**

**L125-128: While hailstorms typically do produce lightning, I am not aware of any work that shows that this lightning is always cloud-to-ground, which is the only type that you consider in your analysis. As such it is possible that you may have inadvertently filtered out hailstorms that produced only intracloud lightning. This should be noted as a caveat of the method described here.**
**We choose to avoid cloud-to-cloud and intra-cloud flashes because the EUCLID lightning detection system had a significant lower detection efficiency (Pohjola and Mäkelä, 2013). Thus only cloud-to-ground flashes were considered in this study. We will mention that some hailstorms may have been inadvertently filter out in our analysis.**

**L140-141: Is tracking only applied to reflectivity areas of ≥55 dBZ? This is a very high threshold for defining convective cells and is likely to lead to much shorter tracks than one would achieve using a more typical threshold such as 35 dBZ. It will also lead to an unrepresentative estimate of the location of**

**convective initiation (Fig. 9), since developing cells may travel some distance before they achieve reflectivities as high as 55 dBZ. In my view, a better approach would be to identify and track cells using a lower threshold, but then only retain those that reach a reflectivity of at least 55 dBZ. This would also allow you to perform a comparative analysis of hailstorms and non-hailstorms. Perhaps this is outside of the scope of the present study, but it would certainly be a nice avenue for future work. At the very least you should note the caveats of using such a high reflectivity threshold for cell identification and tracking.**
**We thank the reviewer for pointing out some misunderstandings about how convective cells are detected by the tracking algorithm. We use in fact few reflectivity thresholds that will be described in the revised manuscript.**

**L145-148: Looking at Fig. 3a from Puskeiler et al. (2016), the difference in HSS between reflectivity thresholds of 55 and 56 dBZ is very small (both are around 0.6). You should note the corresponding values of POD and FAR: 0.7 and 0.4, respectively. While these values demonstrate reasonable skill, they also indicate that 30 % of observed hail events are missed while 40 % of those predicted are false alarms. This provides some idea of the uncertainty in your climatology.**
**This is a very good point. Even though the algorithm used in Puskeiler et al. 2016 is slightly different to our tracking algorithm, the results are almost similar, thus HSS, POD and FAR values are comparable. We will add a statement about the skill and uncertainty of our climatology.**

**L149-159: More details are needed concerning the tracking methodology. For example, what are the intensity and size criteria that are used to match cells between scans? How is a significantly different cell area defined for the purpose of identifying merges and splits? It might help to add a figure illustrating the process schematically.**
**We will provide more details on the tracking algorithm and add a table with thresholds**

used in CCT2D.

**L165: How is the horizontal wind field estimated? Also, I would describe it as a field of motion vectors, since storms do not move with the wind at any particular level.**
**We will add further details about the computation of the horizontal wind field and we will describe it as a field of motion vectors. Basically, each point along a track detected by the CCTA2D algorithm includes a velocity shift-vector in the north-to-south (dv) or west-to-east (du) directions. The so-called shift-vector is denoted as:**

$$\overrightarrow{U}(x,y) = (\,d\,)\,u(x,y)dv(x,y) \qquad (1)$$

As the CCTA2D algorithm only determines the (weighted) center of a track, an n-time parallel duplication of the track is required with a vector field from all locations of the thunderstorm. The parallel shift on position (b; c) is done with normalized vectors **t1** and **t2** on the cell motion direction with a spacing of 20 km maximum on each side of the track. The position (b;c) obtained from the shift vector U is calculated as follows:

$$(b,c) = \left( x + |\frac{\delta \overrightarrow{t}_{1,2}}{\delta x}| \cdot n, y + |\frac{\delta \overrightarrow{t}_{1,2}}{\delta y}| \right) \qquad (2)$$

This gives a displacement field of the entire cell complex.

**L171: You say that ESWD reports were located close to the centre of the storm tracks "in most cases. What percentage of reports were not covered by the tracks? Can you comment on possible reasons (e.g. reflectivity <55 dBZ,**

**erroneous report location/day)?**
**The percentage of ESWD reports without having a track vary from country to country (Kunz et al. 2020), as the quality and reliability of the hail reports in ESWD is very heterogeneous with most of the reports available after 2005. Furthermore, even many European countries are involved in the European Severe Storm Laboratory (ESSL), Germany counts the highest number of reports (Groenemeijer and Kühne, 2014) and only a few hail reports are available for France, Belgium or Luxembourg. In our analysis, less than 5% of confirmed hail reports (QC1) from the ESWD were not located on hail swaths in France against 7% for Germany. Few reports appeared in regions less (or not) covered by a radar station (e.g, far north Belgium, Alps region, Bavaria in Germany). Another reason for finding a hail report without a track is that the day or location of the observation was erroneous (wrong day, mistake by setting the exact hail location, selecting the wrong type of convective event).**

**L180-182: It is good to reiterate that the 55 dBZ reflectivity threshold doesn't guarantee that hail occurred (and similarly, the absence of such high reflectivities doesn't guarantee that hail didn't occur). At this point you could refer back to section 3.3 where the results of Puskeiler et al. (2016) were discussed.**
**We will reiterate the fact that the 55 dBZ threshold doesn't guarantee hail on the ground and vice-versa.**

**L191-194: If the mistral is cold and dry, is it really relevant to hailstorm formation?** #No, we will delete these sentences.

**L263: Hail frequencies are also lower over the high terrain of the Alps and Pyrenees, which is consistent with the results of Nisi et al. (2016, 2018).**
**We thank the reviewer for the comparison with other literature and we will add this sentence in the new paper version.**

**L275: Does the average include only pixels within the area covered by radar? If not, this is how it should be done, otherwise you will artificially lower the average. You also shouldn't include points over the ocean, since these have been masked out in the map plots.**

**Yes, the average includes only pixels within the area covered by radar. An Ocean mask was also applied on each single radar mosaic so that the data on ocean are filtered out of the analysis. We will add a comment on this.**

**L299-300: I wouldn't say this result is particularly surprising. Large hail damage simply requires a few storms passing over densely populated areas, whereas Fig. 5 is considering the average number of hailstorms over a very large area.**

**We agree to this statement. We will rephrase this sentence and get rid of "surprising". (Note that the year 2013 was really exceptional in a sense that the 27/28 July hailstorms were the first damaging events that occurred in that year in Germany. In the last 30 years the first heavy hailstorms have never occurred so late.)**

**L309: For simplicity, I would make all of the subdomains exactly the same shape and size. It looks like you would only need to modify boxes 11 and 13 for this to be the case. I would also suggest using a consistent 3-letter identifier for all regions, rather than numbers. These could be listed in a key/legend in Fig. 3 or in a separate table. The following would be my suggestions for the identifiers: 1 = NWG (North West Germany), 2 = NEG (North-East Germany), 3 = BEL (Belgium), 4 = WCG (West-Central Germany), 5 = ECG (East-Central Germany), 6 = NWF (North-West France), 7 = IDF (Île-de-France), 8 = LUX (Luxembourg), 9 = BAV (Bavaria), 10 = WCF (West-Central France), 11 = MAS (Massive Central), 12 = SWF (South-West France), 13 = MED (Mediterannean).**

**We will follow this suggestion and will change the sizes of boxes 11 and 13 and the box names and will add acronyms to the figure caption.**

[Figure]

**L316: It doesn't make sense to simply accumulate the number of hail days over all grid points in each subdomain. For one thing, some of the subdomains contain large areas that are over the sea and/or outside of radar coverage, which will give them a lower number than subdomains that are entirely over land and within radar coverage. It is also very difficult to interpret what these numbers mean. Instead, you should calculate the average number of hail days over all points with data (i.e. excluding those over the ocean and outside of radar coverage). This approach will give a much fairer comparison between the different regions. It's a good idea to use a moving average; however, from Fig. 3 it appears that you consider the preceding 10 days for this average. Instead, I would recommend using a 15-day moving average centred on the day in question (i.e. ±7 days).**

**We agree with the reviewer and we will swap the total number of haildays for the average number of haildays for each subdomain so that an intercomparison between the regions is possible. After testing moving averages, a 10-day moving average was the best representation of the temporal hail distribution for each subdomain. For example, using a 15-day moving average would have "flattened" some curves and we wouldn't recognize all graph pics.**

**L363: Again, you should consider the average number of hail days for each hour, not the total number of days over the domain. Also, why only consider the first time that a reflectivity of 55 dBZ is detected? Surely you should consider all times with 55 dBZ or higher in order to properly capture the diurnal cycle of hail (not just its initiation)?**

**We will compute the average number of haildays for each hour in the subdomains. The motivation was actually to investigate where the first hail signals appear (e.g. near topography? If yes, on the leeward or on the windward side of the mountain?)**
**L397-409: The problem with this analysis is that it assumes that the first detection of reflectivity ≥55 dBZ corresponds to the initiation of convection. In fact, developing convection may travel some distance before it reaches such an intensity, particularly in the presence of strong background flow (which would be expected in high-shear environments that favour severe storms). This is one reason why it would be advantageous to use a lower reflectivity threshold for identifying and tracking convective cells. At the very least, this caveat needs to be mentioned.**

**We use actually a few and lower reflectivity thresholds to identify a convective cell. The thresholds will be detailed in the revised paper.**

**L418: You can probably just say lengths "between 10 and 20 km". Using 10.1 km as the lower bound implicitly excludes tracks with a length >10 km but <10.1 km. Alternatively, if you want to be more precise, you could define a variable ðİŘ£ to represent track length and then use "$10 < L \leq 20$ km" to represent this particular bin. Either way, the same change should be applied on L420.**

**We will use the nomenclature $10 < L \leq 20$ km.**

**L423-426: How did these studies define hail cells? If they used a lower reflectivity threshold, they are likely to get longer hail tracks because they will be including storms at earlier and later stages of their life cycles and are also less likely to break up tracks where the reflectivity temporarily drops below 55 dBZ.**

**Note that Dessens (1986) didn't use remote-sensing data but only hail observations in Southwest France from 1952 to 1981 (network ANELFA) that lead to crop damages. Only hailstones with a diameter equals or superior to 15 mm and an area of 15km$^2$defined a hail cell, where at least one hailfall have been reported. The different methods and time period analyzed in Dessens (1986) might explain the longer mean track length compared to our analysis. Mallafre et al., (2008) used the Storm Cell**

Identification and Tracking algorithm (SCIT) elaborated by Johnson et al., (1998) on 3D radar data over North Spain during 2004 and 2005. SCIT uses by default 35 dBZ for cell detection out of six other reflectivity thresholds candidates. The selection of the radar threshold depends on the reflectivity values. The higher the reflectivity, the higher is the cell detection threshold. The shorter period used in this study and the fact that a convective cell can be define in Mallafre et al. (2008), the different thresholds from the SCIT could explained the slight longest length track mean compared to our analysis. We will briefly mention the different methods.

**L427-428: While the results for hail track duration may be similar to those for track length, it would still be nice to include the results in Fig. 10. You could also (or alternatively) combine length and duration to compute storm motion estimates for each cell and examine the distribution of this.**
**Indeed the track duration tendency looks like the track length. We will add a graph showing the track duration in the revised version.**

**L429-432: It would make more sense to compute the track width as the average diameter or the cell (computed over its lifetime) in the direction perpendicular to its movement, since this will actually correspond to the width of the underlying hail swath (under the assumption that hail falls where reflectivity ≥55 dBZ). If you make this change, I would strongly encourage you to include the results in Fig. 10.**
**Only the track width maximum was stored by the algorithm.**

**L433-436: Again, rather than considering cells at a particular time, why not use the whole swath? For orientation, you could compute it simply as the angle between the first and last points in the trajectory (similar to how you define track length). Alternatively, you could apply a line of best fit to the set of points defining the trajectory. I would recommend a Theil-Sen fit for this purpose, as**

it is less sensitive to outliers for small sample sizes, compared to a linear least squares fit. Also, you should note that the angle is defined as the direction from which the storm is coming and is measured clockwise from north.

**The only limitation with that method is that the main track trajectory should be a straight line. For longer tracks, the storm direction changes with time (for ex. by cells splitting/merging). By selecting a unique time step, we are confident about the parameters recorded for a specific grid point. We will take note about the angle direction.**

L467-468: The key issue with the lack of 3D radar data is the inability to use more sophisticated proxies such as those based on echo-top height (e.g. POH) or vertical integrals of reflectivity (e.g. MESH), which generally show higher skill in hail prediction (e.g. Skripniková and ÅŸezáčová2015; Kunz and Kugel 2015; Puskeiler et al. 2016).

**We will comment on this.**

Figure 3: I would recommend presenting these results as annual hail frequencies rather than counts of the total number of hail days. This will make it easier to compare your results with climatologies for different periods (including the maps for 2006 and 2010 in Fig. 6) and in other parts of the world (e.g. Cintineo et al. 2012; Warren et al. 2020). Also, you should mask out those grid points that fall outside of radar coverage, as shown in Fig. 2.

**We will follow this suggestion and represent Figure 3 as annual hail frequencies under radar coverage, and with an ocean mask.**

Figure 4: For the zoomed in view of the Massif Central region, it looks like you've just cut out and blown up a section of the fairly coarse-resolution image on the right. As such it's very difficult to make out the details in both the orography and the hail frequency contours. I don't think you need the map of

the full study domain as this is already shown in Fig. 1. Instead, I would make
this a multi-panel figure, showing zoomed-in views (at an appropriately high
resolution) for several of the hail hotspots visible in Fig. 3 and discussed in
section 4.1.

**We will delete the coarse resolution zoomed-in-view of the Massif Central and replace**
it by a high-resolution Global relief map overlaid with hail frequency contours and wind
flow. The result will be discuss in section 4.1.

**Figure 5: Please present these data as the actual number of hail days for
each year rather than the difference from the mean (which can easily be in-
ferred).**

**We will present the data as actual number of hail days for each year (See attached**
Figure 1).

**Figure 7: As noted above, rather than the total number of hail days in each
subdomain, you should plot the mean number of hail days (excluding points
that are over the sea or outside radar coverage). The same change should be
applied to Fig. 8.**

**We will consider the suggested changes.**

**Figure 9: It would be nice to show these plots for other hours, rather than
just 02 and 18 LT. For example, you could group the data into 3h blocks (00-03,
03-06, 06-09, 09-12, 12-15, 15-18, 18-21, and 21-00 LT) and present the results as
an 8 panel figure.**

**We agree to plot the hail tracks starting points over a longer time window and**
consider 3h blocks. A 8 panel figure following this idea is attached in Figure 2 below.

**Technical Corrections**

I am not sure what the standard is for this journal, but in English (both American and British), a period is used as the decimal separator and a comma (or sometimes a thin space) is used to break up numbers of ten thousand or higher. For example, twelve-thousand three-hundred and forty-five point six would be written as 12,345.6. L10-11: Change "spatially most extended" to simply "longest". L12: Change "implied" to "produced" or "were associated with". L12-13: Change "2 Billions Euros" to "€ billion". L20: This is not the correct use of "respectively" - it should only be used when describing two or more items that refer back to a previous statement. For example, "in northern Germany and southern France, hail occurs most frequently in August and June, respectively". The same comment applies to L96 and L269. L23: Change "major part" to "majority". L31: Puskeiler et al. (2016) consider the years 2005-2011, not 2004-2014. L34: Change "criterions" to "criteria". Also "echo top" should be two words. L46: Change "allows" to "allow". L47: "sea" shouldn't be capitalized. L213: Change "weaker" to "lower". L229: Get rid of "recently". L317: I think you meant to put "(Figure 7)" at the start of this line. L386: The definition of overshooting tops should be given when they are introduced on L37. L394: This doesn't need to be a new paragraph. L421: Get rid of "including squall lines". Also, it should be "MCSs". L446: "...from French and German national radar composites..." L448: Duplication of "Mason". L557: Get rid of "to" before "orography". L466-467: Change "allows obtaining hail proxies" to "provides a proxy for hail occurrence". Figure 1: Please use different line thicknesses, styles or colors to distinguish between country and state/distinct borders. The same applies to Fig. 3, 4, and 6. Figure 6: The colour bar is incorrectly labelled. The number of hail days in a single year will always be an integer, so you don't need the range or the decimal place (i.e. the labels should just be 1, 2, 3, . . ., 12). Figure 10: The x axes of these plots are incorrectly labelled. Each bin corresponds to a range of values, so the tick labels should be located under the ticks to illustrate this. So, for example, for panel (a) the ticks should be labelled 0, 10, 20, . . ., 310.

**We are thankful for the technical corrections provided by the reviewer, which will be considered.**

**References**

Cintineo, J.L., Smith, T.M., Lakshmanan, V., Brooks, H.E. and Ortega, K.L. (2012) An objective high-resolution hail climatology of the contiguous United States. Weather and Forecasting , 27 , 1235–1248. Louf, V., Protat, A., Warren, R.A., Collis, S.M., Wolff, D.B, Raunyiar, S., Jakob, C. and Petersen, W.A. (2019) An Integrated Approach to Weather Radar Calibration and Monitoring Using Ground Clutter and Satellite Comparisons. Journal of Atmospheric and Oceanic Technology , 36 , 17–39. Skripniková, K. and ÅŸezáčová, D. (2014) Radar-based hail detection. Atmospheric Research , 144 , 175–185. Soderholm, J., McGowan, H., Richter, H., Walsh, K., Weckwerth, T. and Coleman, M. (2017) An 18-year climatology of hailstorm trends and related drivers across southeast Queensland, Australia. Quarterly Journal of the Royal Meteorological Society , 143 , 1123–1135.

**We thank the reviewer for providing the literature reference.**

![Bar chart titled Number of hail days versus Years from 2005 to 2014, with a red line at 86 representing the average number of hail days during the 10 year period.]

**Fig. 1.** Number of annual hail days from 2005 to 2014. The red line represents the average number of hail days during the 10 years period.

**Fig. 2.** Locations of the 55 dBZ reflectivity detected by CCTA2D. From upper left corner to lower right corner: 0-3, 3-6, 6-9, 9-12, 12-15, 15-18, 18-21 and 21-0 LT.

---

## Author Response (AR1)

**Answers for Anonymous Referee #1**

General Comments

This paper presents a 10-year radar-based climatology of hail frequency in a portion of western Europe encompassing France, Germany, Belgium and Luxembourg. The authors combine 2D reflectivity composites from the French and German weather services into a single mosaic with a resolution of 1 km 2 and then applying storm cell tracking with a 55 dBZ reflectivity threshold to identify likely hail events. The spatial distribution of hail is analysed, with particular focus on the relation to surface topography (coastlines and mountain ranges), along with diurnal and seasonal variations in different parts of the study domain. The authors also examine the characteristics of the identified cell tracks, including their length, width, duration, and orientation.

The length and spatial extent of this analysis alone make it a novel contribution to the hail climatology literature, which often focuses on smaller regions. The paper is largely well written and the figures are mostly of a high quality. However, I see a number of issues that need to be addressed before this work can be accepted for publication. Chief among these is the unacceptable amount of speculation in the results, particularly when it comes to discussion around the role of surface topography in hailstorm formation. I would also like to see more details regarding the construction of the national radar composites and discussion on the importance of radar calibration errors. Detailed comments are provided below.

**We thank again the reviewer for his report and suggestions. We deleted unjustified speculations and added a new section with the role of surface topography in hailstorm formation with a special emphasis on the Massif Central in France, which is the region the most affected by hail in our study. We also added a paragraph about the construction and quality control procedures of both French and German national radar composites.**
All major and minor corrections suggested by the reviewer were addressed in the new manuscript and appear in blue in the corrected version.

Specific Comments

*Major Comment 1:*

Currently, your results section contains too much speculation, particularly when it comes to the role of surface topography in the formation of hail storms. Examples include L197-207, L223-226, and L249-253. It is fine to note the clear correspondence between high hail

frequencies and orographic features, but not to speculate at length on the underlying mechanisms in the absence of detailed observations or numerical simulations (either presented here or in other published studies). A bit of speculation is OK, but this should probably be reserved for the conclusions/discussion, where it can be used to motivate future investigation into physical mechanisms. Alternatively, if you do want to at least start this investigation here, you could use sounding or reanalysis data to examine the flow characteristics (wind speed and direction, Froude number, etc) on hail days in your various subdomains (c.f. section 6 of Kunz and Puskeiler 2010). This would obviously involve a bit of extra work (and additional data), but would make this study more than just "another hail climatology."

**We removed some speculations for example in Sections 4.1, 4.2 and 4.3. As we wanted to start the investigation about the wind flow, we computed the flow (direction and velocity at 10 m) during days with hail in mountainous area (near the Massif-Central, Pyrenees and Vosges in France; and nearby the Black-Forest and Ore Mountains in Germany). The Froude number was also calculated using ERA5 reanalysis that we overlapped on ETOPO1, a high-resolution global relief (1 arc-minute resolution). The low Froude numbers in all regions support the hypothesis of a predominant flow-around wind regime during hail days. In the reviewed manuscript version, an example is shown for the Massif Central only in Figure 4, and a detailed description of Figure 4 is following from Line 316 to Line 346 in the new manuscript version. Finally, we compared our results with some recent literature about hailstorms occurrence in the vicinity of topography in Europe; as well as convection initiation in complex terrain (now Lines 347 to 355).**

*Major Comment 2:*

I'd like to see a bit more detail regarding how the French and German radar composites are produced. For example, do they use radar from the lower (or lowest unblocked) tilt only or do they compute a column-maximum reflectivity across all tilts? How are reflectivities combined in regions of overlapping coverage? Is the nearest radar used, the one with the lowest unblocked beam, or is a more complex quality index applied? Is any account taken for variations in beam diameter with range or differences in beam width between different radars? Such information is important for the reader to understand limitations in the composite product.

**We followed the suggestion of the reviewer by adding two detailed paragraphs, one for France (Lines 101 to 149) and another one for Germany (Lines 150 to 173), both of them including: 1. A short historic of the national radar network. 2. A description of individual radars (types of radar + brief geographical location + enumeration of radar changes + observation techniques). 3. A description of the national radar composite merging using individual radars. 4. Strategies adopted while considering regions with overlapping radar scans 5. Signal processing. 6. Quality-controls applied at each production step.**

*Major Comment 3:*

Radar miscalibration is a major issue in many operational networks and can lead to significant inhomogeneities across large study domains. It is also a very tricky problem to overcome, although methods do exist (see, for example, Louf et al. 2019). However, for the purpose of this study I think you just need to mention it as a potential source of error in your results. Specifically, differences in radar calibration across the study domain may lead to an overestimation of the relative hail frequency in some regions (where radars are calibrated too high) and underestimation of the relative hail frequency in others (where radars are calibrated too low).

**As suggested, we added more comments in the text about radar miscalibrations leading to data inhomogeinities (now Line 179 to Line 185).**

*Minor comments*

L3: My first question on seeing the study period is why does it end six years ago in 2014? Was one (or both) of the national composites not available for later dates? This information should be provided in section 2.

**We added two sentences to explain that this study focuses on the first phase of the project HAMLET (Lines 90 and 91) where radar data from France where available until 2014 only. This is due to the installation of five new X-band radars in 2014 in the French radar composite requiring further calibrations and quality controls into the national radar composite.**

L26-36: You should also mention radar-based hail climatologies from other parts of the world, such as the USA (Cinineo et al. 2012) and Australia (Soderholm et al. 2017, Warren et al. 2020).

**We commented on additional radar-based hail climatologies from other parts of the world including the reviewer's literature suggestions (Lines 56 to 64).**

L39: I would argue that the issue isn't that these satellite- and model-based methods are "not as straightforward as those based on radar reflectivity. Rather it is that the link between the observed quantities and hail occurrence at the surface is less direct than it is with radar-based measurements.

**We changed this statement for "the link between the observed quantities and hail occurrence at the surface is less direct than with radar-based measurements".**

L50: Your study provides information about the frequency of hail but not its intensity; as such this statement should be modified.

**We removed "intensity" from the sentence.**

L53-54: Section 4 also presents results on seasonal and diurnal variations in hail frequency and the characteristics of the hail cells. Maybe mention this here.

**We added these statements in now Lines 85-87.**

L64: Why 2015 when your study covers 2005-2014? If the number or type of radars changed during your study period this should be mentioned.
**We rephrased this sentence (Line 92).**

L69-70: What map projection (coordinate system) does the French mosaic use?
**The French mosaics are available in Cartesian coordinates, as described on the website of the French Weather Service:**

https://donneespubliques.meteofrance.fr/?fond=rubrique&id_rubrique=27

L72-73: This sentence needs rephrasing. What sort of quality checks are performed?
**We deleted this sentence. A strict protocol of quality checks is applied for both French and German national radar composites. We detailed the main quality check stages for the French radar composites in Lines 124 to 141. These steps are also described for the German radar composites from Line 159 to Line 167.**

L76: Does this mean that there is a gap in the data from mid-June to late-July 2009? Are there any other gaps during the study period? These will need to be accounted for if you estimate annual hail frequencies, as recommended below.
**We modified this sentence in the new version of the manuscript (now line 146). There were no missing days without a radar national composite during the whole period from 2005 to 2014. Few local single-radar data might miss on very specific times but Météo-France did not report any major radar failures that could affect our climatology.**

L77: Why did you bother processing the coarse resolution data from 1999 to 2004 when your study period only starts in 2005?
**We removed "1999" and change it for "2004".**

L80: Again, why discuss the state of the network in a year that falls outside your study period? How many radars were operational during 2005 to 2014 and did this number change?
**We only mention the years 2005 to 2014 in the reviewed version (line 150). During 2005 to 2014, one radar station was added in 2012 in Memmingen (South Germany). The number of radar did not changed during 2005 and 2014 for France but few radars were replaced. We added these assumption in L151-152 for Germany and in L103-114 for France.**

L82-83: Figure 2 suggests that all of Germany is covered by the radar composite, with no gaps. The locations that you mention (the far north near the Danish border and southeastern Bavaria) are only covered by a single radar, but they will still surely feature in the composite.

**Looking at long-term radar composites, the far north place near the Danish border and the southeastern part of Bavaria have no reflectivity values; we rather see some values near the location of the radar stations. We added these sentences in L154-156.**

L88-91: This type of data compression is quite common. Since the resolution of the data is quite high (0.5 dB) I don't think this needs to be discussed. It would only be worth mentioning if there were only a few reflectivity levels (as in Puskeiler et al. 2016).
**We deleted this statement.**

L98: I'm not sure what you mean by the "standard coordinate system". What map projection is used? I'm guessing it differs from the ones used in the French and German composites.
Was any account taken of this difference? Given that each domain covers around 1000 km, there could be some distortions introduced in this procedure.
**Details about the coordinate system were added in L185-189. We used WGS84 as geographic coordinate system (EPSG code: 4326) with the following three properties: 1. Datum is set to WGS84 with a 6378137 m equatorial radius for the oblate ellipsoid at the equator and a flattening of 1/298.257223563. 2. The prime meridian is Greenwich. 3. Units are in degrees.**
We used the software ArcGIS that by default plot a map with a Pseudo Plate Carree projection. In the revised version, all maps were projected to a Lambert Conformal Conic projection that best suit for mid-latitudes. An applied example of this projection can be find in Figure 3.

L113-114: You say that "only reflectivity in the range of 35 to 70 dBZ was considered in the Analyses", but all of your analysis considers a single threshold of 55 dBZ, so does this filtering really matter? Or are you saying that reflectivities below 35 dBZ or above 70 dBZ were set as missing values?
**We rephrased these sentences for "To avoid this problem, reflectivities below 35 dBZ or above 70 dBZ were set as missing values" (L211-212). The 35 dBZ threshold is used to define and to detect intense precipitation areas by the tracking algorithm. Details about this tracking algorithm and the thresholds used are provided in a new paragraph (now L236-243).**

L114-116: This explanation is a little confusing. Looking back at Puskeiler's paper I see that reflectivities have to be >45 dBZ and 5 dBZ or more above the values at the neighbouring grid points to be filtered using Eq. 2. Please rephrase to make this clearer. Also the method doesn't really use a range of 2 km; rather it considers the 8 neighbouring grid points.
**We rephrased this sentence and switch the 2 km range with the 8 neighboring grid points (L213-214). Note that for the Puskeiler et al. (2016) study, the authors applied slightly different methods since the focus there was on 3D reflectivity.**

L118: What do you define as "a high reflectivity value"?
**We added an explanation in L216-217. This includes the reflectivity values of the 8 neighboring grid points.**

L119-120: I don't understand what you mean when you say "Reflectivity values near neighboring countries were evaluated and calibrate [sic] with radar stations close to the

border." Please elaborate.

**We changed this sentence and added it in L182-184.**

L123-125: I've personally never heard of lightning causing spurious radar signatures. If this is a real thing, surely it would represent an argument *against* using lightning data to filter out such signatures?

**We removed "lightning" from the sentence and provided another echo type (L220).**

L125-128: While hailstorms typically do produce lightning, I am not aware of any work that shows that this lightning is always cloud-to-ground, which is the only type that you consider in your analysis. As such it is possible that you may have inadvertently filtered out hailstorms that produced only intracloud lightning. This should be noted as a caveat of the method described here.

**We added a sentence explaining why we avoid intra-cloud and cloud-to-cloud lightning (L195-197) and we agree that some hailstorms may have been inadvertently filter out in our analysis.**

L140-141: Is tracking only applied to reflectivity areas of ≥55 dBZ? This is a very high threshold for defining convective cells and is likely to lead to much shorter tracks than one would achieve using a more typical threshold such as 35 dBZ. It will also lead to an unrepresentative estimate of the location of convective initiation (Fig. 9), since developing cells may travel some distance before they achieve reflectivities as high as 55 dBZ. In my view, a better approach would be to identify and track cells using a lower threshold, but then only retain those that reach a reflectivity of at least 55 dBZ. This would also allow you to perform a comparative analysis of hailstorms and non-hailstorms. Perhaps this is outside of the scope of the present study, but it would certainly be a nice avenue for future work. At the very least you should note the caveats of using such a high reflectivity threshold for cell identification and tracking.

**We added a short paragraph (L236-240) with descriptions about the tracking algorithm techniques and thresholds used. Convective cells were detected with a 35 dBZ threshold but only grid points overpassing 55 dBZ were suggested to be more able to represent hail and were selected for our study.**

L145-148: Looking at Fig. 3a from Puskeiler et al. (2016), the difference in HSS between reflectivity thresholds of 55 and 56 dBZ is very small (both are around 0.6). You should note the corresponding values of POD and FAR: 0.7 and 0.4, respectively. While these values demonstrate reasonable skill, they also indicate that 30 \% of observed hail events are missed while 40 \% of those predicted are false alarms. This provides some idea of the uncertainty in your climatology.

**We added a statement about the skill and uncertainty of our climatology in the new manuscript version in L247-252 considering the results of Puskeiler et al. (2016).**

L149-159: More details are needed concerning the tracking methodology. For example, what are the intensity and size criteria that are used to match cells between scans? How is a

significantly different cell area defined for the purpose of identifying merges and splits? It might help to add a figure illustrating the process schematically.

**In the revised manuscript we added a detailed description of the tracking algorithm including explanations about thresholds used and how the algorithm tracks convective cells in space and time. This description is now in L253-268.**

L165: How is the horizontal wind field estimated? Also, I would describe it as a field of motion vectors, since storms do not move with the wind at any particular level.

**We added further details and equations about the computation of the horizontal wind field (L-276-287) and we described it as a field of motion vectors (Equation 2).**

L171: You say that ESWD reports were located close to the centre of the storm tracks "in most cases. What percentage of reports were not covered by the tracks? Can you comment on possible reasons (e.g. reflectivity <55 dBZ, erroneous report location/day)?

**We added a short explanation in the reviewed manuscript about the percentage of ESWD reports without having a track by following the results of Kunz et al. 2020 who focused on this question and used the same database as in this study (L288-291). We also provide some explanations about possible reasons for ESWD reports not covered by tracks (now L292-294).**

L180-182: It is good to reiterate that the 55 dBZ reflectivity threshold doesn't guarantee that hail occurred (and similarly, the absence of such high reflectivities doesn't guarantee that hail didn't occur). At this point you could refer back to section 3.3 where the results of Puskeiler et al. (2016) were discussed.

**We reiterated the fact that the 55 dBZ threshold doesn't guarantee hail on the ground and vice-versa (now line 305 to 307).**

L191-194: If the mistral is cold and dry, is it really relevant to hailstorm formation?

**We deleted these sentences.**

L263: Hail frequencies are also lower over the high terrain of the Alps and Pyrenees, which is consistent with the results of Nisi et al. (2016, 2018).

**We added these literature suggestion in lines 303-304.**

L275: Does the average include only pixels within the area covered by radar? If not, this is how it should be done, otherwise you will artificially lower the average. You also shouldn't include points over the ocean, since these have been masked out in the map plots.

**Yes, the average includes only pixels within the area covered by radar. An ocean mask was also applied on each single radar mosaic so that the data on ocean are filtered out of the analysis.**

L299-300: I wouldn't say this result is particularly surprising. Large hail damage simply requires a few storms passing over densely populated areas, whereas Fig. 5 is considering

the average number of hailstorms over a very large area.
**We deleted this sentence.**

L309: For simplicity, I would make all of the subdomains exactly the same shape and size. It looks like you would only need to modify boxes 11 and 13 for this to be the case. I would also suggest using a consistent 3-letter identifier for all regions, rather than numbers. These could be listed in a key/legend in Fig. 3 or in a separate table. The following would be my suggestions for the identifiers: 1 = NWG (North West Germany), 2 = NEG (North-East Germany), 3 = BEL (Belgium), 4 = WCG (West-Central Germany), 5 = ECG (East-Central Germany), 6 = NWF (North-West France), 7 = IDF (Île-de-France), 8 = LUX (Luxembourg), 9 = BAV (Bavaria), 10 = WCF (West-Central France), 11 = MAS (Massive Central), 12 = SWF (South-West France), 13 = MED (Mediterannean).
**We followed this suggestion and changed the sizes of boxes 11 and 13. We also changed the box names by adding the acronyms suggestions of the reviewer (Figure 3) and applied a Lambert Conic projection on the map.**

L316: It doesn't make sense to simply accumulate the number of hail days over all grid points in each subdomain. For one thing, some of the subdomains contain large areas that are over the sea and/or outside of radar coverage, which will give them a lower number than subdomains that are entirely over land and within radar coverage. It is also very difficult to interpret what these numbers mean. Instead, you should calculate the average number of hail days over all points with data (i.e. excluding those over the ocean and outside of radar coverage). This approach will give a much fairer comparison between the different regions. It's a good idea to use a moving average; however, from Fig. 3 it appears that you consider the preceding 10 days for this average. Instead, I would recommend using a 15-day moving average centred on the day in question (i.e. ±7 days).
**We computed the average number of hail days. The changes are described in lines 451-485 and appear on figure 7. After having tested different moving averages (5, 7, 10, 15 and 20 days), a 10-day moving average was the best representation of the temporal hail distribution of each subdomain.**

L363: Again, you should consider the average number of hail days for each hour, not the total number of days over the domain. Also, why only consider the first time that a reflectivity of 55 dBZ is detected? Surely you should consider all times with 55 dBZ or higher in order to properly capture the diurnal cycle of hail (not just its initiation)?
**We computed the average number of hail days for each hour (new Figure 8) instead of counting the total number of days over the domain. We kept the first time when reflectivity overpassed 55 dBZ as the motivation was actually to investigate where the first hail signals appear (e.g. near topography?).**

L397-409: The problem with this analysis is that it assumes that the first detection of reflectivity ≥55 dBZ corresponds to the initiation of convection. In fact, developing convection may travel some distance before it reaches such an intensity, particularly in the presence of strong background flow (which would be expected in high-shear environments that favour

severe storms). This is one reason why it would be advantageous to use a lower reflectivity threshold for identifying and tracking convective cells. At the very least, this caveat needs to be mentioned.

**Few thresholds were in fact used for the detection of convective cells and hailstorms. We kindly refer the reviewer to Section 3.3.**

L418: You can probably just say lengths "between 10 and 20 km". Using 10.1 km as the lower bound implicitly excludes tracks with a length >10 km but <10.1 km. Alternatively, if you want to be more precise, you could define a variable $L$ to represent track length and then use "10 < L ≤ 20 km" to represent this particular bin. Either way, the same change should be applied on L420.

**We used the sentence suggested by the reviewer: "a length L between 10 and 20 km" (L540).**

L423-426: How did these studies define hail cells? If they used a lower reflectivity threshold, they are likely to get longer hail tracks because they will be including storms at earlier and later stages of their life cycles and are also less likely to break up tracks where the reflectivity temporarily drops below 55 dBZ.

**They used 3D (not 2D) radar data and applied a different algorithm (lines 549-551) to identify hail cells.**

L427-428: While the results for hail track duration may be similar to those for track length, it would still be nice to include the results in Fig. 10. You could also (or alternatively) combine length and duration to compute storm motion estimates for each cell and examine the distribution of this.

**We added Figure 11 showing the mean duration and described the results in L552-554.**

L429-432: It would make more sense to compute the track width as the average diameter or the cell (computed over its lifetime) in the direction perpendicular to its movement, since this will actually correspond to the width of the underlying hail swath (under the assumption that hail falls where reflectivity ≥55 dBZ). If you make this change, I would strongly encourage you to include the results in Fig. 10.

**Only the track width maximum was stored by the algorithm.**

L433-436: Again, rather than considering cells at a particular time, why not use the whole swath? For orientation, you could compute it simply as the angle between the first and last points in the trajectory (similar to how you define track length). Alternatively, you could apply a line of best fit to the set of points defining the trajectory. I would recommend a Theil-Sen fit for this purpose, as it is less sensitive to outliers for small sample sizes, compared to a linear least squares fit. Also, you should note that the angle is defined as the direction from which the storm is coming and is measured clockwise from north.

**We described how the angle is defined in L559-561. The methods described by the reviewer fit for straight (e.g. undeviated) swaths only. In case of a change of storm direction (which was for example the case in July 2013 in Southwest Germany after a cell splitting) the swath is bending**

and shows multiple orientations. Thus, the orientation of the swath on a given time is more precise than a computation of the orientation between the starting and ending swath point.

L467-468: The key issue with the lack of 3D radar data is the inability to use more sophisticated proxies such as those based on echo-top height (e.g. POH) or vertical integrals of reflectivity (e.g. MESH), which generally show higher skill in hail prediction (e.g. Skripniková and Řezáčová2015; Kunz and Kugel 2015; Puskeiler et al. 2016).
**We added a comment in L592-594 about the key issue with the lack of 3D radar data and we have added the literature suggested by the reviewer.**

Figure 3: I would recommend presenting these results as annual hail frequencies rather than counts of the total number of hail days. This will make it easier to compare your results with climatologies for different periods (including the maps for 2006 and 2010 in Fig. 6) and in other parts of the world (e.g. Cintineo et al. 2012; Warren et al. 2020). Also, you should mask out those grid points that fall outside of radar coverage, as shown in Fig. 2.
**We followed this suggestion and presented the results as annual hail frequencies rather than the total number of hail days (new Figure 3).**

Figure 4: For the zoomed in view of the Massif Central region, it looks like you've just cut out and blown up a section of the fairly coarse-resolution image on the right. As such it's very difficult to make out the details in both the orography and the hail frequency contours. I don't think you need the map of the full study domain as this is already shown in Fig. 1. Instead, I would make this a multi-panel figure, showing zoomed-in views (at an appropriately high resolution) for several of the hail hotspots visible in Fig. 3 and discussed in section 4.1.
**We deleted the coarse resolution zoomed-in-view of the Massif Central and replaced it by a high-resolution Global relief map overlaid with hail frequency contours and wind flow during hail days (new Figure 4). The results were described in Section 4.1.**

Figure 5: Please present these data as the actual number of hail days for each year rather than the difference from the mean (which can easily be inferred).
**We presented the data as actual number of hail days for each year in the new Figure 5.**

Figure 7: As noted above, rather than the total number of hail days in each subdomain, you should plot the mean number of hail days (excluding points that are over the sea or outside radar coverage). The same change should be applied to Fig. 8.
**We changed the total number of hail days for the mean number of hail days for Figures 7 and 8.**

Figure 9: It would be nice to show these plots for other hours, rather than just 02 and 18 LT. For example, you could group the data into 3h blocks (00-03, 03-06, 06-09, 09-12, 12-15, 15-18, 18-21, and 21-00 LT) and present the results as an 8 panel figure.
**We added a 8-panel figure following this great idea (now Figure 9).**

**Technical Corrections**

I am not sure what the standard is for this journal, but in English (both American and British),
a period is used as the decimal separator and a comma (or sometimes a thin space) is used
to break up numbers of ten thousand or higher. For example, twelve-thousand
three-hundred and forty-five point six would be written as 12,345.6.
L10-11: Change "spatially most extended" to simply "longest".
L12: Change "implied" to "produced" or "were associated with".
L12-13: Change "2 Billions Euros" to "€2 billion".
L20: This is not the correct use of "respectively" - it should only be used when describing two
or more items that refer back to a previous statement. For example, "in northern Germany
and southern France, hail occurs most frequently in August and June, respectively". The
same comment applies to L96 and L269.
L23: Change "major part" to "majority".
L31: Puskeiler et al. (2016) consider the years 2005-2011, not 2004-2014.
L34: Change "criterions" to "criteria". Also "echo top" should be two words.
L46: Change "allows" to "allow".
L47: "sea" shouldn't be capitalized.
L213: Change "weaker" to "lower".
L229: Get rid of "recently".
L317: I think you meant to put "(Figure 7)" at the start of this line.
L386: The definition of overshooting tops should be given when they are introduced on L37.
L394: This doesn't need to be a new paragraph.
L421: Get rid of "including squall lines". Also, it should be "MCSs".
L446: "...from French and German national radar composites..."
L448: Duplication of "Mason".
L557: Get rid of "to" before "orography".
L466-467: Change "allows obtaining hail proxies" to "provides a proxy for hail occurrence".
Figure 1: Please use different line thicknesses, styles or colors to distinguish between
country and state/distinct borders. The same applies to Fig. 3, 4, and 6.
Figure 6: The colour bar is incorrectly labelled. The number of hail days in a single year will
always be an integer, so you don't need the range or the decimal place (i.e. the labels
should just be 1, 2, 3, ···, 12).
Figure 10: The x axes of these plots are incorrectly labelled. Each bin corresponds to a
range of values, so the tick labels should be located under the ticks to illustrate this. So, for
example, for panel (a) the ticks should be labelled 0, 10, 20, ···, 310.

**All the technical corrections listed above have been inserted in the new paper version.**

Furthermore, validation seems quite crucial here, as without it, the data set may not represent hail events. So I encourage the authors to present the validation done against the hail reports from ESWD and if possible to extend it.

**We reiterate that an identification of hail within the SCS tracks and especially a separation between hail and heavy rainfalls is not possible giving the lack of comprehensive hail observations and the use of a proxy. Concerning the validation of tracks with ESWD reports, we mentioned in the new manuscript the main results established recently by Kunz et al. (2020) who worked with the same dataset as in this study. By using ESWD reports in the vicinity of SCS tracks, the authors could separate the SCS-tracks into two categories: Hailstorms (having a ESWD report near a track) and other Convective storms (tracks without nearby ESWD report) but that may also include few hailstorms (L288-292). We finished by listing the different reasons why a track could not be associated with a hail report (L292-294).**

Other comments:
It would be good to provide a short background on hail formation, under what meteorological conditions we should expect hail events. This would help in understanding the interpretation of the presented results.

**We dedicated a paragraph (L18-36) on hail formation, describing the diverse processes and mechanisms leading to hail formation in convective clouds and added few assumptions about the ambient conditions favoring hailstorms in Europe.**

The data set includes radar data from two countries which goes some different processing procedures. To be sure this does not add any bias in results – is there any overlap region where analysis from both data sources can be compared?

**We added a comment on that in L182-184, where we mention that the Rhine Valley is an overlapped region between France and Germany. During hailstorms crossing both France and Germany, we compared reflectivity values measured from both German and French radars. Only small differences appeared between the radar reflectivity datasets. These differences are corrected with our tracking algorithm using an advection correction along the wind field, and it induces a smoothing of the reflectivity values.**

Lightening filter: "If high reflectivity during a day occurs without lightning, the values at the affected grid points are set to zero." (Line 127-128): why only during day? Are you sure this filter is not too aggressive? Can you provide any information on percent of hail storms that are not associate with lightening? if I understand correctly these storms will be filtered out from the analysis and it is important to verify their fraction is not substantial.

**We clarified in line 224 that the filter runs for day and nighttime (day in the sentence is meant as 24-hour period).**

Split and merge: the authors write that "Special attention is given to cell splitting and merging" (Line 154). Why is that? I did not find in the results any consideration of the splits and merges that were detected.

**We added an explanation about storm splitting and merging in lines 258-268 and mentioned that the results in this study include both of them.**

Line 277: "As shown in Figure 5 the annual variability is very high and without any trend" – for 10 years of data I would not consider a trend for 10 years of data.

**We deleted "without any trend" and replaced the sentence with : "Averaged over the entire investigation area, the annual number of hail days is between 72 (2010) and 103 (2006) with a mean of 86 (Figure 5)."**

Line 283-285: "large-scale lifting (e.g., related to differential vorticity advection) could have led to an increase in convective available potential energy (CAPE) and a low convective inhibition (CIN). The combination of high moisture in the boundary layer, low CIN, high CAPE and lifting mechanisms may give rise to a substantial increase in SCS." It is not clear if this is an assumption or analysis. Why not to check reanalysis data for CAPE, CIN, air moisture anomalies? without it, I think this sentence is too speculative.

**We deleted this speculative statement.**

General comment: the authors provide a very detailed description of the pattern shown in the figures. In my opinion this is too lengthy and could be shorten. I leave this however for the author decision.

**We did try to shorten some pattern descriptions and keep the essential parts only. However the area we investigate is large and the authors found it interesting to describe the hail variability towards Europe.**

List of all relevant changes made in the new manuscript

1. In subsections 2.1.1 and 2.1.2: We added a description of the processing stages performed to produce the French and German national radar composites. This summarizes long technical reports provided by the French and the German National Weather Services.

2. A relevant contribution to the new paper version is the investigation of the role of topography on hail formation. For this purpose, we computed the flow (wind speed and velocity) at 10m during hail days using ERA-5 reanalysis for all subdomains described in this study and computed each time the Froude Number to assess the flow deviation in the vicinity of the topography. For each subdomain, the flow, as well as hail hot spots contours were added on a high-resolution relief map (ETOPO-1). In the new manuscript version, we show the results for the subdomain including the Massif Central only (in Figure 4) in order to avoid a too lengthy manuscript. The description of the wind flow in the Massif Central region is available in L-316-146.

3. Figures: We changed the design of Figures 5, 7, 8, 10, 11 and 12 in the new paper version. Additionally, Figures 7 and 8 present now the average number of hail days rather than the total number of hail days. In the new manuscript, we also added a 8-panel Figure (Figure 9) of the locations of the first convective signatures detected by the tracking algorithm. Furthermore, in the new revised manuscript, all maps were projected on a Lambert Conformal Conic Projection rather than the previous Pseudo Plate Carree projection.

[revised manuscript text omitted]
 is located in the Eastern part of Bavaria. The north-to-south orientated Pre-Alpine valleys may lead to flow deviations and low level convergence, similarly to the Swabian Jura area. Also Alpine pumping, a secondary mountain-plain circulation (Weissmann et al., 2005) leading to flow towards the Alps and creating convergence zones, may play a decisive role for this distribution (Nisi et al., 2018).

In the northeast of Germany, a local maximum of up to 32 3.2 hail days is positioned over the Saxon Ore Mountains (referred to as A in Figure 1) South of the city of Dresden. Note, however, that this maximum is mainly caused by a high number of SCS in the year of 2007 (Piper, 2017), which was characterized by frequent upper air troughs over Western Europe and ridges over Central Europe (Wernli et al., 2010), leading to high-pressure gradients on the eastern part of Germany in combination a southeast-to-northeast flow regime from the Czech Republic (note that the almost same situation occurred in 2019).

The northwestern part of Germany, including the States of Hesse (region 2 in Figure 1) and Rhineland-Palatinate (region 3 in Figure 1), and the southern part of North Rhine-Westphalia (region 1 in Figure 1) are regions affected by approximately 1.4 hail days on average. The location of the hail patterns is partly caused by the local orography with a pronounced maximum in North Hesse that lies directly on the leeward side of the Westerwald low mountain range, which is characterized by rolling terrain.

In contrast to the various hail hot spots located exclusively in the continental domain
and preferably along the mountain's foothills, most of the minima are found along the coastlines,
where hail is also a year-round phenomenon (Dessens, 1986). Hail is rare along the Atlantic as
well as the Mediterranean coasts, in the latter case only 0.2 hail days. The large heat capacity of the water

[revised manuscript text omitted]

---

## Referee Report (RR1)

**Review for NHESS-2020-138, Revision 1**

**General Comments**

The authors have made significant revisions to the paper based on the reviewers' comments, including provision of technical details on the French and German radar datasets and an assessment of the Froude number on hail days over the Massif Central using reanalysis data. I would like to thank them for these efforts. However, some of the new material was very difficult to understand (particularly the details of the radar tracking algorithm), while other parts have raised further questions that need to be addressed. I have a large number of additional comments; however, I expect most of these to be fairly simple to address. As such I am recommending further minor revisions.

**Specific Comments**

L19-20: Hail can also be produced by single-cell "pulse" storms; however, large hail is almost always associated with organised convection, particularly supercells (Smith et al. 2012; Wapler et al. 2016)

L23-24: A decrease in $\theta_e$ with height indicates that the atmosphere is potentially unstable. According to Markowski and Richardson (2010), potential instability is not generally considered to play a role in the destabilisation of the atmosphere that precede convective initiation. The key instability for deep, moist convection is conditional instability, which exists when $\theta_e^*$ (the equivalent potential temperature that the environment would have if it were saturated) decreases with height. However, I don't think you need to get into these definitions; just note that conditional instability is needed for lifted parcels of air to become positively buoyant.

L28-30: Other studies you might mention here include Brooks et al. (2003), Johnson and Sugden (2014), and Taszarek et al. (2017).

L31-39: Since the Spanish Plume isn't mentioned again I don't think you need to discuss it in detail here.

L67-69: I know I requested you add the definition of overshooting tops (OTs), but it rather breaks up the flow of this sentence. Also, this definition is relevant to the visual identification of an OT, but in satellite imagery these features are typically identified based on infrared brightness temperature signatures. As such I would suggest getting rid of the definition and instead including something like the following "...such as overshooting tops (an indicator of strong convective updrafts) in satellite imagery…"

L70-71: The statement "the link between the observed quantities and hail occurrence at the surface is less reliable than using radar measurements" doesn't really make sense and in any case only really applies to satellite measurements, not model data. I would recommend making a more general statement in this sentence about these approaches, then noting the specific limitations of satellite- and model-based hail proxies in the subsequent sentences.

The key benefit of radar proxies over overshooting tops is that the former is a more direct measure of hail within a storm (presence of large reflectivities), whereas the latter just indicates the presence of a strong updraft that might produce large hail.

L73-75: I think there is an opportunity here to emphasise the unique nature of the hail climatology you present and, in doing so, better link these two paragraphs. You could note that the advantage of satellite- and model-based hail proxies is that they can cover a wide geographic area, whereas radar-based climatologies are typically limited to a single country or region. However, yours is, I believe, the first study to combine radar observations from multiple countries.

L96-98: Does this mean that, for a given time, a CAPPI was taken from each radar and these were composited onto a common grid by taking the maximum value at every grid point? If so, at what altitude are the CAPPIs?

L101-113: I don't think you need to mention specific projects such as Panthère or RHyTMME, though you could add the relevant references (Tabary, 2007; Beck and Bousquet, 2013) to the end of the opening sentence of this paragraph. All you really need to note here is how many radars were present in the network and when these were replaced or upgraded.

L117-119: It looks like complete coverage of Luxembourg is only provided by one of the German radars. Perhaps this discussion of the inclusion of Belgium and Luxembourg should be moved to the opening paragraph of section 2.1, since it relies on both French and German data.

L122-123: The dual-pol variables aren't used so no need to mention them here.

L129-130: It's not obvious to me how a VPR (vertical profile of reflectivity) could be used to account for beam broadening with range. My understanding is that VPR corrections are used to extrapolate measurements taken aloft to the surface. This issue is worse at long range because the beam height increases (due to both the non-zero elevation angle and the curvature of the Earth); however, this has nothing to do with beam broadening, which tends to lead to an increase in partial beam filling with range.

L135-141: Is this discussion of QPE and associated quality control relevant since you are using reflectivity data only? Or do you take the corrected rain rates and convert them back to reflectivities? If not then all this detail can probably go.

L154-155: As noted in my original review, these regions are covered, according to Fig. 2, but only by a single radar. In your reply, you say "Looking at long-term radar composites, the far north place near the Danish border and the southeastern part of Bavaria have no reflectivity values; we rather see some values near the location of the radar stations." This leaves two possibilities. Either the German radar composite excludes pixels that are covered by only a single site or both these regions happen to be affected by beam blockage. It would be good to know what the reason is.

L160: Is orographic shading actually corrected for (e.g. by interpolating from higher tilts) or are pixels that are affected by it simply masked or labelled as low quality?

L160-161: Again is the conversion from Z to R relevant, since you're only using reflectivity?

L169-170: How do you get a "terrain-following near-ground reflectivity"? Are VPRs used to extrapolate values to the surface? Also it doesn't make sense to say CAPPI here since CAPPI stands for "constant-altitude plan position indicator". Here the "constant altitude" typically refers to above radar level (ARL) or above sea level (ASL), rather than above ground level (AGL), which is what you'd have if the data really are "terrain-following".

L204: You should say 0.25° resolution rather than 31 km as that is what you used (as I can see from Fig. 4). I think 31 km refers to the underlying Gaussian grid from which the 0.25° products are derived.

L216-217: I'm not sure what you mean here; please rephrase.

L224: How are you defining "high reflectivity" here?

L236-239: I would suggest using $Z_{MAX}$ and $Z_{RC}$ to indicate the maximum reflectivity in an ROIP and the threshold for RCs, respectively.

L239-240: Presumably this means that if an RC's maximum reflectivity briefly (say for one or two time steps) drops below 55 dBZ it will be treated as two separate convective cells, is that correct? If so, this is an important limitation and should be noted.

L241-242: Change "surface area" to just "area" as the former implies a 3D object. Also, rather than "radar bins" I presume you mean pixels? In which case, this quantity should have units of km$^2$, both here and in Table 1. I would suggest adding symbols to represent these variables; maybe $A_{RC}$ for the minimum RC area and $A_{hail}$ for the minimum area with reflectivities ≥ 55 dBZ. Finally, can you provide any justification for the inclusion of these two criteria? It makes sense to filter out very small cells as these could be spurious, but is there much sensitivity to the specific choice of thresholds?

L251: FAR = b / (a + b) is the false alarm *ratio*, whereas the false alarm *rate* (also known as the probability of false detection) is defined as F = b / (b + d). It seems this mistake was present in Puskeiler et al. (2016) but must have been missed by that paper's reviewers. You could maybe add a note to this effect: "...the false alarm ratio (FAR; incorrectly labelled the false alarm *rate* in Puskeiler et al. 2016) was 0.4..."

L253-254: Change "The algorithm assigns the RC of the previous radar composite to the actual composite" to "The algorithm associates RCs between consecutive radar composites".

L255-256: What specifically are the similarity criteria?

L257: I'm guessing that the "velocity factor" is intended to account for uncertainties in the motion estimates. Is there any reason for using 0.6?

L258-260: The right-moving storm tends to be favoured in the northern hemisphere, whereas the left-moving storm is favoured in the southern hemisphere (due to mirrored shear profiles). However, I'm not sure that it's fair to say that "in most cases, the left-moving cell weakens very quickly". It may sometimes evolve into more of a multicell structure, while in unidirectional shear profiles both storms may persist as supercells. Furthermore, it's important to note that cell splitting may also occur due to changes in storm intensity that cause a single RC to break up (or vice versa in the case of cell merging).

L262-268: I'm afraid I really don't follow this explanation at all. It seems to imply that single cell associations where there is a significant change in cell area can be labeled as splits or mergers, but that doesn't make sense. A split would normally be defined where a cell at time $t$ can be associated with two or more cells at time $t + \Delta t$. In this case the choice has to be made as to whether all or only one (e.g. the largest or most intense) of the "child" cells inherit the history of the "parent" cell. Similarly, a merger is defined where multiple cells at time $t$ are associated with a single cell at time $t + \Delta t$. In this case, a choice has to be made how to assign a history to the child cell. I suggest you completely rework this description (possibly adding a schematic) to make it easier to understand.

L276-277: Motion vectors include both speed and direction, so I don't think you need to say "as well as the track direction of the convective cells".

L279: If I am understanding this correctly, the shift vector is a spatial increment rather than a velocity. In this case you shouldn't use d$u$ and d$v$, but rather d$x$ and d$y$. If it *is* a velocity vector then the components should be $u$ and $v$.

L281-287: Again, I found this explanation really hard to follow. I tried reading the equivalent explanation in Puskeiler et al. (2016), but that is equally perplexing. In my head, the way this type of procedure would work would be to take a cell at two consecutive times and then shift the early cell forward in time and the later cell backward in time and average the two. This would be done for multiple intermediate time steps in order to create a smooth track. However, it sounds like the procedure used here is considerably more complicated. Again, I think this needs a complete rewrite in order to make it comprehensible.

L318: Presumably by "hail days" you mean hail days within the subdomain shown in Fig. 4 as opposed to anywhere in your analysis domain. Please clarify in the text.

L320: Again, you should say 0.25° rather than 31 km.

L322: Here and elsewhere in this section you should change "hail days" to "hail days per year"

L326-327: Is this flow convergence perhaps the signature of a surface cold front or pressure trough? The circulation associated with a front/trough might favour the development of severe storms (through the associated generation of mid-level instability and advection of low-level moisture), but flow interactions with the Massif Central could still act as a focussing/initiation mechanism.

L337: Surely the winds should be averaged over the lowest 1200 m to be consistent with the definition of $H$?

L338-341: The definition of the Brunt–Väisälä frequency is pretty standard so I don't think you need to cite Huschke (1959). However, I would recommend that you modify the equation so that it is an expression for $N$ rather than $N^2$. Also, you can just use $\theta_v$ for the virtual temperature (you don't need to refer to it as "ambient").

L345-346: You might want to note that this deflection of the flow is unlikely to show up in Fig. 4 due to the fairly coarse resolution of the ERA5 reanalysis (the Massif Central will be much lower and smoother in the IFS than is shown in the figure).

L350-353: This sentence is overly long and should be rephrased. You could potentially get rid of the bit starting "by referring to"

L390-395: The Froude number that you obtain is actually a bit larger than that estimated by Kunz and Puskeiler (2010), even though they used a smaller value of $H$ (1000 m). Presumably then, your values of $U$ are larger and/or your values of $N$ are smaller. Can you comment on these differences?

L411: I would suggest changing "is partly caused by" to "show an association with" as causality has not been firmly established.

L415: I've checked an Puskeiler et al. (2016) don't actually examine interannual variability in hail frequency. Nisi et al. (2018) do, but they find much higher interannual variability compared to this study. Perhaps this reflects the much larger study domain considered here.

L430-431: You don't show maps for 2012 so this sentence can probably be deleted.

L438-440: I'm not sure this last sentence is needed. Given the size of your study domain it is hardly surprising that years with a below-average number of hail days overall could still have a few localised high-impact events. Perhaps you could simply rephrase what you have to make this point. Also 2013 is only slightly below average and the average is arguably dragged up by the anomalously high number of hail days in 2006.

L451: What you show in Fig. 7 isn't a 10-day moving average as the averaging windows don't overlap. Instead you could say that you calculate the average number of hail days for consecutive 10-day periods. Alternatively, given the short length of your climatology, you might consider just plotting the relative frequency of hail in each month. This might also make for an easier comparison with previous studies.

L468-469: Can it really be argued that these are distinct maxima? This could just be an artefact of the relatively short length of your climatology.

L473-474: It doesn't make sense to describe this as "a right skewed distribution" since you're not really talking about a distribution but a time series.

L479: Which of the aforementioned studies used hail pad data? What data sources did the other studies consider? And why would the hail pads being clustered "near the subdomains influenced by maritime air mass" lead to an earlier seasonal peak?

L484: It's not really right to say "confirmed" here as the Lukach and Delobbe (2013) study obviously came before yours and showed an earlier maximum. To me the seasonal cycle for subdomain BEL looks pretty flat, but this is because you have plotted the number of hail days rather than the relative frequency.

L486: Since you only consider the first time that CCTA2D detects the cell this analysis pertains more to the development of hail storms, rather than their overall diurnal cycle. As stated in my original review, it would make more sense to consider all times when a storm exceeded 55 dBZ as this would account for storms that persist for multiple hours. This would also make comparisons with previous studies easier, since I imagine most of these considered all hours with hailstorms, rather than just the hour in which storms developed.

L492: There's no need to keep reminding the reader where each region is located as this has already been stated and is shown in Fig. 3. The same comment applies to L495-496 and L498.

L495-496: The peak at 16 LT is only slightly above the values for the adjacent hours. Given the relatively short length of your climatology I'm not sure you can read too much into this difference.

L510-511: The peak at 18 LT is much later than what you and most other studies find. Can this difference be explained?

L512-531: I'm really not convinced that this section adds much, if anything, to the manuscript. The distribution of track onset locations seems pretty consistent with the overall distribution of hailstorms shown in Fig. 3, with fewer points overnight and during the morning and more points during the afternoon and evening (as one would expect from Fig. 8). Unless you can quantitatively show that some regions show a *disproportionately* high/low onset frequency for a given time (i.e. many more/less onsets than one would expect based on overall hailstorm frequency) I would suggest getting rid of section 4.4 altogether.

L513-515: Get rid of "and to distinguish between mechanisms triggering nighttime events and convection being triggered within the boundary layer occurring preferably in the afternoon and early evening" - it makes the sentence overly long and isn't needed.

L518: Hours less than 10 should be written as 00, 03, etc. The same comment applies to L519, L522, and 530.

L540: I would say "a length less than 10 km". There's no need to include the symbol "L" if you're not going to include it in an equation.

L541-542: Figure 10 only shows track lengths up to 100 km so perhaps the values you quote here should correspond to track lengths of 20–100 km and > 100 km, rather than 20–150 km and > 150 km.

L559-561: In my original review I suggested that it might make more sense for orientation to be computed either as the angle of a line connecting the first and last points in the track or by fitting a line of best fit to all points in the track. In your response you argued that these methods "fit for straight (e.g. undeviated) swaths only". However, your method still does not account for curved storm tracks since it only considers a single pair of points in the track (before and after the centre point). Furthermore, it is likely to be more sensitive to sudden changes in cell direction associated with splits/mergers or changes in cell area. Assuming you have the start and end positions of the cell track I would suggest using these to compute the orientation as it is simply but consistent with how track length is defined.

L563: Technically a west-to-southwest direction would be from 225 to 270°. As such I would just say "from between 200 and 260°."

L576-578: This sentence needs reworking. First, you should change "none or only several" to "only a few". Second, "along the European coastlines" isn't very specific and is arguably repetition since Brittany and north Germany could be classed as "along the European coastlines". Third, I would say "farther inland" rather than "far off the coasts" as the latter implies offshore. Finally, you should quote hail frequencies for both coastal and inland regions in days per year.

L580-581: "The high spatial variability in the number of radar-derived hail days and the increasing number around orographic structures…" - I'm not sure what you mean by this; please rephrase.

L583: Is the diurnal cycle of hail that different between different regions? It's hard to say from Fig. 8 because it plots the absolute number of hail days rather than the relative frequency.

L603-604: X-band radars are actually less suitable for hail detection because their signal is strongly attenuated by large precipitation particles. You might instead mention the use of dual-polarisation measurements, which can provide more accurate detection of hail (e.g. Heinselman and Ryzhkov, 2006).

Figure 1: In my previous review I requested that you use different line thicknesses or colours for country and region/state borders, so that readers less familiar with European geography can distinguish between the two; however, this change does not appear to have been made.

Figure 2: Since the X-band radars aren't included in your analysis they probably should be removed from this figure. You might also consider using different symbols for those radars that were replaced or upgraded to dual-polarisation during the study period.

Figure 3: As noted in my original review, a colorbar would be more appropriate than the individual blocks with value ranges, since these imply gaps (e.g. between 0.6 and 0.7 day per year). I would also suggest using an increment of 0.5 days per year as this is much more intuitive (0.5 days per year = once every two years). One more thing. In transforming the projection of this plot, the boxes defining the different regions appear to have become distorted, such that their edges don't properly line up. Can this be rectified?

Figure 4: It looks like you have two coastlines in this map - can you get rid of the coarse one? Again, I would suggest using an increment of 0.5 days per year for the hail frequency contours (values of 2.5, 3.0, and 3.5 days per year). Finally, could you add a box showing the area for which the Froude number was evaluated?

Figure 6: Again, use a colorbar rather than the individual blocks for each colour range (with increments of 1 day), if possible.

Figure 7: The shading under the curves isn't needed so I suggest getting rid of it. The same comment applies to Fig. 8. I would also shift the curves so that points are centred on each 10-day window, rather than at the end of the window.

Figure 8: The results in this and the previous figure might be better presented as relative frequencies for each month/hour rather than the absolute number of hail days, as this will allow you to better infer differences in the seasonal and diurnal cycles of hail between the five regions. Just a suggestion.

Figure 9: Hours less than 10 should be written as 00, 03, etc.

Figure 11: My first impression was that this figure, and the ones before and after, show relative frequency of the y axis. However, for this one the values for the first four bars alone add up to more than 100 %. Is this an error?

**Technical Corrections**

L4: Change "reflectivity radar data and lightning data" to "radar reflectivity and lightning data".

L8: Change "or" to "and".

L20: It should be "mesoscale convective *systems*" not "mesoscale convective *storms*".

L22: "subtle" not "subtile".

L30: It should be "aside from" rather than "aside of".

L94-95: "that requested some computation adjustment into the national radar composite" - I'm not sure what you mean here; please rephrase.

L95: Change "up to nowadays" to "onward". Also, should it be "2015 onward" rather than "2014 onward", since data were available for 2014?

L219: Change "Despite" to "Although" and "has included" to "includes".

L240: When referring to RCs you should use the pronoun "an" rather than "a" since the initialism is read as *arr-see*. There are many other occurrences in section 3.3 where this needs to be corrected.

L245: Change "BZ" to "dBZ".

L260: Change "after they have been splitted" to "after they have split" or "following the split".

L290: It should be "Kunz et al. (2020)" not "(Kunz et al., 2020)".

L293-294: I know what you're trying to say here but "prevail" isn't the right word. The key point is that there are far fewer hail reports in France, Belgium, and Luxembourg than in Germany.

L318: Get rid of "(in terms of speed and direction)".

L323-324: Get rid of "named" in both sets of parentheses.

L325: Change "northern" to "southerly" (wind directions refer to the direction from which the wind is blowing).

L355: "require_s_"

L357: Get rid of "referred to as". The same change should be applied on L377, 387, L401, and L404.

L363: "1424 m" (get rid of the period).

L364: Changed "mentioned as" to "labelled". The same change should be applied on L388.

L365-366: You should put "e.g." at the start of the list of references rather than having "among others" at the end.

L384: This shouldn't be a new paragraph.

L427: This shouldn't be a new paragraph.

L437: Change "entire" to "all of".

L458: Change "Pyrenean" to "Pyrenees".

L472: Get rid of "located in the very southwest of France" (this is obvious from the name).

L483: Change "upper western part of Belgium" to "much of Belgium".

L499: Change "reminds" to "remains".

L542: Get rid of the comma after "MCSs".

L548: This shouldn't be a new paragraph.

**References**

Brooks, H.E., Lee, J.W. and Craven, J.P. (2003) The spatial distribution of severe thunderstorm and tornado environments from global reanalysis data. Atmospheric Research, 67, 73–94.

Heinselman, P.L. and Ryzhkov, A.V. (2006) Validation of polarimetric hail detection. Weather and Forecasting, 21, 839–850.

Johnson, A.W. and Sugden, K.E. (2014) Evaluation of sounding-derived thermodynamic and wind-related parameters associated with large hail events. Electronic Journal of Severe Storms Meteorology, 9, 1–42.

Smith, B.T., Thompson, R.L., Grams, J.S., Broyles, C., and Brooks, H.E. (2012) Convective modes for significant severe thunderstorms in the contiguous United States. Part I: Storm classification and climatology. Weather and Forecasting, 27, 1114–1135.

Taszarek, M., Brooks, H.E. and Czernecki, B. (2017) Sounding-derived parameters associated with convective hazards in Europe. Monthly Weather Review, 145, 1511–1528.

Wapler, K., Hengstebeck, T., and Groenemeijer, P. (2016) Mesocyclones in Central Europe as seen by radar. Atmospheric Research, 168, 112–120.

---

## Author Response (AR2)

**Review for NHESS-2020-138, Revision 1**

**General Comments**

The authors have made significant revisions to the paper based on the reviewers'
comments, including provision of technical details on the French and German radar datasets
and an assessment of the Froude number on hail days over the Massif Central using
reanalysis data. I would like to thank them for these efforts. However, some of the new
material was very difficult to understand (particularly the details of the radar tracking
algorithm), while other parts have raised further questions that need to be addressed. I have
a large number of additional comments; however, I expect most of these to be fairly simple
to address. As such I am recommending further minor revisions.
**The authors would like to deeply thank the reviewer for the further time and effort spent on the**
revision of the manuscript. We appreciate very much the detailed and helpful comments provided
by the reviewer that helped for the understanding of the article. In the final version, we will
address each comment and questions raised by the reviewer.

**Specific Comments**

L19-20: Hail can also be produced by single-cell "pulse" storms; however, large hail is
almost always associated with organised convection, particularly supercells (Smith et al.
2012; Wapler et al. 2016)
**We added this statement to the final version of the manuscript.**

L23-24: A decrease in $\theta$ e with height indicates that the atmosphere is potentially unstable.
According to Markowski and Richardson (2010), potential instability is not generally
considered to play a role in the destabilisation of the atmosphere that precede convective
initiation. The key instability for deep, moist convection is conditional instability, which exists
when $\theta$ e * (the equivalent potential temperature that the environment would have if it were
saturated) decreases with height. However, I don't think you need to get into these
definitions; just note that conditional instability is needed for lifted parcels of air to become
positively buoyant.
**We changed lines 23-24 according to the reviewer's suggestion.**

L28-30: Other studies you might mention here include Brooks et al. (2003), Johnson and
Sugden (2014), and Taszarek et al. (2017).
**We included these additional references to the final version.**

L31-39: Since the Spanish Plume isn't mentioned again I don't think you need to discuss it in
detail here.
**We deleted the sentences related to the Spanish Plume.**

L67-69: I know I requested you add the definition of overshooting tops (OTs), but it rather

breaks up the flow of this sentence. Also, this definition is relevant to the visual identification of an OT, but in satellite imagery these features are typically identified based on infrared brightness temperature signatures. As such I would suggest getting rid of the definition and instead including something like the following "...such as overshooting tops (an indicator of strong convective updrafts) in satellite imagery…"

**We removed the definition of OTs and replaced it with "an indicator of strong convective updrafts" as suggested.**

L70-71: The statement "the link between the observed quantities and hail occurrence at the surface is less reliable than using radar measurements" doesn't really make sense and in any case only really applies to satellite measurements, not model data. I would recommend making a more general statement in this sentence about these approaches, then noting the specific limitations of satellite- and model-based hail proxies in the subsequent sentences. The key benefit of radar proxies over overshooting tops is that the former is a more direct measure of hail within a storm (presence of large reflectivities), whereas the latter just indicates the presence of a strong updraft that might produce large hail.

**We deleted the sentence "the link between the observed quantities and hail occurrence at the surface is less reliable than using radar measurements" and replaced it with the suggestion mentioned by the reviewer.**

L73-75: I think there is an opportunity here to emphasise the unique nature of the hail climatology you present and, in doing so, better link these two paragraphs. You could note that the advantage of satellite- and model-based hail proxies is that they can cover a wide geographic area, whereas radar-based climatologies are typically limited to a single country or region. However, yours is, I believe, the first study to combine radar observations from multiple countries.

**The authors followed this suggestion and made a transition between the two paragraphs.**

L96-98: Does this mean that, for a given time, a CAPPI was taken from each radar and these were composited onto a common grid by taking the maximum value at every grid point? If so, at what altitude are the CAPPIs?

**For each single radar, the MaxCAPPI is achieved by projecting the vertical maximum of the 3D radar data on a horizontal plane.**

L101-113: I don't think you need to mention specific projects such as Panthère or RHyTMME, though you could add the relevant references (Tabary, 2007; Beck and Bousquet, 2013) to the end of the opening sentence of this paragraph. All you really need to note here is how many radars were present in the network and when these were replaced or upgraded.

**We fully agree with the reviewer and removed the project names as well as their descriptions.**

L117-119: It looks like complete coverage of Luxembourg is only provided by one of the German radars. Perhaps this discussion of the inclusion of Belgium and Luxembourg should be moved to the opening paragraph of section 2.1, since it relies on both French and

German data.
**The German radar composite did not include Luxembourg. We moved the discussion after the history of the French radar composite.**

L122-123: The dual-pol variables aren't used so no need to mention them here.
**We removed the sentence with dual-pol variables.**

L129-130: It's not obvious to me how a VPR (vertical profile of reflectivity) could be used to account for beam broadening with range. My understanding is that VPR corrections are used to extrapolate measurements taken aloft to the surface. This issue is worse at long range because the beam height increases (due to both the non-zero elevation angle and the curvature of the Earth); however, this has nothing to do with beam broadening, which tends to lead to an increase in partial beam filling with range.
**Many thanks for pointing out a mistake. We mixed up two sentences. The VPR is used to correct the overestimation of reflectivity due to the bright band and to take the underestimation of reflectivity above the 0°C isotherm into account (Tabary et al., 2013); but the VPR is not used to correct beam widening. We changed the sentence in the manuscript.**

L135-141: Is this discussion of QPE and associated quality control relevant since you are using reflectivity data only? Or do you take the corrected rain rates and convert them back to reflectivities? If not then all this detail can probably go.
**We agree with the reviewer and deleted this part.**

L154-155: As noted in my original review, these regions are covered, according to Fig. 2, but only by a single radar. In your reply, you say "Looking at long-term radar composites, the far north place near the Danish border and the southeastern part of Bavaria have no reflectivity values; we rather see some values near the location of the radar stations." This leaves two possibilities. Either the German radar composite excludes pixels that are covered by only a single site or both these regions happen to be affected by beam blockage. It would be good to know what the reason is.
**We think that the southeastern part of Bavaria is affected by beam blockage, whereas the Danish border region is excluded from the German radar composite.**

L160: Is orographic shading actually corrected for (e.g. by interpolating from higher tilts) or are pixels that are affected by it simply masked or labelled as low quality?
**From our knowledge, the orographic shading is not corrected, but rather labelled as low quality in each single radar scan.**

L160-161: Again is the conversion from Z to R relevant, since you're only using reflectivity?
**We deleted "transformation of reflectivity Z to rain rate R".**

L169-170: How do you get a "terrain-following near-ground reflectivity"? Are VPRs used to extrapolate values to the surface? Also it doesn't make sense to say CAPPI here since CAPPI stands for "constant-altitude plan position indicator". Here the "constant altitude"

typically refers to above radar level (ARL) or above sea level (ASL), rather than above ground level (AGL), which is what you'd have if the data really are "terrain-following".
**Thanks for these precisions. We rephrased this sentence.**

L204: You should say 0.25° resolution rather than 31 km as that is what you used (as I can see from Fig. 4). I think 31 km refers to the underlying Gaussian grid from which the 0.25° products are derived.
**We agree with this statement and changed 31km for 0.25°.**

L216-217: I'm not sure what you mean here; please rephrase.
**We rephrased the sentence.**

L224: How are you defining "high reflectivity" here?
**Here, "high reflectivity" means a reflectivity value above 55 dBZ.**

L236-239: I would suggest using $Z$ MAX and $Z$ RC to indicate the maximum reflectivity in an ROIP and the threshold for RCs, respectively.
**We followed the suggestions of the reviewer and included the changes in the final paper version.**

L239-240: Presumably this means that if an RC's maximum reflectivity briefly (say for one or two time steps) drops below 55 dBZ it will be treated as two separate convective cells, is that correct? If so, this is an important limitation and should be noted.
**No, otherwise this would mean that the hailstreak lengths tend to be significantly underestimated.**

L241-242: Change "surface area" to just "area" as the former implies a 3D object. Also, rather than "radar bins" I presume you mean pixels? In which case, this quantity should have units of km 2 , both here and in Table 1. I would suggest adding symbols to represent these variables; maybe $A$ RC for the minimum RC area and $A$ hail for the minimum area with reflectivities $\geq$ 55 dBZ. Finally, can you provide any justification for the inclusion of these two criteria? It makes sense to filter out very small cells as these could be spurious, but is there much sensitivity to the specific choice of thresholds?
**We replaced "surface area" to just "area". In the same way, we changed "radar bins" to "pixels". We added km^2 units to "pixels" in the article and also included this unit to Table 1. We will not add further symbols here as we only mention once the RC minimum area and the minimum area with reflectivity $\geq$ 55 dBZ.**
After testing few thresholds for the RC minimum area, including the overlap of tracks on hail observations, the threshold of 5 km2 was the best fit for our study. Area inferior to 5 km2 leaded to too many non-convective cells as well as too many cells splittings, whereas RC with an area superior to 5 km2 gave us fewer tracks. We chose a minimum of 3 pixels with Z$\geq$ 55 dBZ in the RC, so that more than half of the RC is covered by non-spurious data.

L251: FAR = b / (a + b) is the false alarm *ratio* , whereas the false alarm *rate* (also known as

the probability of false detection) is defined as F = b / (b + d). It seems this mistake was present in Puskeiler et al. (2016) but must have been missed by that paper's reviewers. You could maybe add a note to this effect: "...the false alarm ratio (FAR; incorrectly labelled the false alarm *rate* in Puskeiler et al. 2016) was 0.4..."

**Many thanks for notifying us about this wrong definition. We wouldn't add "...the false alarm ratio (FAR; incorrectly labelled the false alarm *rate* in Puskeiler et al. 2016) was 0.4..." explicitly in the article. As noted in Wilks (2011), we quote: "The FAR has been called the false alarm **rate** (Barnes et al. 2009 sketch a history of the confusion) [..]".**

L253-254: Change "The algorithm assigns the RC of the previous radar composite to the actual composite" to "The algorithm associates RCs between consecutive radar composites".
**Changed.**

L255-256: What specifically are the similarity criteria?
**We reformulated this sentence.**

L257: I'm guessing that the "velocity factor" is intended to account for uncertainties in the motion estimates. Is there any reason for using 0.6?
**Right, the velocity factor is intended to account for uncertainties in the motion estimates. After testing several factors, 0.6 was the best fit in our study. The velocity factor of 0.6 was also used in Handwerker (2002).**

L258-260: The right-moving storm tends to be favoured in the northern hemisphere, whereas the left-moving storm is favoured in the southern hemisphere (due to mirrored shear profiles). However, I'm not sure that it's fair to say that "in most cases, the left-moving cell weakens very quickly". It may sometimes evolve into more of a multicell structure, while in unidirectional shear profiles both storms may persist as supercells. Furthermore, it's important to note that cell splitting may also occur due to changes in storm intensity that cause a single RC to break up (or vice versa in the case of cell merging).
**We rephrased the sentence including "left-moving cell" and added a comment about the cell splitting.**

L262-268: I'm afraid I really don't follow this explanation at all. It seems to imply that single cell associations where there is a significant change in cell area can be labeled as splits or mergers, but that doesn't make sense. A split would normally be defined where a cell at time $t$ can be associated with two or more cells at time $t + \Delta t$. In this case the choice has to be made as to whether all or only one (e.g. the largest or most intense) of the "child" cells inherit the history of the "parent" cell. Similarly, a merger is defined where multiple cells at time $t$ are associated with a single cell at time $t + \Delta t$. In this case, a choice has to be made how to assign a history to the child cell. I suggest you completely rework this description (possibly adding a schematic) to make it easier to understand.

**We apologize for the unclear explanations, and replaced/rephrased this part with the reviewer's suggestions. For the schematic, we kindly invite the reviewer to look at Fig. 5 in Handwerker (2002).**

L276-277: Motion vectors include both speed and direction, so I don't think you need to say "as well as the track direction of the convective cells".
**We deleted this part from the sentence.**

L279: If I am understanding this correctly, the shift vector is a spatial increment rather than a velocity. In this case you shouldn't use $du$ and $dv$, but rather $dx$ and $dy$. If it *is* a velocity vector then the components should be $u$ and $v$.
**No, the shift vector is a velocity with the unit m/s.**

L281-287: Again, I found this explanation really hard to follow. I tried reading the equivalent explanation in Puskeiler et al. (2016), but that is equally perplexing. In my head, the way this type of procedure would work would be to take a cell at two consecutive times and then shift the early cell forward in time and the later cell backward in time and average the two. This would be done for multiple intermediate time steps in order to create a smooth track. However, it sounds like the procedure used here is considerably more complicated. Again, I think this needs a complete rewrite in order to make it comprehensible.
**We agree with the reviewer that our explanations weren't clear enough. In fact the algorithm does not store each cell, but only record the track and the maximum reflectivity values. Thus, we decided to completely rewrite this part and to use some of the terms evoked by the reviewer. Equation 3 was removed to avoid confusions and to simplify the explanations.**

L318: Presumably by "hail days" you mean hail days within the subdomain shown in Fig. 4 as opposed to anywhere in your analysis domain. Please clarify in the text.
**Thanks. Changed.**

L320: Again, you should say $0.25°$ rather than 31 km.
**We changed 31km for $0.25°$.**

L322: Here and elsewhere in this section you should change "hail days" to "hail days per year"
**Good point, thanks for that. We changed all the relevant "hail days" to "hail days per year".**

L326-327: Is this flow convergence perhaps the signature of a surface cold front or pressure trough? The circulation associated with a front/trough might favour the development of severe storms (through the associated generation of mid-level instability and advection of low-level moisture), but flow interactions with the Massif Central could still act as a focussing/initiation mechanism.
**Without further investigations, it is still too early to affirm if the flow convergence is a signature of cold front or pressure trough.**

L337: Surely the winds should be averaged over the lowest 1200 m to be consistent with the definition of $H$?
**Decisive for flow effects evolving over complex terrain is not only the terrain height, but also the layers above. Therefore, the vertical (density-weighted) integration differs from the characteristic mountain height H. In the article, H is still set to 1300m. We changed the altitude of the Livradois mountains in the text.**

L338-341: The definition of the Brunt‑Väisälä frequency is pretty standard so I don't think you need to cite Huschke (1959). However, I would recommend that you modify the equation so that it is an expression for $N$ rather than $N\,2$. Also, you can just use $\theta$ v for the virtual temperature (you don't need to refer to it as "ambient").
**We deleted the citation of Huschke (1959) and changed $\theta$ va for $\theta$ v in the Brunt Vaisala equation. We also modified the equation and used N rather than N^2.**

L345-346: You might want to note that this deflection of the flow is unlikely to show up in Fig. 4 due to the fairly coarse resolution of the ERA5 reanalysis (the Massif Central will be much lower and smoother in the IFS than is shown in the figure).
**We added this thoughtful statement to the manuscript.**

L350-353: This sentence is overly long and should be rephrased. You could potentially get rid of the bit starting "by referring to"
**We rephrased the sentence and followed the reviewer suggestion.**

L390-395: The Froude number that you obtain is actually a bit larger than that estimated by Kunz and Puskeiler (2010), even though they used a smaller value of $H$ (1000 m). Presumably then, your values of $U$ are larger and/or your values of $N$ are smaller. Can you comment on these differences?
**There can be few reasons why the Froude Number we found is higher than in Kunz and Puskeiler (2010). First, the time period investigated is not the same: We calculated the Froude Number for 2005 until 2014, whereas Kunz and Puskeiler (2010) focused on 1997 up to 2007. Then, in this article, U is coming from era5 reanalysis whereas Kunz and Puskeiler (2010) used soundings data. After that, the region considered in this article does not include for example the Vosges mountains or the entire Rhine Valley. This might also explain the differences in U.**

L411: I would suggest changing "is partly caused by" to "show an association with" as causality has not been firmly established.
**We followed the reviewer suggestion.**

L415: I've checked an Puskeiler et al. (2016) don't actually examine interannual variability in hail frequency. Nisi et al. (2018) do, but they find much higher interannual variability compared to this study. Perhaps this reflects the much larger study domain considered here.
**Thanks for that. We deleted the reference Puskeiler et al. (2016).**

L430-431: You don't show maps for 2012 so this sentence can probably be deleted.

**Sentence deleted.**

L438-440: I'm not sure this last sentence is needed. Given the size of your study domain it is hardly surprising that years with a below-average number of hail days overall could still have a few localised high-impact events. Perhaps you could simply rephrase what you have to make this point. Also 2013 is only slightly below average and the average is arguably dragged up by the anomalously high number of hail days in 2006.
**We deleted the last sentence of the paragraph.**

L451: What you show in Fig. 7 isn't a 10-day moving average as the averaging windows don't overlap. Instead you could say that you calculate the average number of hail days for consecutive 10-day periods. Alternatively, given the short length of your climatology, you might consider just plotting the relative frequency of hail in each month. This might also make for an easier comparison with previous studies.
**The sentence "10-day moving average" was replaced by "average number of hail days for consecutive 10-day periods" as suggested. This replacement was also done everywhere else. Concerning the relative frequency, we understand the concern of the reviewer. Note that many authors focusing on hailstorms in Europe do use the number of hail days per year or the total number of hail days per grid points. (Puskeiler, 2013; Puskeiler et al. 2016; Lindloff, 2003 ; Kunz and Puskeiler, 2010; Lukach and Delobbe, 2017 among others).**

L468-469: Can it really be argued that these are distinct maxima? This could just be an artefact of the relatively short length of your climatology.
**Note that this statement is only correct for the period 2005 to 2014.**

L473-474: It doesn't make sense to describe this as "a right skewed distribution" since you're not really talking about a distribution but a time series.
**We removed "a right skewed distribution" from the manuscript.**

L479: Which of the aforementioned studies used hail pad data? What data sources did the other studies consider? And why would the hail pads being clustered "near the subdomains influenced by maritime air mass" lead to an earlier seasonal peak?
**All of the aforementioned studies used hailpad data. The hailpad network of ANELFA is located in Southwestern France, a region particularly influenced by maritime air masses.**
We deleted "Furthermore, the scattered network of hail pads is denser near the subdomains influenced by maritime air mass" from the manuscript.

L484: It's not really right to say "confirmed" here as the Lukach and Delobbe (2013) study obviously came before yours and showed an earlier maximum. To me the seasonal cycle for subdomain BEL looks pretty flat, but this is because you have plotted the number of hail days rather than the relative frequency.
**We replaced "confirmed" by "also found".**

L486: Since you only consider the first time that CCTA2D detects the cell this analysis

pertains more to the development of hail storms, rather than their overall diurnal cycle. As stated in my original review, it would make more sense to consider all times when a storm exceeded 55 dBZ as this would account for storms that persist for multiple hours. This would also make comparisons with previous studies easier, since I imagine most of these considered all hours with hailstorms, rather than just the hour in which storms developed.

\# We replaced "diurnal cycle of hailstorms" by "development of hail storms". We agree with the reviewer that to consider all times would permit to reconstruct the full diurnal cycle. Nevertheless, the main focus here was to compare when the first signatures of cells appear between all regions (When does the first signatures appear in mountainous area compared to regions near coastlines?). Note that the first signatures of cells detected by the algorithm allowed to study the synoptic environments favorable for hail development (Fluck, 2018).

L492: There's no need to keep reminding the reader where each region is located as this has already been stated and is shown in Fig. 3. The same comment applies to L495-496 and L498.

\# We deleted the region locations.

L495-496: The peak at 16 LT is only slightly above the values for the adjacent hours. Given the relatively short length of your climatology I'm not sure you can read too much into this difference.

\# We added "slightly" to the sentence.

L510-511: The peak at 18 LT is much later than what you and most other studies find. Can this difference be explained?

\# We replaced the sentence with the peak at 18 LT with "In this study we found a peak of hail around 16~LT in subdomain SWF, while Malafre, 2009 established a peak later in the afternoon, around 18~LT". Note that only the hail seasons of years 2004 and 2005 in the Ebro Valley were considered in the study of Malafre, 2009. The different regions and time period analyzed in Malafre (2009) and our study might explain the shift in time of the hail maximum.

L512-531: I'm really not convinced that this section adds much, if anything, to the manuscript. The distribution of track onset locations seems pretty consistent with the overall distribution of hailstorms shown in Fig. 3, with fewer points overnight and during the morning and more points during the afternoon and evening (as one would expect from Fig. 8). Unless you can quantitatively show that some regions show a *disproportionately* high/low onset frequency for a given time (i.e. many more/less onsets than one would expect based on overall hailstorm frequency) I would suggest getting rid of section 4.4 altogether.

\# We agree with the reviewer and deleted section 4.4 altogether.

L513-515: Get rid of "and to distinguish between mechanisms triggering nighttime events and convection being triggered within the boundary layer occurring preferably in the afternoon and early evening" - it makes the sentence overly long and isn't needed.

\# We actually think that this sentence explains why we investigated the spatial distribution of hailstorms.

L518: Hours less than 10 should be written as 00, 03, etc. The same comment applies to L519, L522, and 530.
**We added two digits to the hours less than 10.**

L540: I would say "a length less than 10 km". There's no need to include the symbol "L" if You're not going to include it in an equation.
**We removed "L" from the sentence.**

L541-542: Figure 10 only shows track lengths up to 100 km so perhaps the values you quote here should correspond to track lengths of 20‑100 km and > 100 km, rather than 20‑150 km and > 150 km.
**We agree and quote the lengths according to the reviewer suggestion.**

L559-561: In my original review I suggested that it might make more sense for orientation to be computed either as the angle of a line connecting the first and last points in the track or by fitting a line of best fit to all points in the track. In your response you argued that these methods "fit for straight (e.g. undeviated) swaths only". However, your method still does not account for curved storm tracks since it only considers a single pair of points in the track (before and after the centre point). Furthermore, it is likely to be more sensitive to sudden changes in cell direction associated with splits/mergers or changes in cell area. Assuming you have the start and end positions of the cell track I would suggest using these to compute the orientation as it is simply but consistent with how track length is defined.
**In the end, only the center of a swath was available for plotting. Sorry about that. We do like the ideas about the computation of the orientation very much, and might consider them in future investigations.**

L563: Technically a west-to-southwest direction would be from 225 to 270° . As such I would just say "from between 200 and 260° ."
**We deleted "a west-to-southwest direction" from the sentence.**

L576-578: This sentence needs reworking. First, you should change "none or only several" to "only a few". Second, "along the European coastlines" isn't very specific and is arguably repetition since Brittany and north Germany could be classed as "along the European coastlines". Third, I would say "farther inland" rather than "far off the coasts" as the latter implies offshore. Finally, you should quote hail frequencies for both coastal and inland regions in days per year.
**We considered most of these suggestions in the conclusion.**

L580-581: "The high spatial variability in the number of radar-derived hail days and the increasing number around orographic structures⋯" – I'm not sure what you mean by this; please rephrase.
**We removed the first part of the sentence and replaced the second part by "The large number of track onsets around orographic structures..".**

L583: Is the diurnal cycle of hail that different between different regions? It's hard to say from Fig. 8 because it plots the absolute number of hail days rather than the relative frequency.
**We replaced "significant" by "some".**

L603-604: X-band radars are actually less suitable for hail detection because their signal is strongly attenuated by large precipitation particles. You might instead mention the use of dual-polarisation measurements, which can provide more accurate detection of hail (e.g. Heinselman and Ryzhkov, 2006).
**We deleted "the recently installed X-band radars in the French Alps" and replaced it with "the use of dual-polarisation measurements" and added the reference indicated by the reviewer.**

Figure 1: In my previous review I requested that you use different line thicknesses or colours for country and region/state borders, so that readers less familiar with European geography can distinguish between the two; however, this change does not appear to have been made.
**We included the wrong figure in the revised version. Sorry about that. Country borders appear now with a thicker line in the final version of the manuscript.**

Figure 2: Since the X-band radars aren't included in your analysis they probably should be removed from this figure. You might also consider using different symbols for those radars that were replaced or upgraded to dual-polarisation during the study period.
**We removed the X-band radars from the figure. We did not use different symbols for the dual-polarisation radars for visibility purposes.**

Figure 3: As noted in my original review, a colorbar would be more appropriate than the individual blocks with value ranges, since these imply gaps (e.g. between 0.6 and 0.7 day per year). I would also suggest using an increment of 0.5 days per year as this is much more intuitive (0.5 days per year = once every two years). One more thing. In transforming the projection of this plot, the boxes defining the different regions appear to have become distorted, such that their edges don't properly line up. Can this be rectified?
**Thanks for these suggestions and for pointing out some misunderstandings. There are actually no gaps between the value ranges. For example the first interval ranges from 0 up to 0.699 hail days, even "0-6" is labelled in the legend. To avoid confusions, we added an additional digit to the upper limit of each interval in the legend.**
We have tried to plot this figure in many ways (with/without colorbars or individual blocks; and different increments of hail days). The authors have agreed that the increment of 0.6 hail days fits the best for the visualization of the regions affected by hail.
We fixed the projection issue of the boxes.

Figure 4: It looks like you have two coastlines in this map - can you get rid of the coarse one? Again, I would suggest using an increment of 0.5 days per year for the hail frequency contours (values of 2.5, 3.0, and 3.5 days per year). Finally, could you add a box showing the area for which the Froude number was evaluated?

**Good points! We deleted the coarse coastline. Contours with an increment of 0.5 days sometimes overlapped on each other. Therefore we decided not to touch the contours. Furthermore, we wanted to highlight the exact location of the high hail frequency (3.9) in the Massif Central region.**
Finally, concerning the box showing the area for which the Froude number was evaluated: The upper left corner of the box would overlap the legend and "break up" the wind vectors flow. To avoid any confusion, we decided not to represent the box in the Figure.

Figure 6: Again, use a colorbar rather than the individual blocks for each colour range (with increments of 1 day), if possible.
**Same comment as for Figure 3.**

Figure 7: The shading under the curves isn't needed so I suggest getting rid of it. The same comment applies to Fig. 8. I would also shift the curves so that points are centred on each 10-day window, rather than at the end of the window.
**We removed the shading under the curves for Figures 7 and 8. For figure 7, we centered the curves on each 10-days time window.**

Figure 8: The results in this and the previous figure might be better presented as relative frequencies for each month/hour rather than the absolute number of hail days, as this will allow you to better infer differences in the seasonal and diurnal cycles of hail between the five regions. Just a suggestion.
**We appreciate the suggestion very much, but we will keep the absolute number of hail days for now.**

Figure 9: Hours less than 10 should be written as 00, 03, etc.
**We rewrote the hours less than 10 with two digits.**

Figure 11: My first impression was that this figure, and the ones before and after, show relative frequency of the y axis. However, for this one the values for the first four bars alone add up to more than 100 %. Is this an error?
**Thanks a lot for this remark. Indeed there was an error in Figure 11 where the year 2014 was counted twice. We fixed that. Figures 10 and 12 are fine (we checked them again).**

**Technical Corrections**

L4: Change "reflectivity radar data and lightning data" to "radar reflectivity and lightning data".
L8: Change "or" to "and".
L20: It should be "mesoscale convective *systems*" not "mesoscale convective *storms*".
L22: "subtle" not "subtile".
L30: It should be "aside from" rather than "aside of".
L94-95: "that requested some computation adjustment into the national radar composite" - I'm not sure what you mean here; please rephrase.

**We change it for "that requested some additional time to calibrate each X-band radar and to implement their data into the national radar composite".**

L95: Change "up to nowadays" to "onward". Also, should it be "2015 onward" rather than
"2014 onward", since data were available for 2014?
L219: Change "Despite" to "Although" and "has included" to "includes".
L240: When referring to RCs you should use the pronoun "an" rather than "a" since the initialism is read as *arr-see*. There are many other occurrences in section 3.3 where this needs to be corrected.
L245: Change "BZ" to "dBZ".
L260: Change "after they have been splitted" to "after they have split" or "following the split".
L290: It should be "Kunz et al. (2020)" not "(Kunz et al., 2020)".
L293-294: I know what you're trying to say here but "prevail" isn't the right word. The key point is that there are far fewer hail reports in France, Belgium, and Luxembourg than in Germany.
L318: Get rid of "(in terms of speed and direction)".
L323-324: Get rid of "named" in both sets of parentheses.
L325: Change "northern" to "southerly" (wind directions refer to the direction from which the
wind is blowing).
L355: "require s"
L357: Get rid of "referred to as". The same change should be applied on L377, 387, L401, and L404.
L363: "1424 m" (get rid of the period).
L364: Changed "mentioned as" to "labelled". The same change should be applied on L388.
L365-366: You should put "e.g." at the start of the list of references rather than having "among others" at the end.
L384: This shouldn't be a new paragraph.
L427: This shouldn't be a new paragraph.
L437: Change "entire" to "all of".
L458: Change "Pyrenean" to "Pyrenees".
L472: Get rid of "located in the very southwest of France" (this is obvious from the name).
L483: Change "upper western part of Belgium" to "much of Belgium".
L499: Change "reminds" to "remains".
L542: Get rid of the comma after "MCSs".
L548: This shouldn't be a new paragraph.

**We are very thankful for the technical corrections that have all been included in the final version of the manuscript.**

**References**

Brooks, H.E., Lee, J.W. and Craven, J.P. (2003) The spatial distribution of severe thunderstorm and tornado environments from global reanalysis data. Atmospheric Research, 67, 73‑94.

Heinselman, P.L. and Ryzhkov, A.V. (2006) Validation of polarimetric hail detection. Weather and Forecasting, 21, 839‑850.

Johnson, A.W. and Sugden, K.E. (2014) Evaluation of sounding-derived thermodynamic and wind-related parameters associated with large hail events. Electronic Journal of Severe Storms Meteorology, 9, 1‑42.

Smith, B.T., Thompson, R.L., Grams, J.S., Broyles, C., and Brooks, H.E. (2012) Convective modes for significant severe thunderstorms in the contiguous United States. Part I: Storm classification and climatology. Weather and Forecasting, 27, 1114‑1135.

Taszarek, M., Brooks, H.E. and Czernecki, B. (2017) Sounding-derived parameters associated with convective hazards in Europe. Monthly Weather Review, 145, 1511‑1528.

Wapler, K., Hengstebeck, T., and Groenemeijer, P. (2016) Mesocyclones in Central Europe as seen by radar. Atmospheric Research, 168, 112‑120.

**The authors appreciate the additional references that were included in the final version.**